# Multi-Class $\mathcal{H}$-Consistency Bounds

**Pranjal Awasthi**
Google Research
New York, NY 10011
`pranjalawasthi@google.com`

**Anqi Mao**
Courant Institute
New York, NY 10012
`aqmao@cims.nyu.edu`

**Mehryar Mohri**
Google Research & Courant Institute
New York, NY 10011
`mohri@google.com`

**Yutao Zhong**
Courant Institute
New York, NY 10012
`yutao@cims.nyu.edu`

## Abstract

We present an extensive study of $\mathcal{H}$-*consistency bounds* for multi-class classification. These are upper bounds on the target loss estimation error of a predictor in a hypothesis set $\mathcal{H}$, expressed in terms of the surrogate loss estimation error of that predictor. They are stronger and more significant guarantees than Bayes-consistency, $\mathcal{H}$-calibration or $\mathcal{H}$-consistency, and more informative than excess error bounds derived for $\mathcal{H}$ being the family of all measurable functions. We give a series of new $\mathcal{H}$-consistency bounds for surrogate multi-class losses, including max losses, sum losses, and constrained losses, both in the non-adversarial and adversarial cases, and for different differentiable or convex auxiliary functions used. We also prove that no non-trivial $\mathcal{H}$-consistency bound can be given in some cases. To our knowledge, these are the first $\mathcal{H}$-consistency bounds proven for the multi-class setting. Our proof techniques are also novel and likely to be useful in the analysis of other such guarantees.

## 1 Introduction

The loss functions optimized by learning algorithms are often distinct from the original one specified for a task. This is typically because optimizing the original loss is computationally intractable or because it does not admit some favorable properties of differentiability or smoothness. As an example, the loss function minimized by the support vector machine (SVM) algorithm is the hinge loss (Cortes and Vapnik, 1995) or the one associated to AdaBoost is the exponential loss (Schapire and Freund, 2012), both distinct from the binary classification loss used as a benchmark in applications. But, what learning guarantees can we rely on when using a surrogate loss? This is a fundamental question in learning theory that directly relates to the design of algorithms.

The standard property of *Bayes-consistency*, which has been shown to hold for several surrogate losses (Zhang, 2004a,b; Bartlett, Jordan, and McAuliffe, 2006; Tewari and Bartlett, 2007; Steinwart, 2007), does not supply a sufficient guarantee, since it only ensures that, *asymptotically*, near optimal minimizers of the surrogate excess loss nearly optimally minimize the target excess error. Moreover, this asymptotic property only holds for the full family of measurable functions, which of course is distinct from the more restricted hypothesis set used by a learning algorithm. In fact, it has been shown by Long and Servedio (2013), both theoretically and empirically, that for some hypothesis sets and distributions, the expected error of an algorithm minimizing a Bayes-consistent loss is bounded below by a positive constant, while that of an algorithm minimizing an inconsistent loss goes to zero.

36th Conference on Neural Information Processing Systems (NeurIPS 2022).

This suggests that a hypothesis set-dependent notion of $\mathcal{H}$-consistency is more pertinent to the study of consistency for learning (Long and Servedio, 2013), which has been used by Kuznetsov et al. (2014); Cortes et al. (2016a,b) and Zhang and Agarwal (2020) and more generally by Awasthi, Frank, Mao, Mohri, and Zhong (2021a) in an extensive study of both binary classification and *adversarial* binary classification losses, as defined in (Goodfellow et al., 2014; Madry et al., 2017; Tsipras et al., 2018; Carlini and Wagner, 2017). Nevertheless, $\mathcal{H}$-consistency remains an asymptotic property and does not provide guarantees for approximate surrogate loss minimizers that rely on finite samples.

Awasthi, Mao, Mohri, and Zhong (2022a) recently presented a series of results providing $\mathcal{H}$-*consistency bounds* in binary classification. These are upper bounds on the target loss estimation error of a predictor in a hypothesis set $\mathcal{H}$, expressed in terms of the surrogate loss estimation error of that predictor. These guarantees are significantly stronger than the $\mathcal{H}$-calibration or $\mathcal{H}$-consistency properties studied by Awasthi et al. (2021b,c). They are also more informative than similar excess error bounds derived in the literature, which correspond to the special case where $\mathcal{H}$ is the family of all measurable functions (Zhang, 2004a; Bartlett et al., 2006; Mohri et al., 2018). Combining $\mathcal{H}$-consistency bounds with existing surrogate loss estimation bounds directly yields finite sample bounds on the estimation error for the original loss. See Appendix C for a more detailed discussion.

This paper presents an extensive study of $\mathcal{H}$-consistency bounds for multi-class classification. We show in Section 4.1 that, in general, no non-trivial $\mathcal{H}$-consistency bounds can be derived for multi-class *max losses* such as those of Crammer and Singer (2001), when used with a convex loss auxiliary function such as the hinge loss. On the positive side, we prove multi-class $\mathcal{H}$-consistency bounds for max losses under a realizability assumption and give multi-class $\mathcal{H}$-consistency bounds using as an auxiliary function the $\rho$-margin loss, without requiring a realizability assumption. For *sum losses*, that is multi-class losses such as that of Weston and Watkins (1998), we give a series of results, including a negative result when using as auxiliary function the hinge-loss, and $\mathcal{H}$-consistency bounds when using the exponential loss, the squared hinge-loss, and the $\rho$-margin loss (Section 4.2). We also present a series of results for the so-called *constrained losses*, such as the loss function adopted by Lee et al. (2004) in the analysis of multi-class SVM. Here, we prove multi-class $\mathcal{H}$-consistency bounds when using as an auxiliary function the hinge-loss, the squared hinge-loss, the exponential loss, and the $\rho$-margin loss (Section 4.3). We further give multi-class *adversarial* $\mathcal{H}$-consistency bounds for all three of the general multi-class losses just mentioned (max losses, sum losses and constrained losses) in Section 5.

We are not aware of any prior $\mathcal{H}$-consistency bound derived in the multi-class setting, even in the special case of $\mathcal{H}$ being the family of all measurable functions, whether in the non-adversarial or adversarial setting. All of our results are novel, including our proof techniques. Our results are given for the hypothesis set $\mathcal{H}$ being the family of all measurable functions, the family of linear functions, or the family of one-hidden-layer ReLU neural networks. The binary classification results of Awasthi et al. (2022a) do not readily extend to the multi-class setting since the study of calibration and conditional risk is more complex, the form of the surrogate losses is more diverse, and in general the analysis is more involved and requires entirely novel proof techniques in the multi-class setting (see Section 3 for a more detailed discussion of this point).

We give a detailed discussion of related work in Appendix A. We start with the introduction of several multi-class definitions, as well as key concepts and definitions related to the study of $\mathcal{H}$-consistency bounds (Section 2).

## 2 Preliminaries

We consider the familiar multi-class classification scenario with $c \geq 2$ classes. We denote by $\mathcal{X}$ the input space and by $\mathcal{Y} = \{1, \ldots, c\}$ the set of classes or categories. Let $\mathcal{H}$ be a hypothesis set of functions mapping from $\mathcal{X} \times \mathcal{Y}$ to $\mathbb{R}$. The label $\mathsf{h}(x)$ associated by a hypothesis $h \in \mathcal{H}$ to $x \in \mathcal{X}$ is the one with the largest score: $\mathsf{h}(x) = \operatorname{argmax}_{y \in \mathcal{Y}} h(x, y)$ with an arbitrary but fixed deterministic strategy used for breaking ties. For simplicity, we fix that strategy to be the one selecting the label with the highest index under the natural ordering of labels. See Appendix B for a more detailed discussion of this choice.

The *margin* $\rho_h(x, y)$ of a hypothesis $h \in \mathcal{H}$ for a labeled example $(x, y) \in \mathcal{X} \times \mathcal{Y}$ is defined by

$$\rho_h(x, y) = h(x, y) - \max_{y' \neq y} h(x, y'),$$

that is the difference between the score assigned to $(x, y)$ and that of the runner-up. Given a distribution $\mathcal{D}$ over $\mathcal{X} \times \mathcal{Y}$ and a loss function $\ell \colon \mathcal{H} \times \mathcal{X} \times \mathcal{Y} \to \mathbb{R}$, the *generalization error* of a hypothesis $h \in \mathcal{H}$ and the *minimal generalization error* are defined as follows:

$$\mathcal{R}_\ell(h) = \mathop{\mathbb{E}}_{(x,y) \sim \mathcal{D}} [\ell(h, x, y)] \quad \text{and} \quad \mathcal{R}^*_{\ell, \mathcal{H}} = \inf_{h \in \mathcal{H}} \mathcal{R}_\ell(h).$$

The goal in multi-class classification is to select a hypothesis $h \in \mathcal{H}$ with small generalization error with respect to the multi-class $0/1$ loss defined, for any $h \in \mathcal{H}$, by $\ell_{0-1}(h, x, y) = \mathbb{1}_{\mathsf{h}(x) \neq y}$. In the adversarial scenario, the goal is to select a hypothesis $h \in \mathcal{H}$ with small *adversarial generalization error* defined, for any $\gamma \in (0, 1)$ and $p \in [1, +\infty]$, by $\mathcal{R}_{\ell_\gamma}(h) = \mathbb{E}_{(x,y) \sim \mathcal{D}}[\ell_\gamma(h, x, y)]$, where

$$\ell_\gamma(h, x, y) = \sup_{x' : \|x - x'\|_p \leq \gamma} \mathbb{1}_{\rho_h(x', y) \leq 0} = \mathbb{1}_{\inf_{x' : \|x - x'\|_p \leq \gamma} \rho_h(x', y) \leq 0},$$

is the adversarial multi-class $0/1$ loss. More generally, the *adversarial generalization error* and *minimal adversarial generalization error* for a loss function $\ell(h, x, y)$ are defined as follows:

$$\mathcal{R}_{\widetilde{\ell}}(h) = \mathop{\mathbb{E}}_{(x,y) \sim \mathcal{D}} \big[\widetilde{\ell}(h, x, y)\big] \quad \text{and} \quad \mathcal{R}^*_{\widetilde{\ell}, \mathcal{H}} = \inf_{h \in \mathcal{H}} \mathcal{R}_{\widetilde{\ell}}(h),$$

where $\widetilde{\ell}(h, x, y) = \sup_{x' : \|x - x'\|_p \leq \gamma} \ell(h, x', y)$ is the supremum-based counterpart of $\ell$.

For a distribution $\mathcal{D}$ over $\mathcal{X} \times \mathcal{Y}$, we define, for any $x \in \mathcal{X}$, $p(x) = (p(x, 1), \ldots, p(x, c))$, where $p(x, y) = \mathcal{D}(Y = y \mid X = x)$ is the conditional probability of $Y = y$ given $X = x$. We can then write the generalization error as $\mathcal{R}_\ell(h) = \mathbb{E}_X[\mathcal{C}_\ell(h, x)]$, where $\mathcal{C}_\ell(h, x)$ is the *conditional $\ell$-risk* defined by $\mathcal{C}_\ell(h, x) = \sum_{y \in \mathcal{Y}} p(x, y) \ell(h, x, y)$. We will denote by $\mathcal{P}$ a set of distributions $\mathcal{D}$ over $\mathcal{X} \times \mathcal{Y}$ and by $\mathcal{P}_{\text{all}}$ the set of all such distributions. For convenience, we define $y_{\max}$ by $y_{\max} = \operatorname{argmax}_{y \in \mathcal{Y}} p(x, y)$. When there is a tie, we pick the label with the highest index under the natural ordering of labels.

The *minimal conditional $\ell$-risk* is denoted by $\mathcal{C}^*_{\ell, \mathcal{H}}(x) = \inf_{h \in \mathcal{H}} \mathcal{C}_\ell(h, x)$. We also use the following shorthand for the gap $\Delta \mathcal{C}_{\ell, \mathcal{H}}(h, x) = \mathcal{C}_\ell(h, x) - \mathcal{C}^*_{\ell, \mathcal{H}}(x)$ and call $\Delta \mathcal{C}_{\ell, \mathcal{H}}(h, x) \mathbb{1}_{\Delta \mathcal{C}_{\ell, \mathcal{H}}(h, x) > \epsilon}$ the *conditional $\epsilon$-regret* for $\ell$. For convenience, we also define, for any vector $\tau = (\tau_1, \ldots, \tau_c)$ in the probability simplex of $\mathbb{R}^c$, $\mathcal{C}_\ell(h, x, \tau) = \sum_{y \in \mathcal{Y}} \tau_y \ell(h, x, y)$, $\mathcal{C}^*_{\ell, \mathcal{H}}(x, \tau) = \inf_{h \in \mathcal{H}} \mathcal{C}_\ell(h, x, \tau)$ and $\Delta \mathcal{C}_{\ell, \mathcal{H}}(h, x, \tau) = \mathcal{C}_\ell(h, x, \tau) - \mathcal{C}^*_{\ell, \mathcal{H}}(x, \tau)$. Thus, we have $\Delta \mathcal{C}_{\ell, \mathcal{H}}(h, x, p(x)) = \Delta \mathcal{C}_{\ell, \mathcal{H}}(h, x)$. For any $\epsilon > 0$, we will denote by $[t]_\epsilon$ the $\epsilon$-truncation of $t \in \mathbb{R}$ defined by $t \mathbb{1}_{t > \epsilon}$. Thus, the conditional $\epsilon$-regret can be rewritten as $\big[\Delta \mathcal{C}_{\ell, \mathcal{H}}(h, x)\big]_\epsilon$.

For a hypothesis set $\mathcal{H}$ and distribution $\mathcal{D}$, we also define the $(\ell, \mathcal{H})$-*minimizability gap* as $\mathcal{M}_{\ell, \mathcal{H}} = \mathcal{R}^*_{\ell, \mathcal{H}} - \mathbb{E}_X\big[\mathcal{C}^*_{\ell, \mathcal{H}}(x)\big]$, that is the difference between the best-in class error and the expectation of the minimal conditional $\ell$-risk. This is a key quantity appearing in our bounds that we cannot hope to estimate or minimize. Its value only depends on the distribution $\mathcal{D}$ and the hypothesis set $\mathcal{H}$. As an example, when $\mathcal{H}$ is the family of all measurable functions, then the minimizability gap for the multi-class $0/1$ loss is zero for any distribution $\mathcal{D}$.

## 3  General theorems

The general form of the $\mathcal{H}$-*consistency bounds* that we are seeking for a surrogate loss $\ell_1$ of a target loss $\ell_2$ is $\mathcal{R}_{\ell_2}(h) - \mathcal{R}^*_{\ell_2, \mathcal{H}} \leq f(\mathcal{R}_{\ell_1}(h) - \mathcal{R}^*_{\ell_1, \mathcal{H}})$ for all $h \in \mathcal{H}$, for some non-decreasing function $f$. To derive such bounds for surrogate multi-class losses, we draw on the following two general theorems, which show that, under some conditions, the target loss estimation error can be bounded by some functional form of the surrogate loss estimation error involving minimizability gaps.

**Theorem 1 (Distribution-dependent $\Psi$-bound).** *Assume that there exists a convex function $\Psi \colon \mathbb{R}_+ \to \mathbb{R}$ with $\Psi(0) \geq 0$ and $\epsilon \geq 0$ such that the following holds for all $h \in \mathcal{H}$, $x \in \mathcal{X}$ and $\mathcal{D} \in \mathcal{P}$: $\Psi\big(\big[\Delta \mathcal{C}_{\ell_2, \mathcal{H}}(h, x)\big]_\epsilon\big) \leq \Delta \mathcal{C}_{\ell_1, \mathcal{H}}(h, x)$. Then, for any hypothesis $h \in \mathcal{H}$ and any distribution $\mathcal{D} \in \mathcal{P}$,*

$$\Psi\big(\mathcal{R}_{\ell_2}(h) - \mathcal{R}^*_{\ell_2, \mathcal{H}} + \mathcal{M}_{\ell_2, \mathcal{H}}\big) \leq \mathcal{R}_{\ell_1}(h) - \mathcal{R}^*_{\ell_1, \mathcal{H}} + \mathcal{M}_{\ell_1, \mathcal{H}} + \max\{\Psi(0), \Psi(\epsilon)\}.$$

**Theorem 2 (Distribution-dependent $\Gamma$-bound).** *Assume that there exists a concave function $\Gamma \colon \mathbb{R}_+ \to \mathbb{R}$ and $\epsilon \geq 0$ such that the following holds for all $h \in \mathcal{H}$, $x \in \mathcal{X}$ and $\mathcal{D} \in \mathcal{P}$: $\big[\Delta \mathcal{C}_{\ell_2, \mathcal{H}}(h, x)\big]_\epsilon \leq \Gamma(\Delta \mathcal{C}_{\ell_1, \mathcal{H}}(h, x))$. Then, for any hypothesis $h \in \mathcal{H}$ and any distribution $\mathcal{D} \in \mathcal{P}$,*

$$\mathcal{R}_{\ell_2}(h) - \mathcal{R}^*_{\ell_2, \mathcal{H}} \leq \Gamma\big(\mathcal{R}_{\ell_1}(h) - \mathcal{R}^*_{\ell_1, \mathcal{H}} + \mathcal{M}_{\ell_1, \mathcal{H}}\big) - \mathcal{M}_{\ell_2, \mathcal{H}} + \epsilon.$$

The theorems show that, to derive such bounds for a specific hypothesis set and a set of distributions, it suffices to verify that for the same hypothesis set and set of distributions, the conditional $\epsilon$-regret for the target loss can be upper bounded with the same functional form of the gap between the conditional risk and minimal conditional risk of the surrogate loss. These results are similar to their binary classification counterparts due to Awasthi et al. (2022b). In particular, the conditional $\ell$-risk $\mathcal{C}_\ell(h, x)$ in our theorems is the multi-class generalization of their binary definition. The proofs are similar and are included in Appendix E for completeness.

For a given hypothesis set $\mathcal{H}$, the resulting bounds suggest three key ingredients for the choice of a surrogate loss: (1) the functional form of the $\mathcal{H}$-consistency bound, which is specified by the function $\Psi$ or $\Gamma$; (2) the smoothness of the loss and more generally its optimization virtues, as needed for the minimization of $\mathcal{R}_{\ell_1}(h) - \mathcal{R}^*_{\ell_1, \mathcal{H}}$; (3) and the approximation properties of the surrogate loss function which determine the value of the minimizability gap $\mathcal{M}_{\ell_1, \mathcal{H}}$. Our quantitative $\mathcal{H}$-consistency bounds can help select the most favorable surrogate loss function among surrogate losses with good optimization merits and comparable approximation properties.

In Section 4 and Section 5, we will apply Theorem 1 and Theorem 2 to the analysis of multi-class loss functions and hypothesis sets widely used in practice. Here, we wish to first comment on the novelty of our results and proof techniques. Let us emphasize that although the general tools of Theorems 1 and 2 are the multi-class generalization of that in (Awasthi et al., 2022a), the binary classification results of Awasthi et al. (2022a) do not readily extend to the multi-class setting. This is true, even in the classical study of Bayes-consistency, where the multi-class setting (Tewari and Bartlett, 2007) does not readily follow the binary case (Bartlett et al., 2006) and required an alternative analysis and new proofs. Note that, additionally, in the multi-class setting, surrogate losses are more diverse: we will distinguish max losses, sum losses, and constrained losses and present an analysis for each loss family with various auxiliary functions for each (see Section 4).

**Proof techniques**. More specifically, the need for novel proof techniques stems from the following. To use Theorem 1 and Theorem 2, we need to find $\Psi$ and $\Gamma$ such that the inequality conditions in these theorems hold. This requires us to characterize the conditional risk and the minimal conditional risk of the multi-class zero-one loss function and the corresponding ones for diverse surrogate loss functions in both the non-adversarial and adversarial scenario. Unlike the binary case, such a characterization in the multi-class setting is very difficult. For example, for the constrained loss, solving the minimal conditional risk given a hypothesis set is equivalent to solving a $c$-dimensional constrained optimization problem, which does not admit an analytical expression. In contrast, in the binary case, solving the minimal conditional risk is equivalent to solving a minimization problem for a univariate function and the needed function $\Psi$ can be characterized explicitly by the $\mathcal{H}$-estimation error transformation, as shown in (Awasthi et al., 2022a). Unfortunately, such binary classification transformation tools cannot be adapted to the multi-class setting. Instead, in our proof for the multi-class setting, we adopt a new idea that avoids directly characterizing the explicit expression of the minimal conditional risk.

For example, for the constrained loss, we leverage the condition of (Lee et al., 2004) that the scores sum to zero, and appropriately choose a hypothesis $\overline{h}$ that differs from $h$ only by its scores for $\mathsf{h}(x)$ and $y_{\max}$ (see Appendix K). Then, we can upper bound the minimal conditional risk by the conditional risk of $\overline{h}$ without having to derive the closed form expression of the minimal conditional risk. Therefore, the conditional regret of the surrogate loss can be lower bounded by that of the zero-one loss with an appropriate function $\Psi$. To the best of our knowledge, this proof idea and technique are entirely novel. We believe that they can be used for the analysis of other multi-class surrogate losses. Furthermore, all of our multi-class $\mathcal{H}$-consistency results are new. Likewise, our proofs of the $\mathcal{H}$-consistency bounds for sum losses for the squared hinge loss and exponential loss use similarly a new technique and idea, and so does the proof for the $\rho$-margin loss. Furthermore, we also present an analysis of the adversarial scenario (see Section 5), for which the multi-class proofs are also novel. Finally, our bounds in the multi-class setting are more general: for $c = 2$, we recover the binary classification bounds of (Awasthi et al., 2022a). Thus, our bounds benefit from the same tightness guarantees shown by (Awasthi et al., 2022a). A further analysis of the tightness of our guarantees in the multi-class setting is left to future work.

# 4 $\mathcal{H}$-consistency bounds

In this section, we discuss $\mathcal{H}$-consistency bounds in the non-adversarial scenario where the target loss $\ell_2$ is $\ell_{0-1}$, the multi-class $0/1$ loss. The lemma stated next characterizes the minimal conditional $\ell_{0-1}$-risk and the corresponding conditional $\epsilon$-regret, which will be helpful for instantiating Theorems 1 and 2 in the non-adversarial scenario. For any $x \in \mathcal{X}$, we will denote, by $\mathsf{H}(x)$ the set of labels generated by hypotheses in $\mathcal{H}$: $\mathsf{H}(x) = \{\mathsf{h}(x) : h \in \mathcal{H}\}$.

**Lemma 3.** *For any $x \in \mathcal{X}$, the minimal conditional $\ell_{0-1}$-risk and the conditional $\epsilon$-regret for $\ell_{0-1}$ can be expressed as follows:*

$$\mathcal{C}^*_{\ell_{0-1},\mathcal{H}}(x) = 1 - \max_{y \in \mathsf{H}(x)} p(x,y)$$

$$\left[\Delta \mathcal{C}_{\ell_{0-1},\mathcal{H}}(h,x)\right]_\epsilon = \left[\max_{y \in \mathsf{H}(x)} p(x,y) - p(x,\mathsf{h}(x))\right]_\epsilon.$$

The proof of Lemma 3 is given in Appendix F. By Lemma 3, Theorems 1 and 2 can be instantiated as Theorems 4 and 5 in the non-adversarial scenario as follows, where $\mathcal{H}$-consistency bounds are provided between the multi-class $0/1$ loss and a surrogate loss $\ell$.

**Theorem 4** (**Non-adversarial distribution-dependent $\Psi$-bound**). *Assume that there exists a convex function $\Psi \colon \mathbb{R}_+ \to \mathbb{R}$ with $\Psi(0) \geq 0$ and $\epsilon \geq 0$ such that the following holds for all $h \in \mathcal{H}$, $x \in \mathcal{X}$ and $\mathcal{D} \in \mathcal{P}$:*

$$\Psi\left(\left[\max_{y \in \mathsf{H}(x)} p(x,y) - p(x,\mathsf{h}(x))\right]_\epsilon\right) \leq \Delta \mathcal{C}_{\ell,\mathcal{H}}(h,x). \tag{1}$$

*Then, for any hypothesis $h \in \mathcal{H}$ and any distribution $\mathcal{D} \in \mathcal{P}$, we have*

$$\Psi\left(\mathcal{R}_{\ell_{0-1}}(h) - \mathcal{R}^*_{\ell_{0-1},\mathcal{H}} + \mathcal{M}_{\ell_{0-1},\mathcal{H}}\right) \leq \mathcal{R}_\ell(h) - \mathcal{R}^*_{\ell,\mathcal{H}} + \mathcal{M}_{\ell,\mathcal{H}} + \max\{\Psi(0), \Psi(\epsilon)\}. \tag{2}$$

**Theorem 5** (**Non-adversarial distribution-dependent $\Gamma$-bound**). *Assume that there exists a concave function $\Gamma \colon \mathbb{R}_+ \to \mathbb{R}$ and $\epsilon \geq 0$ such that the following holds for all $h \in \mathcal{H}$, $x \in \mathcal{X}$ and $\mathcal{D} \in \mathcal{P}$:*

$$\left[\max_{y \in \mathsf{H}(x)} p(x,y) - p(x,\mathsf{h}(x))\right]_\epsilon \leq \Gamma(\Delta \mathcal{C}_{\ell,\mathcal{H}}(h,x)). \tag{3}$$

*Then, for any hypothesis $h \in \mathcal{H}$ and any distribution $\mathcal{D} \in \mathcal{P}$, we have*

$$\mathcal{R}_{\ell_{0-1}}(h) - \mathcal{R}^*_{\ell_{0-1},\mathcal{H}} \leq \Gamma\left(\mathcal{R}_\ell(h) - \mathcal{R}^*_{\ell,\mathcal{H}} + \mathcal{M}_{\ell,\mathcal{H}}\right) - \mathcal{M}_{\ell_{0-1},\mathcal{H}} + \epsilon. \tag{4}$$

In the following, we will apply Theorems 4 and 5 to study the $\mathcal{H}$-consistency bounds for different families of multi-class losses parameterized by various auxiliary functions, for several general hypothesis sets. It is worth emphasizing that the form of the surrogate losses is more diverse in the multi-class setting and each case requires a careful analysis and that the techniques used in the binary case (Awasthi et al., 2022a) do not apply and cannot be readily extended to our case.

**Hypothesis sets**. Let $B_p^d(r) = \left\{z \in \mathbb{R}^d \mid \|z\|_p \leq r\right\}$ denote the $d$-dimensional $\ell_p$-ball with radius $r$, with $p \in [1, +\infty]$. Without loss of generality, in the following, we choose $\mathcal{X} = B_p^d(1)$. Let $p, q \in [1, +\infty]$ be conjugate indices, that is $\frac{1}{p} + \frac{1}{q} = 1$. In the following, we will specifically study three families: the family of all measurable functions $\mathcal{H}_{\text{all}}$, the family of linear hypotheses

$$\mathcal{H}_{\text{lin}} = \left\{(x,y) \mapsto w_y \cdot x + b_y \mid \|w_y\|_q \leq W, |b_y| \leq B\right\},$$

and that of one-hidden-layer ReLU networks defined by the following, where $(\cdot)_+ = \max(\cdot, 0)$:

$$\mathcal{H}_{\text{NN}} = \left\{(x,y) \mapsto \sum_{j=1}^n u_{y,j}(w_{y,j} \cdot x + b_{y,j})_+ \mid \|u_y\|_1 \leq \Lambda, \|w_{y,j}\|_q \leq W, |b_{y,j}| \leq B\right\}.$$

**Multi-class loss families**. We will study three broad families of multi-class loss functions: *max losses*, *sum losses* and *constrained losses*, each parameterized by an auxiliary function $\Phi$ on $\mathbb{R}$, assumed to be non-increasing and non-negative. In particular, we will consider the following

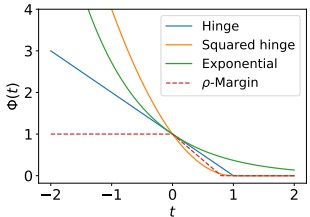 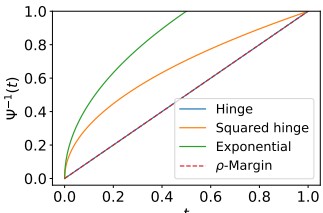

Figure 1: Left: auxiliary functions with $\rho = 0.8$. Right: $\mathcal{H}$-consistency dependence between $\ell_{0-1}$ and $\Phi^{\text{cstnd}}$ with $\rho = 0.8$.

common auxiliary functions: the hinge loss $\Phi_{\text{hinge}}(t) = \max\{0, 1 - t\}$, the squared hinge loss $\Phi_{\text{sq-hinge}}(t) = \max\{0, 1 - t\}^2$, the exponential loss $\Phi_{\text{exp}}(t) = e^{-t}$, and the $\rho$-margin loss $\Phi_\rho(t) = \min\{\max\{0, 1 - t/\rho\}, 1\}$. Note that the first three auxiliary functions are convex, while the last one is not. Figure 1 shows plots of these auxiliary functions.

We will say that a hypothesis set $\mathcal{H}$ is *symmetric* if there exists a family $\mathcal{F}$ of functions $f$ mapping from $\mathcal{X}$ to $\mathbb{R}$ such that $\{[h(x, 1), \ldots, h(x, c)]: h \in \mathcal{H}\} = \{[f_1(x), \ldots, f_c(x)]: f_1, \ldots, f_c \in \mathcal{F}\}$ and $|\{f(x): f \in \mathcal{F}\}| \geq 2$ for any $x \in \mathcal{X}$. The hypothesis sets defined above ($\mathcal{H}_{\text{all}}$, $\mathcal{H}_{\text{lin}}$ and $\mathcal{H}_{\text{NN}}$) are all symmetric. Note that for a symmetric hypothesis set $\mathcal{H}$, we have $\mathsf{H}(x) = \mathcal{Y}$.

We will say that a hypothesis set $\mathcal{H}$ is *complete* if the set of scores it generates spans $\mathbb{R}$, that is, $\{h(x, y): h \in \mathcal{H}\} = \mathbb{R}$, for any $(x, y) \in \mathcal{X} \times \mathcal{Y}$. The hypothesis sets defined above, $\mathcal{H}_{\text{all}}$, $\mathcal{H}_{\text{lin}}$ and $\mathcal{H}_{\text{NN}}$ with $B = +\infty$ are all complete.

## 4.1 Max losses

In this section, we discuss guarantees for *max losses*, that is loss functions that can be defined by the application of an auxiliary function $\Phi$ to the margin $\rho_h(x, y)$, as in (Crammer and Singer, 2001):

$$\forall (x, y) \in \mathcal{X} \times \mathcal{Y}, \quad \Phi^{\max}(h, x, y) = \max_{y' \neq y} \Phi(h(x, y) - h(x, y')) = \Phi(\rho_h(x, y)). \tag{5}$$

**i) Negative results**. We first give negative results showing that max losses $\Phi^{\max}(h, x, y)$ with convex and non-increasing auxiliary functions $\Phi$ do not admit useful $\mathcal{H}$-consistency bounds for multi-class classification ($c > 2$). The proof is given in Appendix G.

**Theorem 6** (**Negative results for convex** $\Phi$). *Assume that $c > 2$. Suppose that $\Phi$ is convex and non-increasing, and $\mathcal{H}$ satisfies there exist $x \in \mathcal{X}$ and $h \in \mathcal{H}$ such that $|\mathsf{H}(x)| \geq 2$ and $h(x, y)$ are equal for all $y \in \mathcal{Y}$. If for a non-decreasing function $f: \mathbb{R}_+ \to \mathbb{R}_+$, the following $\mathcal{H}$-consistency bound holds for any hypothesis $h \in \mathcal{H}$ and any distribution $\mathcal{D}$:*

$$\mathcal{R}_{\ell_{0-1}}(h) - \mathcal{R}^*_{\ell_{0-1}, \mathcal{H}} \leq f\left(\mathcal{R}_{\Phi^{\max}}(h) - \mathcal{R}^*_{\Phi^{\max}, \mathcal{H}}\right), \tag{6}$$

*then, $f$ is lower bounded by $\frac{1}{2}$.*

The condition on the hypothesis set in Theorem 6 is very general and all symmetric hypothesis sets verify the condition, e.g. $\mathcal{H}_{\text{all}}$, $\mathcal{H}_{\text{lin}}$ and $\mathcal{H}_{\text{NN}}$. It is also worth pointing out that when $c = 2$, that is, in binary classification, Theorem 6 does not hold. Indeed, Awasthi et al. (2022a) present a series of results providing $\mathcal{H}$-consistency bounds for convex $\Phi$ in the binary case. In the proof, we make use of the assumption that $c > 2$ and thus are able to take a probability vector $p(x)$ whose dimension is at least three, which is crucial for the proof.

**ii) Positive results without distributional assumptions**. On the positive side, the max loss with the non-convex auxiliary function $\Phi = \Phi_\rho$ admits $\mathcal{H}$-consistency bounds.

**Theorem 7** ($\mathcal{H}$-**consistency bound of** $\Phi_\rho^{\max}$). *Suppose that $\mathcal{H}$ is symmetric. Then, for any hypothesis $h \in \mathcal{H}$ and any distribution $\mathcal{D}$,*

$$\mathcal{R}_{\ell_{0-1}}(h) - \mathcal{R}^*_{\ell_{0-1}, \mathcal{H}} \leq \frac{\mathcal{R}_{\Phi_\rho^{\max}}(h) - \mathcal{R}^*_{\Phi_\rho^{\max}, \mathcal{H}} + \mathcal{M}_{\Phi_\rho^{\max}, \mathcal{H}}}{\min\left\{1, \frac{\inf_{x \in \mathcal{X}} \sup_{h \in \mathcal{H}} \rho_h(x, \mathsf{h}(x))}{\rho}\right\}} - \mathcal{M}_{\ell_{0-1}, \mathcal{H}}. \tag{7}$$

See Appendix G for the proof. Theorem 7 is very powerful since it only requires $\mathcal{H}$ to be symmetric. We can use it to derive $\mathcal{H}$-consistency bounds for $\Phi_\rho^{\max}$ with common symmetric hypothesis sets

Table 1: $\mathcal{H}$-consistency bounds for $\Phi_\rho^{\max}$ with common symmetric hypothesis sets.

| Hypothesis set | $\mathcal{H}$-consistency bound of $\Phi_\rho^{\max}$ (Corollaries 18, 19 and 20) |
|---|---|
| $\mathcal{H}_{\text{all}}$ | $\mathcal{R}_{\ell_{0-1}}(h) - \mathcal{R}^*_{\ell_{0-1},\mathcal{H}_{\text{all}}} \leq \mathcal{R}_{\Phi_\rho^{\max}}(h) - \mathcal{R}^*_{\Phi_\rho^{\max},\mathcal{H}_{\text{all}}}$ |
| $\mathcal{H}_{\text{lin}}$ | $\mathcal{R}_{\ell_{0-1}}(h) - \mathcal{R}^*_{\ell_{0-1},\mathcal{H}_{\text{lin}}} \leq \dfrac{\mathcal{R}_{\Phi_\rho^{\max}}(h) - \mathcal{R}^*_{\Phi_\rho^{\max},\mathcal{H}_{\text{lin}}} + \mathcal{M}_{\Phi_\rho^{\max},\mathcal{H}_{\text{lin}}}}{\min\left\{1,\frac{2B}{\rho}\right\}} - \mathcal{M}_{\ell_{0-1},\mathcal{H}_{\text{lin}}}$ |
| $\mathcal{H}_{\text{NN}}$ | $\mathcal{R}_{\ell_{0-1}}(h) - \mathcal{R}^*_{\ell_{0-1},\mathcal{H}_{\text{NN}}} \leq \dfrac{\mathcal{R}_{\Phi_\rho^{\max}}(h) - \mathcal{R}^*_{\Phi_\rho^{\max},\mathcal{H}_{\text{NN}}} + \mathcal{M}_{\Phi_\rho^{\max},\mathcal{H}_{\text{NN}}}}{\min\left\{1,\frac{2\Lambda B}{\rho}\right\}} - \mathcal{M}_{\ell_{0-1},\mathcal{H}_{\text{NN}}}$ |

such as $\mathcal{H}_{\text{all}}$, $\mathcal{H}_{\text{lin}}$ and $\mathcal{H}_{\text{NN}}$, as summarized in Table 1. The proofs with corresponding summarized Corollaries 18, 19 and 20 are included in Appendix H. In the proofs, we characterize the term $\inf_{x \in \mathcal{X}} \sup_{h \in \mathcal{H}} \rho_h(x, h(x))$ for each hypothesis set.

Note that by Theorem 6, there is no useful $\mathcal{H}$-consistency bound for the max loss with $\Phi = \Phi_{\text{hinge}}$, $\Phi_{\text{sq-hinge}}$ or $\Phi_{\text{exp}}$ in these cases. However, under the realizability assumption (Definition 8), we will show that such bounds hold.

**iii) Positive results with realizable distributions**. We consider the $\mathcal{H}$-realizability condition (Long and Servedio, 2013; Kuznetsov et al., 2014; Cortes et al., 2016a,b; Zhang and Agarwal, 2020; Awasthi et al., 2021a) which is defined as follows.

**Definition 8** ($\mathcal{H}$-**realizability**). *A distribution $\mathcal{D}$ over $\mathcal{X} \times \mathcal{Y}$ is $\mathcal{H}$-realizable if it labels points according to a deterministic model in $\mathcal{H}$, i.e., if $\exists h \in \mathcal{H}$ such that $\mathbb{P}_{(x,y) \sim \mathcal{D}}(\rho_h(x,y) > 0) = 1$.*

**Theorem 9** (**Realizable $\mathcal{H}$-consistency bound of $\Phi^{\max}$**). *Suppose that $\mathcal{H}$ is symmetric and complete, and $\Phi$ is non-increasing and satisfies that $\lim_{t \to +\infty} \Phi(t) = 0$. Then, for any hypothesis $h \in \mathcal{H}$ and any $\mathcal{H}$-realizable distribution $\mathcal{D}$, we have*

$$\mathcal{R}_{\ell_{0-1}}(h) - \mathcal{R}^*_{\ell_{0-1},\mathcal{H}} \leq \mathcal{R}_{\Phi^{\max}}(h) - \mathcal{R}^*_{\Phi^{\max},\mathcal{H}} + \mathcal{M}_{\Phi^{\max},\mathcal{H}}. \tag{8}$$

See Appendix G for the proof. Long and Servedio (2013, Theorem 9) show that $\Phi_{\text{hinge}}^{\max}$ is realizable $\mathcal{H}$-consistent for any symmetric hypothesis set $\mathcal{H}$ that is closed under scaling. Since for any $\mathcal{H}$-realizable distribution, the assumption that $\mathcal{H}$ is closed under scaling implies that $\mathcal{H}$ is complete and $\mathcal{M}_{\Phi^{\max},\mathcal{H}} = 0$, Theorem 9 also yields a quantitative relationship in that case that is stronger than the asymptotic consistency property of that previous work.

## 4.2 Sum losses

In this section, we discuss guarantees for *sum losses*, that is loss functions defined via a sum, as in (Weston and Watkins, 1998):

$$\Phi^{\text{sum}}(h, x, y) = \sum_{y' \neq y} \Phi(h(x,y) - h(x,y')). \tag{9}$$

**i) Negative results**. We first give a negative result showing that when using as auxiliary function the hinge-loss, the sum loss cannot benefit from any useful $\mathcal{H}$-consistency guarantee. The proof is deferred to Appendix J.

**Theorem 10** (**Negative results for hinge loss**). *Assume that $c > 2$. Suppose that $\mathcal{H}$ is symmetric and complete. If for a non-decreasing function $f: \mathbb{R}_+ \to \mathbb{R}_+$, the following $\mathcal{H}$-consistency bound holds for any hypothesis $h \in \mathcal{H}$ and any distribution $\mathcal{D}$:*

$$\mathcal{R}_{\ell_{0-1}}(h) - \mathcal{R}^*_{\ell_{0-1},\mathcal{H}} \leq f\Big(\mathcal{R}_{\Phi_{\text{hinge}}^{\text{sum}}}(h) - \mathcal{R}^*_{\Phi_{\text{hinge}}^{\text{sum}},\mathcal{H}}\Big), \tag{10}$$

*then, $f$ is lower bounded by $\frac{1}{6}$.*

**ii) Positive results**. We then complement this negative result with positive results when using the exponential loss, the squared hinge-loss, and the $\rho$-margin loss, as summarized in Table 2. The proofs with corresponding summarized Theorems 22, 23 and 24 are included in Appendix J for completeness. For $\Phi_\rho^{\text{sum}}$, the symmetry and completeness assumption can be relaxed to symmetry and the condition that for any $x \in \mathcal{X}$, there exists a hypothesis $h \in \mathcal{H}$ such that $|h(x,i) - h(x,j)| \geq \rho$ for any $i \neq j \in \mathcal{Y}$, as shown in Theorem 24. In the proof, we introduce an auxiliary Lemma 21 in Appendix I, which would be helpful for lower bounding the conditional regret of $\Phi_\rho^{\text{sum}}$ with that of the multi-class 0/1 loss.

Table 2: $\mathcal{H}$-consistency bounds for sum losses with symmetric and complete hypothesis sets.

| Sum loss | $\mathcal{H}$-consistency bound (Theorems 22, 23 and 24) |
|---|---|
| $\Phi_{\text{sq-hinge}}^{\text{sum}}$ | $\mathcal{R}_{\ell_{0-1}}(h) - \mathcal{R}_{\ell_{0-1},\mathcal{H}}^* \le \left( \mathcal{R}_{\Phi_{\text{sq-hinge}}^{\text{sum}}}(h) - \mathcal{R}_{\Phi_{\text{sq-hinge}}^{\text{sum}},\mathcal{H}}^* + \mathcal{M}_{\Phi_{\text{sq-hinge}}^{\text{sum}},\mathcal{H}} \right)^{\frac{1}{2}} - \mathcal{M}_{\ell_{0-1},\mathcal{H}}$ |
| $\Phi_{\exp}^{\text{sum}}$ | $\mathcal{R}_{\ell_{0-1}}(h) - \mathcal{R}_{\ell_{0-1},\mathcal{H}}^* \le \sqrt{2}\left( \mathcal{R}_{\Phi_{\exp}^{\text{sum}}}(h) - \mathcal{R}_{\Phi_{\exp}^{\text{sum}},\mathcal{H}}^* + \mathcal{M}_{\Phi_{\exp}^{\text{sum}},\mathcal{H}} \right)^{\frac{1}{2}} - \mathcal{M}_{\ell_{0-1},\mathcal{H}}$ |
| $\Phi_{\rho}^{\text{sum}}$ | $\mathcal{R}_{\ell_{0-1}}(h) - \mathcal{R}_{\ell_{0-1},\mathcal{H}}^* \le \mathcal{R}_{\Phi_{\rho}^{\text{sum}}}(h) - \mathcal{R}_{\Phi_{\rho}^{\text{sum}},\mathcal{H}}^* + \mathcal{M}_{\Phi_{\rho}^{\text{sum}},\mathcal{H}} - \mathcal{M}_{\ell_{0-1},\mathcal{H}}$ |

Table 3: $\mathcal{H}$-consistency bounds for constrained losses with symmetric and complete hypothesis sets.

| Constrained loss | $\mathcal{H}$-consistency bound (Theorems 25, 26, 27 and 28) |
|---|---|
| $\Phi_{\text{hinge}}^{\text{cstnd}}$ | $\mathcal{R}_{\ell_{0-1}}(h) - \mathcal{R}_{\ell_{0-1},\mathcal{H}}^* \le \mathcal{R}_{\Phi_{\text{hinge}}^{\text{cstnd}}}(h) - \mathcal{R}_{\Phi_{\text{hinge}}^{\text{cstnd}},\mathcal{H}}^* + \mathcal{M}_{\Phi_{\text{hinge}}^{\text{cstnd}},\mathcal{H}} - \mathcal{M}_{\ell_{0-1},\mathcal{H}}$ |
| $\Phi_{\text{sq-hinge}}^{\text{cstnd}}$ | $\mathcal{R}_{\ell_{0-1}}(h) - \mathcal{R}_{\ell_{0-1},\mathcal{H}}^* \le \left( \mathcal{R}_{\Phi_{\text{sq-hinge}}^{\text{cstnd}}}(h) - \mathcal{R}_{\Phi_{\text{sq-hinge}}^{\text{cstnd}},\mathcal{H}}^* + \mathcal{M}_{\Phi_{\text{sq-hinge}}^{\text{cstnd}},\mathcal{H}} \right)^{\frac{1}{2}} - \mathcal{M}_{\ell_{0-1},\mathcal{H}}$ |
| $\Phi_{\exp}^{\text{cstnd}}$ | $\mathcal{R}_{\ell_{0-1}}(h) - \mathcal{R}_{\ell_{0-1},\mathcal{H}}^* \le \sqrt{2}\left( \mathcal{R}_{\Phi_{\exp}^{\text{cstnd}}}(h) - \mathcal{R}_{\Phi_{\exp}^{\text{cstnd}},\mathcal{H}}^* + \mathcal{M}_{\Phi_{\exp}^{\text{cstnd}},\mathcal{H}} \right)^{\frac{1}{2}} - \mathcal{M}_{\ell_{0-1},\mathcal{H}}$ |
| $\Phi_{\rho}^{\text{cstnd}}$ | $\mathcal{R}_{\ell_{0-1}}(h) - \mathcal{R}_{\ell_{0-1},\mathcal{H}}^* \le \mathcal{R}_{\Phi_{\rho}^{\text{cstnd}}}(h) - \mathcal{R}_{\Phi_{\rho}^{\text{cstnd}},\mathcal{H}}^* + \mathcal{M}_{\Phi_{\rho}^{\text{cstnd}},\mathcal{H}} - \mathcal{M}_{\ell_{0-1},\mathcal{H}}$ |

### 4.3 Constrained losses

In this section, we discuss guarantees for *constrained loss*, that is loss functions defined via a constraint, as in (Lee et al., 2004):

$$\Phi^{\text{cstnd}}(h, x, y) = \sum_{y' \ne y} \Phi(-h(x, y')) \tag{11}$$

with the constraint that $\sum_{y \in \mathcal{Y}} h(x, y) = 0$. We present a series of positive results by proving multi-class $\mathcal{H}$-consistency bounds when using as an auxiliary function the hinge-loss, the squared hinge-loss, the exponential loss, and the $\rho$-margin loss, as summarized in Table 3. As with the binary case (Awasthi et al., 2022a), the bound admits a linear dependency for $\Phi_{\text{hinge}}^{\text{cstnd}}$ and $\Phi_{\rho}^{\text{cstnd}}$, in contrast with a square-root dependency for $\Phi_{\text{sq-hinge}}^{\text{cstnd}}$ and $\Phi_{\exp}^{\text{cstnd}}$, as illustrated in Figure 1. The proofs with corresponding summarized Theorems 25, 26, 27 and 28 are included in Appendix K for completeness. For $\Phi_{\rho}^{\text{cstnd}}$, the symmetric and complete assumption can be relaxed to be symmetric and satisfy that for any $x \in \mathcal{X}$, there exists a hypothesis $h \in \mathcal{H}$ such that $h(x, y) \le -\rho$ for any $y \ne y_{\max}$, as shown in Theorem 28.

The main idea of the proofs in this section is to leverage the constraint condition of Lee et al. (2004) that the scores sum to zero, and appropriately choose a hypothesis $\overline{h}$ that differs from $h$ only by its scores for $h(x)$ and $y_{\max}$. We can then upper bound the minimal conditional risk by the conditional risk of $\overline{h}$, without having to derive the closed form expression of the minimal conditional risk.

As shown by Steinwart (2007, Theorem 3.2), for the family of all measurable functions, the minimizability gaps vanish: $\mathcal{M}_{\ell_{0-1},\mathcal{H}_{\text{all}}} = \mathcal{M}_{\Phi^{\text{sum}},\mathcal{H}_{\text{all}}} = \mathcal{M}_{\Phi^{\text{cstnd}},\mathcal{H}_{\text{all}}} = 0$, for $\Phi = \Phi_{\text{hinge}}$, $\Phi_{\text{sq-hinge}}$, $\Phi_{\exp}$ and $\Phi_{\rho}$. Therefore, when $\mathcal{H} = \mathcal{H}_{\text{all}}$, our quantitative bounds in Table 2 and Table 3 imply the asymptotic consistency results of those multi-class losses in (Tewari and Bartlett, 2007), which shows that our results are stronger and more significant. We also provide bounds for multi-class losses using a non-convex auxiliary function, which are not studied in the previous work.

## 5 Adversarial $\mathcal{H}$-consistency bounds

In this section, we analyze multi-class $\mathcal{H}$-consistency bounds in the adversarial scenario ($\ell_2 = \ell_\gamma$).

For any $x \in \mathcal{X}$, we denote by $\mathcal{H}_\gamma(x)$ the set of hypotheses $h$ with a positive margin on the ball of radius $\gamma$ around $x$, $\mathcal{H}_\gamma(x) = \left\{ h \in \mathcal{H} : \inf_{x' : \|x - x'\|_p \le \gamma} \rho_h(x', h(x)) > 0 \right\}$, and by $\mathsf{H}_\gamma(x)$ the set of labels generated by these hypotheses, $\mathsf{H}_\gamma(x) = \{h(x) : h \in \mathcal{H}_\gamma(x)\}$. When $\mathcal{H}$ is symmetric, we have $\mathsf{H}_\gamma(x) = \mathcal{Y}$ iff $\mathcal{H}_\gamma(x) \ne \varnothing$. The following lemma characterizes the conditional $\epsilon$-regret for

adversarial 0/1 loss, which will be helpful for applying Theorem 1 and Theorem 2 to the adversarial scenario.

**Lemma 11.** *For any $x \in \mathfrak{X}$, the minimal conditional $\ell_\gamma$-risk and the conditional $\epsilon$-regret for $\ell_\gamma$ can be expressed as follows:*

$$\mathcal{C}^*_{\ell_\gamma, \mathcal{H}}(x) = 1 - \max_{y \in \mathsf{H}_\gamma(x)} p(x, y) \mathbb{1}_{\mathcal{H}_\gamma(x) \neq \varnothing}$$

$$\left[\Delta \mathcal{C}_{\ell_\gamma, \mathcal{H}}(h, x)\right]_\epsilon = \begin{cases} \left[\max_{y \in \mathsf{H}_\gamma(x)} p(x, y) - p(x, \mathsf{h}(x)) \mathbb{1}_{h \in \mathcal{H}_\gamma(x)}\right]_\epsilon & \text{if } \mathcal{H}_\gamma(x) \neq \varnothing \\ 0 & \text{otherwise.} \end{cases}$$

The proof of Lemma 11 is presented in Appendix F. By Lemma 11, Theorems 1 and 2 can be instantiated as Theorems 12 and 13 in the adversarial scenario as follows, where $\mathcal{H}$-consistency bounds are provided between the adversarial multi-class $0/1$ loss and a surrogate loss $\ell$.

**Theorem 12** (**Adversarial distribution-dependent $\Psi$-bound**). *Assume that there exists a convex function $\Psi : \mathbb{R}_+ \to \mathbb{R}$ with $\Psi(0) = 0$ and $\epsilon \geq 0$ such that the following holds for all $h \in \mathcal{H}$, $x \in \{x \in \mathfrak{X} : \mathcal{H}_\gamma(x) \neq \varnothing\}$ and $\mathcal{D} \in \mathcal{P}$:*

$$\Psi\left(\left[\max_{y \in \mathsf{H}_\gamma(x)} p(x, y) - p(x, \mathsf{h}(x)) \mathbb{1}_{h \in \mathcal{H}_\gamma(x)}\right]_\epsilon\right) \leq \Delta \mathcal{C}_{\ell, \mathcal{H}}(h, x). \tag{12}$$

*Then, for any hypothesis $h \in \mathcal{H}$ and any distribution $\mathcal{D} \in \mathcal{P}$, we have*

$$\Psi\left(\mathcal{R}_{\ell_\gamma}(h) - \mathcal{R}^*_{\ell_\gamma, \mathcal{H}} + \mathcal{M}_{\ell_\gamma, \mathcal{H}}\right) \leq \mathcal{R}_\ell(h) - \mathcal{R}^*_{\ell, \mathcal{H}} + \mathcal{M}_{\ell, \mathcal{H}} + \max\{0, \Psi(\epsilon)\}. \tag{13}$$

**Theorem 13** (**Adversarial distribution-dependent $\Gamma$-bound**). *Assume that there exists a non-negative concave function $\Gamma : \mathbb{R}_+ \to \mathbb{R}$ and $\epsilon \geq 0$ such that the following holds for all $h \in \mathcal{H}$, $x \in \{x \in \mathfrak{X} : \mathcal{H}_\gamma(x) \neq \varnothing\}$ and $\mathcal{D} \in \mathcal{P}$:*

$$\left[\max_{y \in \mathsf{H}_\gamma(x)} p(x, y) - p(x, \mathsf{h}(x)) \mathbb{1}_{h \in \mathcal{H}_\gamma(x)}\right]_\epsilon \leq \Gamma(\Delta \mathcal{C}_{\ell, \mathcal{H}}(h, x)). \tag{14}$$

*Then, for any hypothesis $h \in \mathcal{H}$ and any distribution $\mathcal{D} \in \mathcal{P}$, we have*

$$\mathcal{R}_{\ell_\gamma}(h) - \mathcal{R}^*_{\ell_\gamma, \mathcal{H}} \leq \Gamma\left(\mathcal{R}_\ell(h) - \mathcal{R}^*_{\ell, \mathcal{H}} + \mathcal{M}_{\ell, \mathcal{H}}\right) - \mathcal{M}_{\ell_\gamma, \mathcal{H}} + \epsilon. \tag{15}$$

Next, we will apply Theorem 12 and Theorem 13 to study various hypothesis sets and adversarial surrogate loss functions in Sections 5.1 for negative results and Section 5.2, 5.3, and 5.4 for positive results. A careful analysis is presented in each case (see Appendix L, M, N and O).

## 5.1 Negative results for adversarial robustness

The following result rules out the $\mathcal{H}$-consistency guarantee of multi-class losses with a convex auxiliary function, which are commonly used in practice. The proof is given in Appendix L.

**Theorem 14** (**Negative results for convex functions**). *Fix $c = 2$. Suppose that $\Phi$ is convex and non-increasing, and $\mathcal{H}$ contains $0$ and satisfies the condition that there exists $x \in \mathfrak{X}$ such that $\mathcal{H}_\gamma(x) \neq \varnothing$. If for a non-decreasing function $f : \mathbb{R}_+ \to \mathbb{R}_+$, the following $\mathcal{H}$-consistency bound holds for any hypothesis $h \in \mathcal{H}$ and any distribution $\mathcal{D}$:*

$$\mathcal{R}_{\ell_\gamma}(h) - \mathcal{R}^*_{\ell_\gamma, \mathcal{H}} \leq f\left(\mathcal{R}_{\widetilde{\ell}}(h) - \mathcal{R}^*_{\widetilde{\ell}, \mathcal{H}}\right), \tag{16}$$

*then, $f$ is lower bounded by $\frac{1}{2}$, for $\widetilde{\ell} = \widetilde{\Phi}^{\max}$, $\widetilde{\Phi}^{\mathrm{sum}}$ and $\widetilde{\Phi}^{\mathrm{cstnd}}$.*

Instead, we show in Sections 5.2, 5.3, and 5.4 that the max, sum and constrained losses using as auxiliary function the non-convex $\rho$-margin loss admit favorable $\mathcal{H}$-consistency bounds in the multi-class setting, thereby significantly generalizing the binary counterpart in (Awasthi et al., 2022a).

## 5.2 Adversarial max losses

We first consider the adversarial max loss $\widetilde{\Phi}^{\max}$ defined as the supremum based counterpart of (5):

$$\widetilde{\Phi}^{\max}(h, x, y) = \sup_{x' : \|x - x'\|_p \leq \gamma} \Phi(\rho_h(x', y)). \tag{17}$$

For the adversarial max loss with $\Phi = \Phi_\rho$, we can obtain $\mathcal{H}$-consistency bounds as follows.

**Theorem 15** ($\mathcal{H}$-**consistency bound of** $\widetilde{\Phi}_\rho^{\max}$). *Suppose that $\mathcal{H}$ is symmetric. Then, for any hypothesis $h \in \mathcal{H}$ and any distribution $\mathcal{D}$, we have*

$$\mathcal{R}_{\ell_\gamma}(h) - \mathcal{R}^*_{\ell_\gamma, \mathcal{H}} \le \frac{\mathcal{R}_{\widetilde{\Phi}_\rho^{\max}}(h) - \mathcal{R}^*_{\widetilde{\Phi}_\rho^{\max}, \mathcal{H}} + \mathcal{M}_{\widetilde{\Phi}_\rho^{\max}, \mathcal{H}}}{\min\left\{1, \frac{\inf_{x \in \{x \in \mathcal{X}: \mathcal{H}_\gamma(x) \ne \varnothing\}} \sup_{h \in \mathcal{H}_\gamma(x)} \inf_{x': \|x - x'\|_p \le \gamma} \rho_h(x', h(x))}{\rho}\right\}} - \mathcal{M}_{\ell_\gamma, \mathcal{H}}. \quad (18)$$

### 5.3 Adversarial sum losses

Next, we consider the adversarial sum loss $\widetilde{\Phi}^{\mathrm{sum}}$ defined as the supremum based counterpart of (9):

$$\widetilde{\Phi}^{\mathrm{sum}}(h, x, y) = \sup_{x': \|x - x'\|_p \le \gamma} \sum_{y' \ne y} \Phi(h(x', y) - h(x', y')). \quad (19)$$

Using the auxiliary Lemma 21 in Appendix I, we can obtain the $\mathcal{H}$-consistency bound of $\widetilde{\Phi}_\rho^{\mathrm{sum}}$.

**Theorem 16** ($\mathcal{H}$-**consistency bound of** $\widetilde{\Phi}_\rho^{\mathrm{sum}}$). *Assume that $\mathcal{H}$ is symmetric and that for any $x \in \mathcal{X}$, there exists a hypothesis $h \in \mathcal{H}$ inducing the same ordering of the labels for any $x' \in \left\{x': \|x - x'\|_p \le \gamma\right\}$ and such that $\inf_{x': \|x - x'\|_p \le \gamma} |h(x', i) - h(x', j)| \ge \rho$ for any $i \ne j \in \mathcal{Y}$. Then, for any hypothesis $h \in \mathcal{H}$ and any distribution $\mathcal{D}$, the following inequality holds:*

$$\mathcal{R}_{\ell_\gamma}(h) - \mathcal{R}^*_{\ell_\gamma, \mathcal{H}} \le \mathcal{R}_{\widetilde{\Phi}_\rho^{\mathrm{sum}}}(h) - \mathcal{R}^*_{\widetilde{\Phi}_\rho^{\mathrm{sum}}, \mathcal{H}} + \mathcal{M}_{\widetilde{\Phi}_\rho^{\mathrm{sum}}, \mathcal{H}} - \mathcal{M}_{\ell_\gamma, \mathcal{H}}. \quad (20)$$

### 5.4 Adversarial constrained loss

Similarly, we define the adversarial constrained loss $\widetilde{\Phi}^{\mathrm{cstnd}}$ as supremum based counterpart of (11):

$$\widetilde{\Phi}^{\mathrm{cstnd}}(h, x, y) = \sup_{x': \|x - x'\|_p \le \gamma} \sum_{y' \ne y} \Phi(-h(x', y')) \quad (21)$$

with the constraint that $\sum_{y \in \mathcal{Y}} h(x, y) = 0$. For the adversarial constrained loss with $\Phi = \Phi_\rho$, we can obtain the $\mathcal{H}$-consistency bound of $\widetilde{\Phi}_\rho^{\mathrm{cstnd}}$ as follows.

**Theorem 17** ($\mathcal{H}$-**consistency bound of** $\widetilde{\Phi}_\rho^{\mathrm{cstnd}}$). *Suppose that $\mathcal{H}$ is symmetric and satisfies that for any $x \in \mathcal{X}$, there exists a hypothesis $h \in \mathcal{H}$ with the constraint $\sum_{y \in \mathcal{Y}} h(x, y) = 0$ such that $\sup_{x': \|x - x'\|_p \le \gamma} h(x', y) \le -\rho$ for any $y \ne y_{\max}$. Then, for any hypothesis $h \in \mathcal{H}$ and any distribution,*

$$\mathcal{R}_{\ell_\gamma}(h) - \mathcal{R}^*_{\ell_\gamma, \mathcal{H}} \le \mathcal{R}_{\widetilde{\Phi}_\rho^{\mathrm{cstnd}}}(h) - \mathcal{R}^*_{\widetilde{\Phi}_\rho^{\mathrm{cstnd}}, \mathcal{H}} + \mathcal{M}_{\widetilde{\Phi}_\rho^{\mathrm{cstnd}}, \mathcal{H}} - \mathcal{M}_{\ell_\gamma, \mathcal{H}}. \quad (22)$$

The proofs of Theorems 15, 16 and 17 are included in Appendix M, N and O respectively. These results are significant since they apply to general hypothesis sets. In particular, symmetric hypothesis sets $\mathcal{H}_{\mathrm{all}}$, $\mathcal{H}_{\mathrm{lin}}$ and $\mathcal{H}_{\mathrm{NN}}$ with $B = +\infty$ all verify the conditions of those theorems. When $B < +\infty$, the conditions in Theorems 16 and 17 can still be verified with a suitable choice of $\rho$, where we can consider the hypotheses such that $w_y = 0$ in $\mathcal{H}_{\mathrm{lin}}$ and $\mathcal{H}_{\mathrm{NN}}$, while Theorem 15 holds for any $\rho > 0$.

## 6 Conclusion

We presented a comprehensive study of $\mathcal{H}$-consistency bounds for multi-class classification, including the analysis of the three most commonly used families of multi-class surrogate losses (max losses, sum losses and constrained losses) and including the study of surrogate losses for the adversarial robustness. Our theoretical analysis helps determine which surrogate losses admit a favorable guarantee for a given hypothesis set $\mathcal{H}$. Our bounds can help guide the design of multi-class classification algorithms for both the adversarial and non-adversarial settings. They also help compare different surrogate losses for the same setting and the same hypothesis set. Of course, in addition to the functional form of the $\mathcal{H}$-consistency bound, the approximation property of a surrogate loss function combined with the hypothesis set plays an important role.

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
