## Contents of Appendix

# A  Related work

The notions of Bayes-consistency (also known as consistency) and calibration have been well studied not only with respect to the binary zero-one loss (Zhang, 2004a; Bartlett et al., 2006; Steinwart, 2007; Mohri et al., 2018), but also with respect to the multi-class zero-one loss (Zhang, 2004b; Tewari and Bartlett, 2007), the general multi-class losses (Ramaswamy and Agarwal, 2012; Narasimhan et al., 2015; Ramaswamy and Agarwal, 2016), the multi-class SVMs (Chen and Sun, 2006; Chen and Xiang, 2006; Liu, 2007; Dogan et al., 2016; Wang and Scott, 2020), the multi-label losses (Gao and Zhou, 2011; Dembczynski et al., 2012; Zhang et al., 2020), the losses with a reject option (Ramaswamy et al., 2015), the ranking losses (Ravikumar et al., 2011; Ramaswamy et al., 2013; Gao and Zhou, 2015; Uematsu and Lee, 2017), the cost sensitive losses (Pires et al., 2013; Pires and Szepesvári, 2016), the structured losses (Ciliberto et al., 2016; Osokin et al., 2017; Blondel, 2019), the polyhedral losses (Frongillo and Waggoner, 2021; Finocchiaro et al., 2022), the Top-$k$ classification losses (Thilagar et al., 2022), the proper losses (Agarwal and Agarwal, 2015; Williamson et al., 2016) and the losses of ordinal regression (Pedregosa et al., 2017).

Bayes-consistency only holds for the full family of measurable functions, which of course is distinct from the more restricted hypothesis set used by a learning algorithm. Therefore, a hypothesis set-dependent notion of $\mathcal{H}$-consistency has been proposed by Long and Servedio (2013) in the realizable setting, used by Zhang and Agarwal (2020) for linear models, and generalized by Kuznetsov et al. (2014) to the structured prediction case. Long and Servedio (2013) showed that there exists a case where a Bayes-consistent loss is not $\mathcal{H}$-consistent while inconsistent losses can be $\mathcal{H}$-consistent. Zhang and Agarwal (2020) further investigated the phenomenon in (Long and Servedio, 2013) and showed that the situation of losses that are not $\mathcal{H}$-consistent with linear models can be remedied by carefully choosing a larger piecewise linear hypothesis set. Kuznetsov et al. (2014) proved positive results for the $\mathcal{H}$-consistency of several multi-class ensemble algorithms, as an extension of $\mathcal{H}$-consistency results in (Long and Servedio, 2013).

Recently, the notions of $\mathcal{H}$-calibration and $\mathcal{H}$-consistency have been used by Bao et al. (2020); Awasthi et al. (2021a) in the study of adversarial binary classification losses, as defined in (Goodfellow et al., 2014; Madry et al., 2017; Tsipras et al., 2018; Carlini and Wagner, 2017; Awasthi et al., 2023). The calibration and consistency of adversarial losses present new challenges and require more careful analysis. The work of Bao et al. (2020) showed that for the linear hypothesis set, convex margin based losses are not calibrated with respect to the adversarial $0/1$ loss. Instead, they proposed a class of non-convex losses that could be calibrated under some necessary and sufficient conditions. The work of Awasthi et al. (2021a) generalized the results in (Bao et al., 2020) to the nonlinear hypothesis sets. They also pointed out that $\mathcal{H}$-calibration and $\mathcal{H}$-consistency are not equivalent in the adversarial scenario by showing that no continuous surrogates can be $\mathcal{H}$-consistent with linear models. They further provided sufficient conditions guaranteeing $\mathcal{H}$-consistency for $\mathcal{H}$-calibrated surrogates.

Most recently, Awasthi et al. (2022a) presented a series of results providing $\mathcal{H}$-consistency bounds in binary classification, for both the adversarial and non-adversarial settings. These guarantees are significantly stronger than the $\mathcal{H}$-calibration or $\mathcal{H}$-consistency properties studied by Awasthi et al. (2021a,c). They are also more informative than similar excess error bounds derived in the literature, which correspond to the special case where $\mathcal{H}$ is the family of all measurable functions (Zhang, 2004a; Bartlett et al., 2006; Mohri et al., 2018). Our work significantly generalizes the results in (Awasthi et al., 2022b) to the multi-class setting, in both the adversarial and non-adversarial scenarios, where the study of calibration and conditional risk is more complex, the form of the surrogate losses is more diverse, and in general the analysis is more involved and entirely novel proof techniques are required. As a by-product, our work contributes more significant results of consistency for the insufficiently understood setting of adversarial robustness.

## B    Discussion on multi-class $0/1$ loss

The multi-class $0/1$ loss can be defined in multiple ways, e.g. $\mathbb{1}_{\rho_h(x,y)\leq 0}$, $\mathbb{1}_{\rho_h(x,y)<0}$ and $\mathbb{1}_{\mathsf{h}(x)\neq y}$ where $\mathsf{h}(x) = \operatorname{argmax}_{y\in\mathcal{Y}} h(x,y)$ with an arbitrary but fixed deterministic strategy used for breaking ties. The counterparts of these three formulas in binary classification are $\mathbb{1}_{yh(x)\leq 0}$, $\mathbb{1}_{yh(x)<0}$ and $\mathbb{1}_{\operatorname{sgn}(h(x))\neq y}$ where $\operatorname{sgn}(0)$ is defined as $+1$ or $-1$. To be consistent with the literature on Bayes-consistency (Bartlett et al., 2006; Tewari and Bartlett, 2007), in this paper we adopt the last formula $\mathbb{1}_{\mathsf{h}(x)\neq y}$ of multi-class $0/1$ loss. Moreover, to be consistent with the binary case (Awasthi et al., 2022a), we assume that in case of a tie, $\mathsf{h}(x)$ is defined as the label with the highest index under the natural ordering of labels. This assumption corresponds to the binary case where we always predict $+1$ in case of a tie, that is, the case where the binary $0/1$ loss is defined by $\mathbb{1}_{\operatorname{sgn}(h(x))\neq y}$ with $\operatorname{sgn}(0) = +1$, as in (Awasthi et al., 2022a). Nevertheless, other deterministic strategies would lead to similar results.

## C    Discussion on finite sample bounds

Here, we discuss several ways to derive the finite sample bounds on the estimation error for the target $0/1$ loss. One can directly derive estimation error bounds for the $0/1$ loss, typically for Empirical Risk Minimization (ERM), e.g. $\mathcal{R}_{\ell_{0-1}}\big(h_S^{\mathrm{ERM}}\big) - \mathcal{R}^*_{\ell_{0-1},\mathcal{H}}$ with $h_S^{\mathrm{ERM}} = \operatorname{argmin}_{h\in\mathcal{H}} \widehat{\mathcal{R}}_S(h)$ can be upper-bounded using the standard generalization bounds, as shown in (Mohri et al., 2018). But, those bounds would not say anything about the use of a surrogate loss.

An alternative is to use the excess error bound for the target $0/1$ loss and split the excess error of the surrogate loss into an estimation term and an approximation term, i.e. for some function $f: \mathbb{R}_+ \to \mathbb{R}_+$, the following inequality holds:

$$\mathcal{R}_{\ell_{0-1}}(h) - \mathcal{R}^*_{\ell_{0-1},\mathcal{H}_{\mathrm{all}}} \leq f\big(\mathcal{R}_{\ell_{\mathrm{sur}}}(h) - \mathcal{R}^*_{\ell_{\mathrm{sur}},\mathcal{H}} + \mathcal{R}^*_{\ell_{\mathrm{sur}},\mathcal{H}} - \mathcal{R}^*_{\ell_{\mathrm{sur}},\mathcal{H}_{\mathrm{all}}}\big).$$

Then, an estimation error bound for the surrogate loss can be used to upper bound $\mathcal{R}_{\ell_{\mathrm{sur}}}(h) - \mathcal{R}^*_{\ell_{\mathrm{sur}},\mathcal{H}}$, as shown in (Bartlett et al., 2006). But, those bounds would not be an estimation error guarantee for the target loss $\ell_{0-1}$.

Finally, using the $\mathcal{H}$-consistency bound proposed by Awasthi et al. (2022a), that is, for some non-decreasing function $f: \mathbb{R}_+ \to \mathbb{R}_+$,

$$\mathcal{R}_{\ell_{0-1}}(h) - \mathcal{R}^*_{\ell_{0-1},\mathcal{H}} \leq f\big(\mathcal{R}_{\ell_{\mathrm{sur}}}(h) - \mathcal{R}^*_{\ell_{\mathrm{sur}},\mathcal{H}}\big),$$

we can directly derive the estimation error bound for the target $0/1$ loss by upper bounding $\mathcal{R}_{\ell_{\mathrm{sur}}}(h) - \mathcal{R}^*_{\ell_{\mathrm{sur}},\mathcal{H}}$ with the estimation error bound for the surrogate loss. In conclusion, the $\mathcal{H}$-consistency bound is a useful tool to derive non-trivial

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

Since $\{h(x) : h \in \mathcal{H}\} = H(x)$, the minimal conditional $\ell_{0-1}$-risk can be expressed as follows:

$$\mathcal{C}^*_{\ell_{0-1}, \mathcal{H}}(x) = 1 - \max_{y \in H(x)} p(x, y),$$

which proves the first part of the lemma. By the definition,

$$\Delta \mathcal{C}_{\ell_{0-1}, \mathcal{H}}(h, x) = \mathcal{C}_{\ell_{0-1}}(h, x) - \mathcal{C}^*_{\ell_{0-1}, \mathcal{H}}(x) = \max_{y \in H(x)} p(x, y) - p(x, h(x)).$$

This leads to

$$\left[ \Delta \mathcal{C}_{\ell_{0-1}, \mathcal{H}}(h, x) \right]_\epsilon = \left[ \max_{y \in H(x)} p(x, y) - p(x, h(x)) \right]_\epsilon.$$

$\square$

**Lemma 11.** *For any $x \in \mathcal{X}$, the minimal conditional $\ell_\gamma$-risk and the conditional $\epsilon$-regret for $\ell_\gamma$ can be expressed as follows:*

$$\mathcal{C}^*_{\ell_\gamma, \mathcal{H}}(x) = 1 - \max_{y \in H_\gamma(x)} p(x, y) \mathbb{1}_{\mathcal{H}_\gamma(x) \neq \varnothing}$$

$$\left[ \Delta \mathcal{C}_{\ell_\gamma, \mathcal{H}}(h, x) \right]_\epsilon = \begin{cases} \left[ \max_{y \in H_\gamma(x)} p(x, y) - p(x, h(x)) \mathbb{1}_{h \in \mathcal{H}_\gamma(x)} \right]_\epsilon & \text{if } \mathcal{H}_\gamma(x) \neq \varnothing \\ 0 & \text{otherwise.} \end{cases}$$

*Proof.* By the definition, the conditional $\ell_\gamma$-risk can be expressed as follows:

$$\mathcal{C}_{\ell_\gamma}(h, x) = \sum_{y \in \mathcal{Y}} p(x, y) \sup_{x' : \|x - x'\|_p \leq \gamma} \mathbb{1}_{\rho_h(x', y) \leq 0} = \begin{cases} 1 - p(x, h(x)) & h \in \mathcal{H}_\gamma(x) \\ 1 & \text{otherwise.} \end{cases} \tag{24}$$

When $\mathcal{H}_\gamma(x) = \varnothing$, (24) implies that $\mathcal{C}^*_{\ell_\gamma, \mathcal{H}}(x) = 1$. When $\mathcal{H}_\gamma(x) \neq \varnothing$, $H_\gamma(x)$ is also non-empty. By (24), $y \in \mathcal{Y}_\gamma(x)$ if and only if there exists $h \in \mathcal{H}_\gamma$ such that $\mathcal{C}_{\ell_\gamma}(h, x) = 1 - p(x, y)$. Therefore, the minimal conditional $\ell_\gamma$-risk can be expressed as follows:

$$\mathcal{C}^*_{\ell_\gamma, \mathcal{H}}(x) = 1 - \max_{y \in H_\gamma(x)} p(x, y) \mathbb{1}_{\mathcal{H}_\gamma(x) \neq \varnothing},$$

which proves the first part of lemma. When $\mathcal{H}_\gamma(x) = \varnothing$, $\mathcal{C}_{\ell_\gamma}(h, x) \equiv 1$, which implies that $\Delta \mathcal{C}_{\ell_\gamma, \mathcal{H}}(h, x) \equiv 0$. When $\mathcal{H}_\gamma(x) \neq \varnothing$, $H_\gamma(x)$ is also non-empty, for $h \in \mathcal{H}_\gamma(x)$, $\Delta \mathcal{C}_{\ell_\gamma, \mathcal{H}}(h, x) = 1 - p(x, h(x)) - \left(1 - \max_{y \in H_\gamma(x)} p(x, y)\right) = \max_{y \in H_\gamma(x)} p(x, y) - p(x, h(x))$; for $h \notin \mathcal{H}_\gamma(x)$, $\Delta \mathcal{C}_{\ell_\gamma, \mathcal{H}}(h, x) = 1 - \left(1 - \max_{y \in H_\gamma(x)} p(x, y)\right) = \max_{y \in H_\gamma(x)} p(x, y)$. Therefore,

$$\Delta \mathcal{C}_{\ell_\gamma, \mathcal{H}}(h, x) = \begin{cases} \max_{y \in H_\gamma(x)} p(x, y) - p(x, h(x)) \mathbb{1}_{h \in \mathcal{H}_\gamma(x)} & \mathcal{H}_\gamma(x) \neq \varnothing \\ 0 & \text{otherwise.} \end{cases}$$

This leads to

$$\left[ \Delta \mathcal{C}_{\ell_\gamma, \mathcal{H}}(h, x) \right]_\epsilon = \begin{cases} \left[ \max_{y \in H_\gamma(x)} p(x, y) - p(x, h(x)) \mathbb{1}_{h \in \mathcal{H}_\gamma(x)} \right]_\epsilon & \mathcal{H}_\gamma(x) \neq \varnothing \\ 0 & \text{otherwise.} \end{cases}$$

$\square$

# G  Proof of negative results and $\mathcal{H}$-consistency bounds for max losses $\Phi^{\max}$

**Theorem 6** (**Negative results for convex $\Phi$**)**.** *Assume that $c > 2$. Suppose that $\Phi$ is convex and non-increasing, and $\mathcal{H}$ satisfies there exist $x \in \mathcal{X}$ and $h \in \mathcal{H}$ such that $|H(x)| \geq 2$ and $h(x, y)$ are equal for all $y \in \mathcal{Y}$. If for a non-decreasing function $f : \mathbb{R}_+ \to \mathbb{R}_+$, the following $\mathcal{H}$-consistency bound holds for any hypothesis $h \in \mathcal{H}$ and any distribution $\mathcal{D}$:*

$$\mathcal{R}_{\ell_{0-1}}(h) - \mathcal{R}^*_{\ell_{0-1}, \mathcal{H}} \leq f\left( \mathcal{R}_{\Phi^{\max}}(h) - \mathcal{R}^*_{\Phi^{\max}, \mathcal{H}} \right), \tag{6}$$

*then, $f$ is lower bounded by $\frac{1}{2}$.*

*Proof.* Consider the distribution that supports on a singleton domain $\{x\}$ with $x$ satisfying that $|\mathsf{H}(x)| \geq 2$. Take $y_1 \in \mathsf{H}(x)$ such that $y_1 \neq c$ and $y_2 \in \mathcal{Y}$ such that $y_2 \neq y_1$, $y_2 \neq c$. We define $p(x)$ as $p(x, y_1) = p(x, y_2) = \frac{1}{2}$ and $p(x, y) = 0$ for other $y \in \mathcal{Y}$. Let $h_0 \in \mathcal{H}$ such that $h_0(x, 1) = h_0(x, 2) = \ldots = h_0(x, c)$. By Lemma 3 and the fact that $y_1 \in \mathsf{H}(x)$, the minimal conditional $\ell_{0-1}$-risk is

$$\mathcal{R}^*_{\ell_{0-1}, \mathcal{H}} = \mathcal{C}^*_{\ell_{0-1}, \mathcal{H}}(x) = 1 - \max_{y \in \mathsf{H}(x)} p(x, y) = 1 - p(x, y_1) = \frac{1}{2}.$$

For $h = h_0$, we have

$$\mathcal{R}_{\ell_{0-1}}(h_0) = \mathcal{C}_{\ell_{0-1}}(h_0, x) = \sum_{y \in \mathcal{Y}} p(x, y) \mathbb{1}_{h_0(x) \neq y} = 1 - p(x, h_0(x)) = 1 - p(x, c) = 1.$$

For the max loss, the conditional $\Phi^{\max}$-risk can be expressed as follows:

$$\mathcal{C}_{\Phi^{\max}}(h, x) = \sum_{y \in \mathcal{Y}} p(x, y) \Phi(\rho_h(x, y)) = \frac{1}{2} \Phi(\rho_h(x, y_1)) + \frac{1}{2} \Phi(\rho_h(x, y_2)).$$

If $\Phi$ is convex and non-increasing, we obtain for any $h \in \mathcal{H}$,

$$
\begin{aligned}
\mathcal{R}_{\Phi^{\max}}(h) = \mathcal{C}_{\Phi^{\max}}(h, x) &= \frac{1}{2} \Phi(\rho_h(x, y_1)) + \frac{1}{2} \Phi(\rho_h(x, y_2)) \\
&\geq \Phi\left(\frac{1}{2} \rho_h(x, y_1) + \frac{1}{2} \rho_h(x, y_2)\right) && (\Phi \text{ is convex}) \\
&= \Phi\left(\frac{1}{2}\left(h(x, y_1) + h(x, y_2) - \max_{y \neq y_1} h(x, y) - \max_{y \neq y_2} h(x, y)\right)\right) \\
&\geq \Phi(0), && (\Phi \text{ is non-increasing})
\end{aligned}
$$

where both equality can be achieved by $h_0$. Therefore,

$$\mathcal{R}^*_{\Phi^{\max}, \mathcal{H}} = \mathcal{C}^*_{\Phi^{\max}, \mathcal{H}}(x) = \mathcal{R}_{\Phi^{\max}}(h_0) = \Phi(0).$$

If (6) holds for some non-decreasing function $f$, then, we obtain for any $h \in \mathcal{H}$,

$$\mathcal{R}_{\ell_{0-1}}(h) - \frac{1}{2} \leq f(\mathcal{R}_{\Phi^{\max}}(h) - \Phi(0)).$$

Let $h = h_0$, then $f(0) \geq 1/2$. Since $f$ is non-decreasing, for any $t \geq 0$, $f(t) \geq 1/2$. $\square$

**Theorem 7** ($\mathcal{H}$**-consistency bound of** $\Phi_\rho^{\max}$). *Suppose that $\mathcal{H}$ is symmetric. Then, for any hypothesis $h \in \mathcal{H}$ and any distribution $\mathcal{D}$,*

$$\mathcal{R}_{\ell_{0-1}}(h) - \mathcal{R}^*_{\ell_{0-1}, \mathcal{H}} \leq \frac{\mathcal{R}_{\Phi_\rho^{\max}}(h) - \mathcal{R}^*_{\Phi_\rho^{\max}, \mathcal{H}} + \mathcal{M}_{\Phi_\rho^{\max}, \mathcal{H}}}{\min\left\{1, \frac{\inf_{x \in \mathcal{X}} \sup_{h \in \mathcal{H}} \rho_h(x, h(x))}{\rho}\right\}} - \mathcal{M}_{\ell_{0-1}, \mathcal{H}}. \tag{7}$$

*Proof.* By the definition, the conditional $\Phi_\rho^{\max}$-risk can be expressed as follows:

$$
\begin{aligned}
\mathcal{C}_{\Phi_\rho^{\max}}(h, x) &= \sum_{y \in \mathcal{Y}} p(x, y) \Phi_\rho(\rho_h(x, y)) \\
&= 1 - p(x, h(x)) + \max\left\{0, 1 - \frac{\rho_h(x, h(x))}{\rho}\right\} p(x, h(x)) \\
&= 1 - \min\left\{1, \frac{\rho_h(x, h(x))}{\rho}\right\} p(x, h(x))
\end{aligned}
\tag{25}
$$

Since $\mathcal{H}$ is symmetric, for any $x \in \mathcal{X}$ and $y \in \mathcal{Y}$,

$$\sup_{h \in \{h \in \mathcal{H}: h(x)=y\}} \rho_h(x, h(x)) = \sup_{h \in \mathcal{H}} \rho_h(x, h(x))$$

Therefore, the minimal conditional $\Phi_\rho^{\max}$-risk can be expressed as follows:

$$\mathcal{C}^*_{\Phi_\rho^{\max}, \mathcal{H}}(x) = 1 - \min\left\{1, \frac{\sup_{h \in \mathcal{H}} \rho_h(x, h(x))}{\rho}\right\} \max_{y \in \mathcal{Y}} p(x, y).$$

By the definition and using the fact that $H(x) = \mathcal{Y}$ when $\mathcal{H}$ is symmetric, we obtain

$$\Delta\mathcal{C}_{\Phi_\rho^{\max},\mathcal{H}}(h,x) = \mathcal{C}_{\Phi_\rho^{\max}}(h,x) - \mathcal{C}_{\Phi_\rho^{\max},\mathcal{H}}^*(x)$$

$$= \min\left\{1, \frac{\sup_{h\in\mathcal{H}}\rho_h(x,h(x))}{\rho}\right\}\max_{y\in\mathcal{Y}} p(x,y) - \min\left\{1, \frac{\rho_h(x,h(x))}{\rho}\right\}p(x,h(x))$$

$$\geq \min\left\{1, \frac{\sup_{h\in\mathcal{H}}\rho_h(x,h(x))}{\rho}\right\}\left(\max_{y\in\mathcal{Y}} p(x,y) - p(x,h(x))\right)$$

$$\geq \min\left\{1, \frac{\sup_{h\in\mathcal{H}}\rho_h(x,h(x))}{\rho}\right\}\Delta\mathcal{C}_{\ell_{0-1},\mathcal{H}}(h,x) \qquad (H(x) = \mathcal{Y})$$

$$\geq \min\left\{1, \frac{\sup_{h\in\mathcal{H}}\rho_h(x,h(x))}{\rho}\right\}\left[\Delta\mathcal{C}_{\ell_{0-1},\mathcal{H}}(h,x)\right]_\epsilon \qquad ([x]_\epsilon \leq x)$$

$$\geq \min\left\{1, \frac{\inf_{x\in\mathcal{X}}\sup_{h\in\mathcal{H}}\rho_h(x,h(x))}{\rho}\right\}\left[\Delta\mathcal{C}_{\ell_{0-1},\mathcal{H}}(h,x)\right]_\epsilon$$

for any $\epsilon \geq 0$. Therefore, taking $\mathcal{P}$ be the set of all distributions, $\mathcal{H}$ be the symmetric hypothesis set, $\epsilon = 0$ and

$$\Psi(t) = \min\left\{1, \frac{\inf_{x\in\mathcal{X}}\sup_{h\in\mathcal{H}}\rho_h(x,h(x))}{\rho}\right\}t$$

in Theorem 4, or, equivalently, $\Gamma(t) = \Psi^{-1}(t)$ in Theorem 5, we obtain for any hypothesis $h \in \mathcal{H}$ and any distribution,

$$\mathcal{R}_{\ell_{0-1}}(h) - \mathcal{R}_{\ell_{0-1},\mathcal{H}}^* \leq \frac{\mathcal{R}_{\Phi_\rho^{\max}}(h) - \mathcal{R}_{\Phi_\rho^{\max},\mathcal{H}}^* + \mathcal{M}_{\Phi_\rho^{\max},\mathcal{H}}}{\min\left\{1, \frac{\inf_{x\in\mathcal{X}}\sup_{h\in\mathcal{H}}\rho_h(x,h(x))}{\rho}\right\}} - \mathcal{M}_{\ell_{0-1},\mathcal{H}}.$$

$\square$

**Theorem 9** (**Realizable $\mathcal{H}$-consistency bound of $\Phi^{\max}$**). *Suppose that $\mathcal{H}$ is symmetric and complete, and $\Phi$ is non-increasing and satisfies that $\lim_{t\to+\infty}\Phi(t) = 0$. Then, for any hypothesis $h \in \mathcal{H}$ and any $\mathcal{H}$-realizable distribution $\mathcal{D}$, we have*

$$\mathcal{R}_{\ell_{0-1}}(h) - \mathcal{R}_{\ell_{0-1},\mathcal{H}}^* \leq \mathcal{R}_{\Phi^{\max}}(h) - \mathcal{R}_{\Phi^{\max},\mathcal{H}}^* + \mathcal{M}_{\Phi^{\max},\mathcal{H}}. \qquad (8)$$

*Proof.* Under the $\mathcal{H}$-realizability assumption of distribution, for any $x \in \mathcal{X}$, there exists $y \in \mathcal{Y}$ such that $p(x,y) = 1$. Then, the conditional $\Phi^{\max}$-risk can be expressed as follows:

$$\mathcal{C}_{\Phi^{\max}}(h,x) = \sum_{y\in\mathcal{Y}} p(x,y)\Phi(\rho_h(x,y))$$
$$= \Phi(\rho_h(x,y_{\max})). \qquad (26)$$

Since $\mathcal{H}$ is symmetric and complete, there exists $h \in \mathcal{H}$ such that $h(x) = y_{\max}$ and we have

$$\sup_{h\in\{h\in\mathcal{H}:h(x)=y_{\max}\}}\rho_h(x,h(x)) = \sup_{h\in\mathcal{H}}\rho_h(x,h(x))$$

$$= \sup_{h\in\mathcal{H}}\left(\max_{y\in\mathcal{Y}} h(x,y) - \max_{y\neq h(x)} h(x,y)\right)$$

$$= +\infty.$$

Thus, using the fact that $\lim_{t\to+\infty}\Phi(t) = 0$, the minimal conditional $\Phi^{\max}$-risk can be expressed as follows:

$$\mathcal{C}_{\Phi^{\max},\mathcal{H}}^*(x) = \inf_{h\in\mathcal{H}}\mathcal{C}_{\Phi^{\max}}(h,x)$$

$$= \inf_{h\in\mathcal{H}}\Phi(\rho_h(x,h(x)))$$

$$= \Phi\left(\sup_{h\in\mathcal{H}}\rho_h(x,h(x))\right) \qquad (\Phi \text{ is non-increasing})$$

$$= 0 \qquad (\lim_{t\to+\infty}\Phi(t) = 0)$$

By the definition and using the fact that $\mathsf{H}(x) = \mathcal{Y}$ when $\mathcal{H}$ is symmetric, we obtain

$$
\begin{aligned}
\Delta \mathcal{C}_{\Phi^{\max}, \mathcal{H}}(h, x) &= \mathcal{C}_{\Phi^{\max}}(h, x) - \mathcal{C}^*_{\Phi^{\max}, \mathcal{H}}(x) \\
&= \Phi\left(\rho_h(x, y_{\max})\right) \\
&\geq \Phi(0) \mathbb{1}_{y_{\max} \neq \mathsf{h}(x)} && (\Phi \text{ is non-increasing}) \\
&\geq \max_{y \in \mathcal{Y}} p(x, y) - p(x, \mathsf{h}(x)) \\
&= \Delta \mathcal{C}_{\ell_{0-1}, \mathcal{H}}(h, x) && (\text{by Lemma 3 and } \mathsf{H}(x) = \mathcal{Y}) \\
&\geq \left[\Delta \mathcal{C}_{\ell_{0-1}, \mathcal{H}}(h, x)\right]_\epsilon && ([t]_\epsilon \leq t)
\end{aligned}
$$

for any $\epsilon \geq 0$. Note that $\mathcal{M}_{\ell_{0-1}, \mathcal{H}} = 0$ under the realizability assumption. Therefore, taking $\mathcal{P}$ be the set of $\mathcal{H}$-realizable distributions, $\mathcal{H}$ be the symmetric and complete hypothesis set, $\epsilon = 0$ and $\Psi(t) = t$ in Theorem 4, or, equivalently, $\Gamma(t) = t$ in Theorem 5, we obtain for any hypothesis $h \in \mathcal{H}$ and any $\mathcal{H}$-realizable distribution,

$$
\mathcal{R}_{\ell_{0-1}}(h) - \mathcal{R}^*_{\ell_{0-1}, \mathcal{H}} \leq \mathcal{R}_{\Phi^{\max}}(h) - \mathcal{R}^*_{\Phi^{\max}, \mathcal{H}} + \mathcal{M}_{\Phi^{\max}, \mathcal{H}}
$$

$\square$

# H   Proof of $\mathcal{H}_{\mathrm{all}}, \mathcal{H}_{\mathrm{lin}}, \mathcal{H}_{\mathrm{NN}}$-consistency bounds for max $\rho$-margin loss $\Phi_\rho^{\max}$

**Corollary 18** ($\mathcal{H}_{\mathrm{all}}$-**consistency bound of** $\Phi_\rho^{\max}$). *For any hypothesis $h \in \mathcal{H}_{\mathrm{all}}$ and any distribution,*

$$
\mathcal{R}_{\ell_{0-1}}(h) - \mathcal{R}^*_{\ell_{0-1}, \mathcal{H}_{\mathrm{all}}} \leq \mathcal{R}_{\Phi_\rho^{\max}}(h) - \mathcal{R}^*_{\Phi_\rho^{\max}, \mathcal{H}_{\mathrm{all}}}. \tag{27}
$$

*Proof.* For $\mathcal{H} = \mathcal{H}_{\mathrm{all}}$, we have for all $x \in \mathcal{X}$, $\sup_{h \in \mathcal{H}_{\mathrm{all}}} \rho_h(x, \mathsf{h}(x)) > \rho$. Furthermore, as shown by Steinwart (2007, Theorem 3.2), the minimizability gaps $\mathcal{M}_{\ell_{0-1}, \mathcal{H}_{\mathrm{all}}} = \mathcal{M}_{\Phi_\rho^{\max}, \mathcal{H}_{\mathrm{all}}} = 0$. Therefore, by Theorem 7, the $\mathcal{H}_{\mathrm{all}}$-consistency bound of $\Phi_\rho^{\max}$ can be expressed as follows:

$$
\mathcal{R}_{\ell_{0-1}}(h) - \mathcal{R}^*_{\ell_{0-1}, \mathcal{H}_{\mathrm{all}}} \leq \mathcal{R}_{\Phi_\rho^{\max}}(h) - \mathcal{R}^*_{\Phi_\rho^{\max}, \mathcal{H}_{\mathrm{all}}}.
$$

$\square$

**Corollary 19** ($\mathcal{H}_{\mathrm{lin}}$-**consistency bound of** $\Phi_\rho^{\max}$). *For any hypothesis $h \in \mathcal{H}_{\mathrm{lin}}$ and any distribution,*

$$
\mathcal{R}_{\ell_{0-1}}(h) - \mathcal{R}^*_{\ell_{0-1}, \mathcal{H}_{\mathrm{lin}}} \leq \frac{\mathcal{R}_{\Phi_\rho^{\max}}(h) - \mathcal{R}^*_{\Phi_\rho^{\max}, \mathcal{H}_{\mathrm{lin}}} + \mathcal{M}_{\Phi_\rho^{\max}, \mathcal{H}_{\mathrm{lin}}}}{\min\left\{1, \frac{2B}{\rho}\right\}} - \mathcal{M}_{\ell_{0-1}, \mathcal{H}_{\mathrm{lin}}}, \tag{28}
$$

*where* $\mathcal{M}_{\ell_{0-1}, \mathcal{H}_{\mathrm{lin}}} = \mathcal{R}^*_{\ell_{0-1}, \mathcal{H}_{\mathrm{lin}}} - \mathbb{E}_X\left[1 - \max_{y \in \mathcal{Y}} p(x, y)\right]$ *and* $\mathcal{M}_{\Phi_\rho^{\max}, \mathcal{H}_{\mathrm{lin}}} = \mathcal{R}^*_{\Phi_\rho^{\max}, \mathcal{H}_{\mathrm{lin}}} - \mathbb{E}_X\left[1 - \min\left\{1, \frac{2\left(W\|x\|_p + B\right)}{\rho}\right\} \max_{y \in \mathcal{Y}} p(x, y)\right]$.

*Proof.* For $\mathcal{H} = \mathcal{H}_{\mathrm{lin}}$, we have for all $x \in \mathcal{X}$,

$$
\begin{aligned}
\sup_{h \in \mathcal{H}_{\mathrm{lin}}} \rho_h(x, \mathsf{h}(x)) &= \sup_{h \in \mathcal{H}_{\mathrm{lin}}} \left(\max_{y \in \mathcal{Y}} h(x, y) - \max_{y \neq \mathsf{h}(x)} h(x, y)\right) \\
&= \max_{\|w\|_q \leq W, |b| \leq B} (w \cdot x + b) - \min_{\|w\|_q \leq W, |b| \leq B} (w \cdot x + b) \\
&= 2\left(W\|x\|_p + B\right)
\end{aligned} \tag{29}
$$

Thus, $\inf_{x \in \mathcal{X}} \sup_{h \in \mathcal{H}_{\mathrm{lin}}} \rho_h(x, \mathsf{h}(x)) = \inf_{x \in \mathcal{X}} 2\left(W\|x\|_p + B\right) = 2B$. Since $\mathcal{H} = \mathcal{H}_{\mathrm{lin}}$ is symmetric, by lemma 3, we have

$$
\mathcal{M}_{\ell_{0-1}, \mathcal{H}_{\mathrm{lin}}} = \mathcal{R}^*_{\ell_{0-1}, \mathcal{H}_{\mathrm{lin}}} - \mathbb{E}_X\left[1 - \max_{y \in \mathcal{Y}} p(x, y)\right]. \tag{30}
$$

By the definition, the conditional $\Phi_\rho^{\max}$-risk can be expressed as follows:

$$\mathcal{C}_{\Phi_\rho^{\max}}(h, x) = \sum_{y \in \mathcal{Y}} p(x, y) \Phi_\rho(\rho_h(x, y))$$

$$= 1 - p(x, \mathsf{h}(x)) + \max\left\{0, 1 - \frac{\rho_h(x, \mathsf{h}(x))}{\rho}\right\} p(x, \mathsf{h}(x))$$

$$= 1 - \min\left\{1, \frac{\rho_h(x, \mathsf{h}(x))}{\rho}\right\} p(x, \mathsf{h}(x))$$

Since $\mathcal{H}_{\mathrm{lin}}$ is symmetric, for any $x \in \mathcal{X}$ and $y \in \mathcal{Y}$,

$$\sup_{h \in \{h \in \mathcal{H}_{\mathrm{lin}} : \mathsf{h}(x) = y\}} \rho_h(x, \mathsf{h}(x)) = \sup_{h \in \mathcal{H}_{\mathrm{lin}}} \rho_h(x, \mathsf{h}(x)).$$

Thus, using (29), the minimal conditional $\Phi_\rho^{\max}$-risk can be expressed as follows:

$$\mathcal{C}^*_{\Phi_\rho^{\max}, \mathcal{H}_{\mathrm{lin}}}(x) = 1 - \min\left\{1, \frac{\sup_{h \in \mathcal{H}_{\mathrm{lin}}} \rho_h(x, \mathsf{h}(x))}{\rho}\right\} \max_{y \in \mathcal{Y}} p(x, y)$$

$$= 1 - \min\left\{1, \frac{2(W\|x\|_p + B)}{\rho}\right\} \max_{y \in \mathcal{Y}} p(x, y) \qquad \text{(by (29))}$$

Therefore, the $(\Phi_\rho^{\max}, \mathcal{H}_{\mathrm{lin}})$-minimizability gap is

$$\mathcal{M}_{\Phi_\rho^{\max}, \mathcal{H}_{\mathrm{lin}}} = \mathcal{R}^*_{\Phi_\rho^{\max}, \mathcal{H}_{\mathrm{lin}}} - \mathbb{E}_X\left[1 - \min\left\{1, \frac{2(W\|x\|_p + B)}{\rho}\right\} \max_{y \in \mathcal{Y}} p(x, y)\right]. \qquad (31)$$

By Theorem 7, the $\mathcal{H}_{\mathrm{lin}}$-consistency bound of $\Phi_\rho^{\max}$ can be expressed as follows:

$$\mathcal{R}_{\ell_{0-1}}(h) - \mathcal{R}^*_{\ell_{0-1}, \mathcal{H}_{\mathrm{lin}}} \leq \frac{\mathcal{R}_{\Phi_\rho^{\max}}(h) - \mathcal{R}^*_{\Phi_\rho^{\max}, \mathcal{H}_{\mathrm{lin}}} + \mathcal{M}_{\Phi_\rho^{\max}, \mathcal{H}_{\mathrm{lin}}}}{\min\left\{1, \frac{2B}{\rho}\right\}} - \mathcal{M}_{\ell_{0-1}, \mathcal{H}_{\mathrm{lin}}}.$$

where $\mathcal{M}_{\ell_{0-1}, \mathcal{H}_{\mathrm{lin}}}$ and $\mathcal{M}_{\Phi_\rho^{\max}, \mathcal{H}_{\mathrm{lin}}}$ are given by (30) and (31) respectively. $\qquad \square$

**Corollary 20** ($\mathcal{H}_{\mathrm{NN}}$-**consistency bound of** $\Phi_\rho^{\max}$). *For any hypothesis $h \in \mathcal{H}_{\mathrm{NN}}$ and any distribution,*

$$\mathcal{R}_{\ell_{0-1}}(h) - \mathcal{R}^*_{\ell_{0-1}, \mathcal{H}_{\mathrm{NN}}} \leq \frac{\mathcal{R}_{\Phi_\rho^{\max}}(h) - \mathcal{R}^*_{\Phi_\rho^{\max}, \mathcal{H}_{\mathrm{NN}}} + \mathcal{M}_{\Phi_\rho^{\max}, \mathcal{H}_{\mathrm{NN}}}}{\min\left\{1, \frac{2\Lambda B}{\rho}\right\}} - \mathcal{M}_{\ell_{0-1}, \mathcal{H}_{\mathrm{NN}}}, \qquad (32)$$

*where $\mathcal{M}_{\ell_{0-1}, \mathcal{H}_{\mathrm{NN}}} = \mathcal{R}^*_{\ell_{0-1}, \mathcal{H}_{\mathrm{NN}}} - \mathbb{E}_X[1 - \max_{y \in \mathcal{Y}} p(x, y)]$ and $\mathcal{M}_{\Phi_\rho^{\max}, \mathcal{H}_{\mathrm{NN}}} = \mathcal{R}^*_{\Phi_\rho^{\max}, \mathcal{H}_{\mathrm{NN}}} - \mathbb{E}_X\left[1 - \min\left\{1, \frac{2\Lambda(W\|x\|_p + B)}{\rho}\right\} \max_{y \in \mathcal{Y}} p(x, y)\right].$*

*Proof.* For $\mathcal{H} = \mathcal{H}_{\mathrm{NN}}$, we have for all $x \in \mathcal{X}$,

$$\sup_{h \in \mathcal{H}_{\mathrm{NN}}} \rho_h(x, \mathsf{h}(x)) = \sup_{h \in \mathcal{H}_{\mathrm{NN}}} \left(\max_{y \in \mathcal{Y}} h(x, y) - \max_{y \neq \mathsf{h}(x)} h(x, y)\right)$$

$$= \max_{\|u\|_1 \leq \Lambda, \|w_j\|_q \leq W, |b_j| \leq B} \left(\sum_{j=1}^n u_j(w_j \cdot x + b_j)_+\right) - \min_{\|u\|_1 \leq \Lambda, \|w_j\|_q \leq W, |b_j| \leq B} \left(\sum_{j=1}^n u_j(w_j \cdot x + b_j)_+\right) \qquad (33)$$

$$= 2\Lambda(W\|x\|_p + B)$$

Thus, $\inf_{x \in \mathcal{X}} \sup_{h \in \mathcal{H}_{\mathrm{NN}}} \rho_h(x, \mathsf{h}(x)) = \inf_{x \in \mathcal{X}} 2\Lambda(W\|x\|_p + B) = 2\Lambda B$. Since $\mathcal{H} = \mathcal{H}_{\mathrm{NN}}$ is symmetric, by lemma 3, we have

$$\mathcal{M}_{\ell_{0-1}, \mathcal{H}_{\mathrm{NN}}} = \mathcal{R}^*_{\ell_{0-1}, \mathcal{H}_{\mathrm{NN}}} - \mathbb{E}_X\left[1 - \max_{y \in \mathcal{Y}} p(x, y)\right]. \qquad (34)$$

By the definition, the conditional $\Phi_\rho^{\max}$-risk can be expressed as follows:

$$\mathcal{C}_{\Phi_\rho^{\max}}(h,x) = \sum_{y \in \mathcal{Y}} p(x,y) \Phi_\rho(\rho_h(x,y))$$

$$= 1 - p(x,\mathsf{h}(x)) + \max\left\{0, 1 - \frac{\rho_h(x,\mathsf{h}(x))}{\rho}\right\} p(x,\mathsf{h}(x))$$

$$= 1 - \min\left\{1, \frac{\rho_h(x,\mathsf{h}(x))}{\rho}\right\} p(x,\mathsf{h}(x))$$

Since $\mathcal{H}_{\mathrm{NN}}$ is symmetric, for any $x \in \mathcal{X}$ and $y \in \mathcal{Y}$,

$$\sup_{h \in \{h \in \mathcal{H}_{\mathrm{NN}}: \mathsf{h}(x)=y\}} \rho_h(x,\mathsf{h}(x)) = \sup_{h \in \mathcal{H}_{\mathrm{NN}}} \rho_h(x,\mathsf{h}(x)).$$

Thus, using (33), the minimal conditional $\Phi_\rho^{\max}$-risk can be expressed as follows:

$$\mathcal{C}_{\Phi_\rho^{\max}, \mathcal{H}_{\mathrm{NN}}}^*(x) = 1 - \min\left\{1, \frac{\sup_{h \in \mathcal{H}_{\mathrm{NN}}} \rho_h(x,\mathsf{h}(x))}{\rho}\right\} \max_{y \in \mathcal{Y}} p(x,y)$$

$$= 1 - \min\left\{1, \frac{2\Lambda\big(W\|x\|_p + B\big)}{\rho}\right\} \max_{y \in \mathcal{Y}} p(x,y) \qquad \text{(by (33))}$$

Therefore, the $\big(\Phi_\rho^{\max}, \mathcal{H}_{\mathrm{NN}}\big)$-minimizability gap is

$$\mathcal{M}_{\Phi_\rho^{\max}, \mathcal{H}_{\mathrm{NN}}} = \mathcal{R}_{\Phi_\rho^{\max}, \mathcal{H}_{\mathrm{NN}}}^* - \mathbb{E}_X\left[1 - \min\left\{1, \frac{2\Lambda\big(W\|x\|_p + B\big)}{\rho}\right\} \max_{y \in \mathcal{Y}} p(x,y)\right]. \qquad (35)$$

By Theorem 7, the $\mathcal{H}_{\mathrm{NN}}$-consistency bound of $\Phi_\rho^{\max}$ can be expressed as follows:

$$\mathcal{R}_{\ell_{0-1}}(h) - \mathcal{R}_{\ell_{0-1}, \mathcal{H}_{\mathrm{NN}}}^* \leq \frac{\mathcal{R}_{\Phi_\rho^{\max}}(h) - \mathcal{R}_{\Phi_\rho^{\max}, \mathcal{H}_{\mathrm{NN}}}^* + \mathcal{M}_{\Phi_\rho^{\max}, \mathcal{H}_{\mathrm{NN}}}}{\min\left\{1, \frac{2\Lambda B}{\rho}\right\}} - \mathcal{M}_{\ell_{0-1}, \mathcal{H}_{\mathrm{NN}}}.$$

where $\mathcal{M}_{\ell_{0-1}, \mathcal{H}_{\mathrm{NN}}}$ and $\mathcal{M}_{\Phi_\rho^{\max}, \mathcal{H}_{\mathrm{NN}}}$ are given by (34) and (35) respectively. $\qquad \square$

# I  Auxiliary Lemma for sum losses

**Lemma 21.** *Fix a vector $\tau = (\tau_1, \ldots, \tau_c)$ in the probability simplex of $\mathbb{R}^c$ and any real values $a_1 \leq a_2 \leq \cdots \leq a_c$ in increasing order. Then, for any permutation $\sigma$ of the set $\{1, \ldots, c\}$,*

$$\begin{bmatrix} a_1 \\ a_2 \\ \vdots \\ a_c \end{bmatrix} \cdot \begin{bmatrix} \tau_{\sigma(1)} \\ \tau_{\sigma(2)} \\ \vdots \\ \tau_{\sigma(c)} \end{bmatrix} \leq \begin{bmatrix} a_1 \\ a_2 \\ \vdots \\ a_c \end{bmatrix} \cdot \begin{bmatrix} \tau_{[1]} \\ \tau_{[2]} \\ \vdots \\ \tau_{[c]} \end{bmatrix},$$

*where we define $\tau_{[1]}, \tau_{[2]}, \ldots, \tau_{[c]}$ by sorting the probabilities $\{\tau_y : y \in \{1, \ldots, c\}\}$ in increasing order.*

*Proof.* For any permutation $\sigma$ of the set $\{1, \ldots, c\}$, we prove by induction. At the first step, if $\sigma(c) = [c]$, then let $\sigma_1 = \sigma$. Otherwise, denote $k_1 \in \{1, \ldots, c-1\}$ such that $\sigma(k_1) = [c]$ and choose $\sigma_1$ to be the permutation that differs from $\sigma$ only by permuting $c$ and $k_1$. Thus,

$$\begin{bmatrix} a_1 \\ a_2 \\ \vdots \\ a_c \end{bmatrix} \cdot \begin{bmatrix} \tau_{\sigma(1)} \\ \tau_{\sigma(2)} \\ \vdots \\ \tau_{\sigma(c)} \end{bmatrix} - \begin{bmatrix} a_1 \\ a_2 \\ \vdots \\ a_c \end{bmatrix} \cdot \begin{bmatrix} \tau_{\sigma_1(1)} \\ \tau_{\sigma_1(2)} \\ \vdots \\ \tau_{\sigma_1(c)} \end{bmatrix} = a_{k_1} \tau_{[c]} + a_c \tau_{\sigma(c)} - \big(a_{k_1} \tau_{\sigma(c)} + a_c \tau_{[c]}\big)$$

$$= \big(a_{k_1} - a_c\big)\big(\tau_{[c]} - \tau_{\sigma(c)}\big) \leq 0.$$

At the second step, if $\sigma_1(c-1) = [c-1]$, then let $\sigma_2 = \sigma_1$. Otherwise, denote $k_2 \in \{1, \ldots, c-2\}$ such that $\sigma_1(k_2) = [c-1]$ and choose $\sigma_2$ to be the permutation that differs from $\sigma_1$ only by permuting $c-1$ and $k_2$. Thus,

$$\begin{bmatrix} a_1 \\ a_2 \\ \vdots \\ a_c \end{bmatrix} \cdot \begin{bmatrix} \tau_{\sigma_1(1)} \\ \tau_{\sigma_1(2)} \\ \vdots \\ \tau_{\sigma_1(c)} \end{bmatrix} - \begin{bmatrix} a_1 \\ a_2 \\ \vdots \\ a_c \end{bmatrix} \cdot \begin{bmatrix} \tau_{\sigma_2(1)} \\ \tau_{\sigma_2(2)} \\ \vdots \\ \tau_{\sigma_2(c)} \end{bmatrix} = (a_{k_2} - a_{c-1})(\tau_{[c-1]} - \tau_{\sigma_1(c-1)}) \leq 0.$$

And so on, at the $n$th step, if $\sigma_{n-1}(c-n+1) = [c-n+1]$, then let $\sigma_n = \sigma_{n-1}$. Otherwise, denote $k_n \in \{1, \ldots, c-n\}$ such that $\sigma_{n-1}(k_n) = [c-n+1]$ and choose $\sigma_n$ to be the permutation that differs from $\sigma_{n-1}$ only by permuting $c-n+1$ and $k_n$. We have

$$\begin{bmatrix} a_1 \\ a_2 \\ \vdots \\ a_c \end{bmatrix} \cdot \begin{bmatrix} \tau_{\sigma_{n-1}(1)} \\ \tau_{\sigma_{n-1}(2)} \\ \vdots \\ \tau_{\sigma_{n-1}(c)} \end{bmatrix} \leq \begin{bmatrix} a_1 \\ a_2 \\ \vdots \\ a_c \end{bmatrix} \cdot \begin{bmatrix} \tau_{\sigma_n(1)} \\ \tau_{\sigma_n(2)} \\ \vdots \\ \tau_{\sigma_n(c)} \end{bmatrix}.$$

Finally, after $c$ steps, we will obtain $\sigma_c$ which satisfies $\sigma_c(y) = [y]$ for any $y \in \{1, \ldots, c\}$. Therefore, we obtain

$$\begin{bmatrix} a_1 \\ a_2 \\ \vdots \\ a_c \end{bmatrix} \cdot \begin{bmatrix} \tau_{\sigma(1)} \\ \tau_{\sigma(2)} \\ \vdots \\ \tau_{\sigma(c)} \end{bmatrix} \leq \begin{bmatrix} a_1 \\ a_2 \\ \vdots \\ a_c \end{bmatrix} \begin{bmatrix} \tau_{\sigma_1(1)} \\ \tau_{\sigma_1(2)} \\ \vdots \\ \tau_{\sigma_1(c)} \end{bmatrix} \leq \ldots \leq \begin{bmatrix} a_1 \\ a_2 \\ \vdots \\ a_c \end{bmatrix} \begin{bmatrix} \tau_{\sigma_n(1)} \\ \tau_{\sigma_n(2)} \\ \vdots \\ \tau_{\sigma_n(c)} \end{bmatrix} \leq \ldots \leq \begin{bmatrix} a_1 \\ a_2 \\ \vdots \\ a_c \end{bmatrix} \cdot \begin{bmatrix} \tau_{[1]} \\ \tau_{[2]} \\ \vdots \\ \tau_{[c]} \end{bmatrix}$$

which proves the lemma. □

## J   Proof of negative and $\mathcal{H}$-consistency bounds for sum losses $\Phi^{\mathrm{sum}}$

By the definition, the conditional $\Phi^{\mathrm{sum}}$-risk can be expressed as follows:

$$\begin{aligned} \mathcal{C}_{\Phi^{\mathrm{sum}}}(h, x) &= \sum_{y \in \mathcal{Y}} p(x, y) \sum_{y' \neq y} \Phi(h(x, y) - h(x, y')) \\ &= \sum_{y \in \mathcal{Y}} p(x, y) \sum_{y' \in \mathcal{Y}} \Phi(h(x, y) - h(x, y')) - \Phi(0) \end{aligned} \tag{36}$$

**Theorem 10 (Negative results for hinge loss).** *Assume that $c > 2$. Suppose that $\mathcal{H}$ is symmetric and complete. If for a non-decreasing function $f : \mathbb{R}_+ \to \mathbb{R}_+$, the following $\mathcal{H}$-consistency bound holds for any hypothesis $h \in \mathcal{H}$ and any distribution $\mathcal{D}$:*

$$\mathcal{R}_{\ell_{0-1}}(h) - \mathcal{R}^*_{\ell_{0-1}, \mathcal{H}} \leq f\Big(\mathcal{R}_{\Phi^{\mathrm{sum}}_{\mathrm{hinge}}}(h) - \mathcal{R}^*_{\Phi^{\mathrm{sum}}_{\mathrm{hinge}}, \mathcal{H}}\Big), \tag{10}$$

*then, $f$ is lower bounded by $\frac{1}{6}$.*

*Proof.* Consider the distribution that supports on a singleton domain $\{x\}$. We define $p(x)$ as $p(x, 1) = \frac{1}{2} - \epsilon$, $p(x, 2) = \frac{1}{3}$, $p(x, 3) = \frac{1}{6} + \epsilon$ and $p(x, y) = 0$ for other $y \in \mathcal{Y}$, where $0 < \epsilon < \frac{1}{6}$. Note $p(x, 1) > p(x, 2) > p(x, 3) > p(x, y) = 0$, $y \notin \{1, 2, 3\}$. Let $h_0 \in \mathcal{H}$ such that $h_0(x, 1) = 1$, $h_0(x, 2) = 1$, $h_0(x, 3) = 0$ and $h_0(x, y) = -1$ for other $y \in \mathcal{Y}$. By the completeness of $\mathcal{H}$, the hypothesis $h$ is in $\mathcal{H}$. By Lemma 3 and the fact that $\mathsf{H}(x) = \mathcal{Y}$ when $\mathcal{H}$ is symmetric, the minimal conditional $\ell_{0-1}$-risk is

$$\mathcal{R}^*_{\ell_{0-1}, \mathcal{H}} = \mathcal{C}^*_{\ell_{0-1}, \mathcal{H}}(x) = 1 - \max_{y \in \mathcal{Y}} p(x, y) = 1 - p(x, 1) = \frac{1}{2} + \epsilon.$$

For $h = h_0$, we have

$$\mathcal{R}_{\ell_{0-1}}(h_0) = \mathcal{C}_{\ell_{0-1}}(h_0, x) = \sum_{y \in \mathcal{Y}} p(x, y) \mathbb{1}_{h_0(x) \neq y} = 1 - p(x, h_0(x)) = 1 - p(x, 2) = \frac{2}{3}.$$

For the sum hinge loss, by (36), the conditional $\Phi_{\text{hinge}}^{\text{sum}}$-risk can be expressed as follows:

$$
\begin{aligned}
\mathcal{C}_{\Phi_{\text{hinge}}^{\text{sum}}}(h,x) &= \sum_{y \in \mathcal{Y}} p(x,y) \sum_{y' \neq y} \max\{0, 1 + h(x,y') - h(x,y)\} \\
&= \sum_{y \in \{1,2,3\}} p(x,y) \sum_{y' \neq y} \max\{0, 1 + h(x,y') - h(x,y)\} \\
&\geq \sum_{y \in \{1,2,3\}} p(x,y) \sum_{y' \neq y, y' \in \{1,2,3\}} \max\{0, 1 + h(x,y') - h(x,y)\} \\
&= \left(\frac{1}{2} - \epsilon\right)[\max\{0, 1 + h(x,2) - h(x,1)\} + \max\{0, 1 + h(x,3) - h(x,1)\}] \\
&\quad + \frac{1}{3}[\max\{0, 1 + h(x,1) - h(x,2)\} + \max\{0, 1 + h(x,3) - h(x,2)\}] \\
&\quad + \left(\frac{1}{6} + \epsilon\right)[\max\{0, 1 + h(x,1) - h(x,3)\} + \max\{0, 1 + h(x,2) - h(x,3)\}] \\
&= g(h).
\end{aligned}
$$

Note $\mathcal{C}_{\Phi_{\text{hinge}}^{\text{sum}}}(h_0, x) = 3\epsilon + \frac{3}{2}$. Since $\frac{1}{2} - \epsilon > \frac{1}{3} > \frac{1}{6} + \epsilon$, by Lemma 21, we have

$$
\inf_{h \in \mathcal{H}} g(h) = \inf_{h \in \mathcal{H}: h(x,1) \geq h(x,2) \geq h(x,3)} g(h).
$$

When $h(x,1) \geq h(x,2) \geq h(x,3)$, $g(h)$ can be written as

$$
\begin{aligned}
g(h) &= \left(\frac{1}{2} - \epsilon\right)[\max\{0, 1 + h(x,2) - h(x,1)\} + \max\{0, 1 + h(x,3) - h(x,1)\}] \\
&\quad + \frac{1}{3}[(1 + h(x,1) - h(x,2)) + \max\{0, 1 + h(x,3) - h(x,2)\}] \\
&\quad + \left(\frac{1}{6} + \epsilon\right)[(1 + h(x,1) - h(x,3)) + (1 + h(x,2) - h(x,3))]
\end{aligned}
$$

If $h(x,1) - h(x,2) > 1$, define the hypothesis $\overline{h} \in \mathcal{H}$ by

$$
\overline{h}(x,y) = \begin{cases} h(x,1) - \frac{h(x,1) - h(x,2) - 1}{2}, & \text{if } y = 1 \\ h(x,y) & \text{otherwise.} \end{cases}
$$

By the completeness of $\mathcal{H}$ and some computation, the new hypothesis $\overline{h}$ is in $\mathcal{H}$ and satisfies that $g(\overline{h}) < g(h)$. Similarly, if $h(x,2) - h(x,3) > 1$, define the hypothesis $\overline{h} \in \mathcal{H}$ by

$$
\overline{h}(x,y) = \begin{cases} h(x,2) - \frac{h(x,2) - h(x,3) - 1}{2}, & \text{if } y = 2 \\ h(x,y) & \text{otherwise.} \end{cases}
$$

By the completeness of $\mathcal{H}$ and some computation, the new hypothesis $\overline{h}$ is in $\mathcal{H}$ and satisfies that $g(\overline{h}) < g(h)$. Therefore,

$$
\inf_{h \in \mathcal{H}} g(h) = \inf_{h \in \mathcal{H}: h(x,1) \geq h(x,2) \geq h(x,3)} g(h) = \inf_{h \in \mathcal{H}: h(x,1) \geq h(x,2) \geq h(x,3),\, h(x,1) - h(x,2) \leq 1,\, h(x_2) - h(x,3) \leq 1} g(h)
$$

When $h(x,1) \geq h(x,2) \geq h(x,3)$, $h(x,1) - h(x,2) \leq 1$ and $h(x_2) - h(x,3) \leq 1$, $g(h)$ can be written as

$$
\begin{aligned}
g(h) &= \left(\frac{1}{2} - \epsilon\right)[(1 + h(x,2) - h(x,1)) + \max\{0, 1 + h(x,3) - h(x,1)\}] \\
&\quad + \frac{1}{3}[(1 + h(x,1) - h(x,2)) + (1 + h(x,3) - h(x,2))] \\
&\quad + \left(\frac{1}{6} + \epsilon\right)[(1 + h(x,1) - h(x,3)) + (1 + h(x,2) - h(x,3))]
\end{aligned}
$$

If $h(x,1) - h(x,3) > 1$, define the hypothesis $\overline{h} \in \mathcal{H}$ by

$$\overline{h}(x,y) = \begin{cases} h(x,1) - \frac{h(x,1)-h(x,3)-1}{2}, & \text{if } y = 1 \\ h(x,y) & \text{otherwise.} \end{cases}$$

By the completeness of $\mathcal{H}$ and some computation using the fact that $0 < \epsilon < \frac{1}{6}$, the new hypothesis $\overline{h}$ is in $\mathcal{H}$ and satisfies that $g(\overline{h}) < g(h)$. Therefore,

$$\inf_{h \in \mathcal{H}} g(h) = \inf_{h \in \mathcal{H}: h(x,1) \geq h(x,2) \geq h(x,3),\, h(x,1)-h(x,2) \leq 1,\, h(x_2)-h(x,3) \leq 1,\, h(x,1)-h(x,3) \leq 1} g(h)$$

$$= \inf_{h \in \mathcal{H}: h(x,1) \geq h(x,2) \geq h(x,3),\, h(x,1)-h(x,2) \leq 1,\, h(x_2)-h(x,3) \leq 1,\, h(x,1)-h(x,3) \leq 1} \left(3\epsilon - \frac{1}{2}\right)(h(x,1) - h(x,3)) + 2$$

$$= 3\epsilon + \frac{3}{2}$$

Thus, we obtain for any $h \in \mathcal{H}$,

$$\mathcal{R}_{\Phi_{\text{hinge}}^{\text{sum}}}(h) = \mathcal{C}_{\Phi_{\text{hinge}}^{\text{sum}}}(h,x) \geq g(h) \geq 3\epsilon + \frac{3}{2} = \mathcal{C}_{\Phi_{\text{hinge}}^{\text{sum}}}(h_0,x)$$

Therefore,

$$\mathcal{R}_{\Phi_{\text{hinge}}^{\text{sum}},\mathcal{H}}^* = \mathcal{C}_{\Phi_{\text{hinge}}^{\text{sum}},\mathcal{H}}^*(x) = \mathcal{R}_{\Phi_{\text{hinge}}^{\text{sum}}}(h_0) = 3\epsilon + \frac{3}{2}.$$

If (10) holds for some non-decreasing function $f$, then, we obtain for any $h \in \mathcal{H}$,

$$\mathcal{R}_{\ell_{0-1}}(h) - \frac{1}{2} - \epsilon \leq f\left(\mathcal{R}_{\Phi_{\text{hinge}}^{\text{sum}}}(h) - \mathcal{R}_{\Phi_{\text{hinge}}^{\text{sum}}}(h_0)\right).$$

Let $h = h_0$, then $f(0) \geq 1/6 - \epsilon$. Since $f$ is non-decreasing, for any $t \geq 0$ and $0 < \epsilon < \frac{1}{6}$, $f(t) \geq 1/6 - \epsilon$. Let $\epsilon \to 0$, we obtain that $f$ is lower bounded by $\frac{1}{6}$. $\qquad\square$

**Theorem 22** ($\mathcal{H}$-**consistency bound of** $\Phi_{\text{sq-hinge}}^{\text{sum}}$). *Suppose that $\mathcal{H}$ is symmetric and complete. Then, for any hypothesis $h \in \mathcal{H}$ and any distribution,*

$$\mathcal{R}_{\ell_{0-1}}(h) - \mathcal{R}_{\ell_{0-1},\mathcal{H}}^* \leq \left(\mathcal{R}_{\Phi_{\text{sq-hinge}}^{\text{sum}}}(h) - \mathcal{R}_{\Phi_{\text{sq-hinge}}^{\text{sum}},\mathcal{H}}^* + \mathcal{M}_{\Phi_{\text{sq-hinge}}^{\text{sum}},\mathcal{H}}\right)^{\frac{1}{2}} - \mathcal{M}_{\ell_{0-1},\mathcal{H}}. \qquad (37)$$

*Proof.* For the sum squared hinge loss $\Phi_{\text{sq-hinge}}^{\text{sum}}$, by (36), the conditional $\Phi_{\text{sq-hinge}}^{\text{sum}}$-risk can be expressed as follows:

$$\mathcal{C}_{\Phi_{\text{sq-hinge}}^{\text{sum}}}(h,x)$$

$$= \sum_{y \in \mathcal{Y}} p(x,y) \sum_{y' \neq y} \max\{0, 1 + h(x,y') - h(x,y)\}^2$$

$$= p(x,y_{\max}) \sum_{y' \neq y_{\max}} \max\{0, 1 + h(x,y') - h(x,y)\}^2 + \sum_{y \neq y_{\max}} p(x,y) \sum_{y' \neq y} \max\{0, 1 + h(x,y') - h(x,y)\}^2$$

$$= p(x,y_{\max}) \sum_{y' \neq y_{\max}} \max\{0, 1 + h(x,y') - h(x,y_{\max})\}^2 + \sum_{y \neq y_{\max}} p(x,y) \max\{0, 1 + h(x,y_{\max}) - h(x,y)\}^2$$

$$+ \sum_{y \neq y_{\max}} p(x,y) \sum_{y' \notin \{y_{\max},y\}} \max\{0, 1 + h(x,y') - h(x,y)\}^2$$

For any $h \in \mathcal{H}$, define the hypothesis $\overline{h}_\lambda \in \mathcal{H}$ by

$$\overline{h}_\lambda(x,y) = \begin{cases} h(x,y) & \text{if } y \neq y_{\max} \\ \lambda & \text{if } y = y_{\max} \end{cases}$$

for any $\lambda \in \mathbb{R}$. By the completeness of $\mathcal{H}$, the new hypothesis $\overline{h}_\lambda$ is in $\mathcal{H}$. Therefore, the minimal conditional $\Phi_{\text{sq-hinge}}^{\text{sum}}$-risk satisfies that for any $\lambda \in \mathbb{R}$, $\mathcal{C}_{\Phi_{\text{sq-hinge}}^{\text{sum}},\mathcal{H}}^*(x) \leq \mathcal{C}_{\Phi_{\text{sq-hinge}}^{\text{sum}}}(\overline{h}_\lambda,x)$. Let

$h \in \mathcal{H}$ be a hypothesis such that $\mathsf{h}(x) \neq y_{\max}$. By the definition and using the fact that $\mathsf{H}(x) = \mathcal{Y}$ when $\mathcal{H}$ is symmetric, we obtain

$$\Delta \mathcal{C}_{\Phi^{\mathrm{sum}}_{\mathrm{sq-hinge}}, \mathcal{H}}(h, x) = \mathcal{C}_{\Phi^{\mathrm{sum}}_{\mathrm{sq-hinge}}}(h, x) - \mathcal{C}^*_{\Phi^{\mathrm{sum}}_{\mathrm{sq-hinge}}, \mathcal{H}}(x)$$

$$\geq \mathcal{C}_{\Phi^{\mathrm{sum}}_{\mathrm{sq-hinge}}}(h, x) - \mathcal{C}_{\Phi^{\mathrm{sum}}_{\mathrm{sq-hinge}}}(\overline{h}_\lambda, x)$$

$$\geq p(x, y_{\max}) \max\{0, 1 + h(x, \mathsf{h}(x)) - h(x, y_{\max})\}^2 + p(x, \mathsf{h}(x)) \max\{0, 1 + h(x, y_{\max}) - h(x, \mathsf{h}(x))\}^2$$

$$- \frac{4 p(x, y_{\max}) p(x, \mathsf{h}(x))}{p(x, y_{\max} + p(x, \mathsf{h}(x))} \qquad \text{(taking supremum with respect to } \lambda\text{)}$$

$$\geq p(x, y_{\max}) + p(x, \mathsf{h}(x)) - \frac{4 p(x, y_{\max}) p(x, \mathsf{h}(x))}{p(x, y_{\max}) + p(x, \mathsf{h}(x))} \qquad (h(x, \mathsf{h}(x)) - h(x, y_{\max}) \geq 0)$$

$$= \frac{(p(x, y_{\max}) - p(x, \mathsf{h}(x)))^2}{p(x, y_{\max}) + p(x, \mathsf{h}(x))}$$

$$\geq \left(\max_{y \in \mathcal{Y}} p(x, y) - p(x, \mathsf{h}(x))\right)^2 \qquad (0 \leq p(x, y_{\max}) + p(x, \mathsf{h}(x)) \leq 1)$$

$$= (\Delta \mathcal{C}_{\ell_{0-1}, \mathcal{H}}(h, x))^2 \qquad \text{(by Lemma 3 and } \mathsf{H}(x) = \mathcal{Y})$$

$$\geq \left([\Delta \mathcal{C}_{\ell_{0-1}, \mathcal{H}}(h, x)]_\epsilon\right)^2 \qquad ([t]_\epsilon \leq t)$$

for any $\epsilon \geq 0$. Therefore, taking $\mathcal{P}$ be the set of all distributions, $\mathcal{H}$ be the symmetric and complete hypothesis set, $\epsilon = 0$ and $\Psi(t) = t^2$ in Theorem 4, or, equivalently, $\Gamma(t) = \sqrt{t}$ in Theorem 5, we obtain for any hypothesis $h \in \mathcal{H}$ and any distribution,

$$\mathcal{R}_{\ell_{0-1}}(h) - \mathcal{R}^*_{\ell_{0-1}, \mathcal{H}} \leq \left(\mathcal{R}_{\Phi^{\mathrm{sum}}_{\mathrm{sq-hinge}}}(h) - \mathcal{R}^*_{\Phi^{\mathrm{sum}}_{\mathrm{sq-hinge}}, \mathcal{H}} + \mathcal{M}_{\Phi^{\mathrm{sum}}_{\mathrm{sq-hinge}}, \mathcal{H}}\right)^{\frac{1}{2}} - \mathcal{M}_{\ell_{0-1}, \mathcal{H}}.$$

$\square$

**Theorem 23** ($\mathcal{H}$-**consistency bound of** $\Phi^{\mathrm{sum}}_{\exp}$)**.** *Suppose that $\mathcal{H}$ is symmetric and complete. Then, for any hypothesis $h \in \mathcal{H}$ and any distribution,*

$$\mathcal{R}_{\ell_{0-1}}(h) - \mathcal{R}^*_{\ell_{0-1}, \mathcal{H}} \leq \sqrt{2}\left(\mathcal{R}_{\Phi^{\mathrm{sum}}_{\exp}}(h) - \mathcal{R}^*_{\Phi^{\mathrm{sum}}_{\exp}, \mathcal{H}} + \mathcal{M}_{\Phi^{\mathrm{sum}}_{\exp}, \mathcal{H}}\right)^{\frac{1}{2}} - \mathcal{M}_{\ell_{0-1}, \mathcal{H}}. \tag{38}$$

*Proof.* For the sum exponential loss $\Phi^{\mathrm{sum}}_{\exp}$, by (36), the conditional $\Phi^{\mathrm{sum}}_{\exp}$-risk can be expressed as follows:

$$\mathcal{C}_{\Phi^{\mathrm{sum}}_{\exp}}(h, x) = \sum_{y \in \mathcal{Y}} p(x, y) \sum_{y' \neq y} \exp(h(x, y') - h(x, y))$$

$$= p(x, y_{\max}) \sum_{y' \neq y_{\max}} \exp(h(x, y') - h(x, y_{\max})) + \sum_{y \neq y_{\max}} p(x, y) \sum_{y' \neq y} \exp(h(x, y') - h(x, y))$$

$$= p(x, y_{\max}) \sum_{y' \neq y_{\max}} \exp(h(x, y') - h(x, y_{\max})) + \sum_{y \neq y_{\max}} p(x, y) \exp(h(x, y_{\max}) - h(x, y))$$

$$+ \sum_{y \neq y_{\max}} p(x, y) \sum_{y' \notin \{y_{\max}, y\}} \exp(h(x, y') - h(x, y))$$

For any $h \in \mathcal{H}$, define the hypothesis $\overline{h}_\lambda \in \mathcal{H}$ by

$$\overline{h}_\lambda(x, y) = \begin{cases} h(x, y) & \text{if } y \notin \{y_{\max}, \mathsf{h}(x)\} \\ \log(\exp[h(x, y_{\max})] + \lambda) & \text{if } y = \mathsf{h}(x) \\ \log(\exp[h(x, \mathsf{h}(x))] - \lambda) & \text{if } y = y_{\max} \end{cases}$$

for any $\lambda \in \mathbb{R}$. By the completeness of $\mathcal{H}$, the new hypothesis $\overline{h}_\lambda$ is in $\mathcal{H}$. Therefore, the minimal conditional $\Phi^{\mathrm{sum}}_{\exp}$-risk satisfies that for any $\lambda \in \mathbb{R}$, $\mathcal{C}^*_{\Phi^{\mathrm{sum}}_{\exp}, \mathcal{H}}(x) \leq \mathcal{C}_{\Phi^{\mathrm{sum}}_{\exp}}(\overline{h}_\lambda, x)$. Let $h \in \mathcal{H}$ be a hypothesis such that $\mathsf{h}(x) \neq y_{\max}$. By the definition and using the fact that $\mathsf{H}(x) = \mathcal{Y}$ when $\mathcal{H}$ is

symmetric, we obtain

$$\Delta \mathcal{C}_{\Phi^{\mathrm{sum}}_{\exp},\mathcal{H}}(h,x) = \mathcal{C}_{\Phi^{\mathrm{sum}}_{\exp}}(h,x) - \mathcal{C}^*_{\Phi^{\mathrm{sum}}_{\exp},\mathcal{H}}(x)$$

$$\geq \mathcal{C}_{\Phi^{\mathrm{sum}}_{\exp}}(h,x) - \mathcal{C}_{\Phi^{\mathrm{sum}}_{\exp}}(\overline{h}_\lambda,x)$$

$$\geq \sum_{y' \in \mathcal{Y}} e^{h(x,y')}\left[ p(x,y_{\max})e^{-h(x,y_{\max})} + p(x,\mathsf{h}(x))e^{-h(x,\mathsf{h}(x))} - \frac{\left(\sqrt{p(x,y_{\max})} + \sqrt{p(x,\mathsf{h}(x))}\right)^2}{e^{h(x,\mathsf{h}(x))} + e^{h(x,y_{\max})}} \right]$$

$$\text{(taking supremum with respect to } \lambda)$$

$$\geq \left( \sqrt{p(x,y_{\max})} - \sqrt{p(x,\mathsf{h}(x))} \right)^2 \qquad (h(x,\mathsf{h}(x)) \geq h(x,y_{\max}) \text{ and } p(x,\mathsf{h}(x)) \leq p(x,y_{\max}))$$

$$= \left( \frac{p(x,y_{\max}) - p(x,\mathsf{h}(x))}{\sqrt{p(x,\mathsf{h}(x))} + \sqrt{p(x,y_{\max})}} \right)^2$$

$$\geq \frac{1}{2}\left( \max_{y \in \mathcal{Y}} p(x,y) - p(x,\mathsf{h}(x)) \right)^2 \qquad (0 \leq p(x,y_{\max}) + p(x,\mathsf{h}(x)) \leq 1)$$

$$= \frac{1}{2}\left( \Delta \mathcal{C}_{\ell_{0-1},\mathcal{H}}(h,x) \right)^2 \qquad (\text{by Lemma 3 and } \mathsf{H}(x) = \mathcal{Y})$$

$$\geq \frac{1}{2}\left( [\Delta \mathcal{C}_{\ell_{0-1},\mathcal{H}}(h,x)]_\epsilon \right)^2 \qquad ([t]_\epsilon \leq t)$$

for any $\epsilon \geq 0$. Therefore, taking $\mathcal{P}$ be the set of all distributions, $\mathcal{H}$ be the symmetric and complete hypothesis set, $\epsilon = 0$ and $\Psi(t) = \frac{t^2}{2}$ in Theorem 4, or, equivalently, $\Gamma(t) = \sqrt{2t}$ in Theorem 5, we obtain for any hypothesis $h \in \mathcal{H}$ and any distribution,

$$\mathcal{R}_{\ell_{0-1}}(h) - \mathcal{R}^*_{\ell_{0-1},\mathcal{H}} \leq \sqrt{2}\left( \mathcal{R}_{\Phi^{\mathrm{sum}}_{\exp}}(h) - \mathcal{R}^*_{\Phi^{\mathrm{sum}}_{\exp},\mathcal{H}} + \mathcal{M}_{\Phi^{\mathrm{sum}}_{\exp},\mathcal{H}} \right)^{\frac{1}{2}} - \mathcal{M}_{\ell_{0-1},\mathcal{H}}.$$

$$\square$$

**Theorem 24** ($\mathcal{H}$-**consistency bound of** $\Phi^{\mathrm{sum}}_\rho$). *Suppose that $\mathcal{H}$ is symmetric and satisfies that for any $x \in \mathcal{X}$, there exists a hypothesis $h \in \mathcal{H}$ such that $|h(x,i) - h(x,j)| \geq \rho$ for any $i \neq j \in \mathcal{Y}$. Then, for any hypothesis $h \in \mathcal{H}$ and any distribution,*

$$\mathcal{R}_{\ell_{0-1}}(h) - \mathcal{R}^*_{\ell_{0-1},\mathcal{H}} \leq \mathcal{R}_{\Phi^{\mathrm{sum}}_\rho}(h) - \mathcal{R}^*_{\Phi^{\mathrm{sum}}_\rho,\mathcal{H}} + \mathcal{M}_{\Phi^{\mathrm{sum}}_\rho,\mathcal{H}} - \mathcal{M}_{\ell_{0-1},\mathcal{H}}. \tag{39}$$

*Proof.* For any $x \in \mathcal{X}$, we define $p_{[1]}(x), p_{[2]}(x), \ldots, p_{[c]}(x)$ by sorting the probabilities $\{p(x,y) : y \in \mathcal{Y}\}$ in increasing order. Similarly, for any $x \in \mathcal{X}$ and $h \in \mathcal{H}$, we define $h(x,\{1\}_x), h(x,\{2\}_x), \ldots, h(x,\{c\}_x)$ by sorting the scores $\{h(x,y) : y \in \mathcal{Y}\}$ in increasing order. In particular, we have

$$h(x,\{1\}_x) = \min_{y \in \mathcal{Y}} h(x,y), \quad h(x,\{c\}_x) = \max_{y \in \mathcal{Y}} h(x,y), \quad h(x,\{i\}_x) \leq h(x,\{j\}_j), \ \forall i \leq j.$$

If there is a tie for the maximum, we pick the label with the highest index under the natural ordering of labels, i.e. $\{c\}_x = \mathsf{h}(x)$. By the definition, the conditional $\Phi^{\mathrm{sum}}_\rho$-risk can be expressed as follows:

$$\mathcal{C}_{\Phi^{\mathrm{sum}}_\rho}(h,x) = \sum_{y \in \mathcal{Y}} p(x,y) \sum_{y' \neq y} \Phi_\rho(h(x,y) - h(x,y'))$$

$$= \sum_{i=1}^{c} p(x,\{i\}_x)\left[ \sum_{j=1}^{i-1} \Phi_\rho(h(x,\{i\}_x) - h(x,\{j\}_x)) + \sum_{j=i+1}^{c} \Phi_\rho(h(x,\{i\}_x) - h(x,\{j\}_x)) \right]$$

$$= \sum_{i=1}^{c} p(x,\{i\}_x)\left[ \sum_{j=1}^{i-1} \Phi_\rho(h(x,\{i\}_x) - h(x,\{j\}_x)) + c - i \right] \qquad (\Phi_\rho(t) = 1 \text{ for } t \leq 0)$$

By the assumption, there exists a hypotheses $h \in \mathcal{H}$ such that $|h(x,i) - h(x,j)| \geq \rho$ for any $i \neq j \in \mathcal{Y}$. Since $\mathcal{H}$ is symmetric, we can always choose $h^*$ among these hypotheses such that $h^*$ and $p(x)$

induce the same ordering of the labels, i.e. $p(x, \{k\}_x) = p_{[k]}(x)$ for any $k \in \mathcal{Y}$. Then, we have

$$
\begin{aligned}
\mathcal{C}^*_{\Phi^{\mathrm{sum}}_\rho, \mathcal{H}}(x) &\le \mathcal{C}_{\Phi^{\mathrm{sum}}_\rho}(h^*, x) \\
&= \sum_{i=1}^c p(x, \{i\}_x) \left[ \sum_{j=1}^{i-1} \Phi_\rho(h^*(x, \{i\}_x) - h^*(x, \{j\}_x)) + c - i \right] \\
&= \sum_{i=1}^c p(x, \{i\}_x)(c - i) \quad (|h^*(x, i) - h^*(x, j)| \ge \rho \text{ for any } i \ne j \text{ and } \Phi_\rho(t) = 0,\ \forall t \ge \rho) \\
&= \sum_{i=1}^c p_{[i]}(x)(c - i) \qquad\qquad (h^* \text{ and } p(x) \text{ induce the same ordering of the labels}) \\
&= c - \sum_{i=1}^c i\, p_{[i]}(x) \qquad\qquad (\textstyle\sum_{i=1}^c p_{[i]}(x) = 1)
\end{aligned}
$$

By the definition and using the fact that $\mathsf{H}(x) = \mathcal{Y}$ when $\mathcal{H}$ is symmetric, we obtain

$$
\begin{aligned}
&\Delta\mathcal{C}_{\Phi^{\mathrm{sum}}_\rho, \mathcal{H}}(h, x) \\
&= \mathcal{C}_{\Phi^{\mathrm{sum}}_\rho}(h, x) - \mathcal{C}^*_{\Phi^{\mathrm{sum}}_\rho, \mathcal{H}}(x) \\
&= \sum_{i=1}^c p(x, \{i\}_x) \left[ \sum_{j=1}^{i-1} \Phi_\rho(h(x, \{i\}_x) - h(x, \{j\}_x)) + c - i \right] - \left( c - \sum_{i=1}^c i\, p_{[i]}(x) \right) \\
&\ge \sum_{i=1}^c p(x, \{i\}_x)(c - i) - \left( c - \sum_{i=1}^c i\, p_{[i]}(x) \right) \qquad\qquad (\Phi_\rho \ge 0) \\
&= \sum_{i=1}^c i\, p_{[i]}(x) - \sum_{i=1}^c i\, p(x, \{i\}_x) \qquad\qquad (\textstyle\sum_{i=1}^c p(x, \{i\}) = 1) \\
&= \max_{y \in \mathcal{Y}} p(x, y) - p(x, \mathsf{h}(x)) + 
\begin{bmatrix} c-1 \\ c-1 \\ c-2 \\ \vdots \\ 1 \end{bmatrix} \cdot 
\begin{bmatrix} p_{[c]}(x) \\ p_{[c-1]}(x) \\ p_{[c-2]}(x) \\ \vdots \\ p_{[1]}(x) \end{bmatrix} - 
\begin{bmatrix} c-1 \\ c-1 \\ c-2 \\ \vdots \\ 1 \end{bmatrix} \cdot 
\begin{bmatrix} p(x, \{c\}_x) \\ p(x, \{c-1\}_x) \\ p(x, \{c-2\}_x) \\ \vdots \\ p(x, \{1\}_x) \end{bmatrix} \\
&\qquad\qquad\qquad\qquad\qquad\qquad (p_{[c]}(x) = \max_{y \in \mathcal{Y}} p(x, y) \text{ and } \{c\}_x = \mathsf{h}(x)) \\
&\ge \max_{y \in \mathcal{Y}} p(x, y) - p(x, \mathsf{h}(x)) \qquad\qquad\qquad\qquad (\text{by Lemma } 21) \\
&= \Delta\mathcal{C}_{\ell_{0-1}, \mathcal{H}}(h, x) \qquad\qquad\qquad\qquad\qquad (\text{by Lemma } 3) \\
&\ge \left[ \Delta\mathcal{C}_{\ell_{0-1}, \mathcal{H}}(h, x) \right]_\epsilon \qquad\qquad\qquad\qquad\qquad ([t]_\epsilon \le t)
\end{aligned}
$$

for any $\epsilon \ge 0$. Therefore, taking $\mathcal{P}$ be the set of all distributions, $\mathcal{H}$ be the symmetric hypothesis set, $\epsilon = 0$ and $\Psi(t) = t$ in Theorem 12, or, equivalently, $\Gamma(t) = t$ in Theorem 5, we obtain for any hypothesis $h \in \mathcal{H}$ and any distribution,

$$
\mathcal{R}_{\ell_{0-1}}(h) - \mathcal{R}^*_{\ell_{0-1}, \mathcal{H}} \le \mathcal{R}_{\Phi^{\mathrm{sum}}_\rho}(h) - \mathcal{R}^*_{\Phi^{\mathrm{sum}}_\rho, \mathcal{H}} + \mathcal{M}_{\Phi^{\mathrm{sum}}_\rho, \mathcal{H}} - \mathcal{M}_{\ell_{0-1}, \mathcal{H}}.
$$

$\square$

## K  Proof of $\mathcal{H}$-consistency bounds for constrained losses $\Phi^{\mathrm{cstnd}}$

Recall that $\mathsf{h}(x)$ and $y_{\max}$ are defined by $\mathsf{h}(x) = \operatorname{argmax}_{y \in \mathcal{Y}} h(x, y)$ and $y_{\max} = \operatorname{argmax}_{y \in \mathcal{Y}} p(x, y)$. If there is a tie, we pick the label with the highest index under the natural ordering of labels. The main idea of the proofs in this section is to leverage the constraint condition of Lee et al. (2004) that the scores sum to zero, and appropriately choose a hypothesis $\overline{h}$ that differs from $h$ only for its scores for $\mathsf{h}(x)$ and $y_{\max}$. Then, we can upper bound the minimal conditional risk by the conditional risk of $\overline{h}$ without requiring complicated computation of the minimal conditional risk. By the definition, the

conditional $\Phi^{\mathrm{cstnd}}$-risk can be expressed as follows:

$$\mathcal{C}_{\Phi^{\mathrm{cstnd}}}(h, x) = \sum_{y \in \mathcal{Y}} p(x, y) \sum_{y' \neq y} \Phi(-h(x, y'))$$

$$= \sum_{y \in \mathcal{Y}} \Phi(-h(x, y)) \sum_{y' \neq y} p(x, y') \tag{40}$$

$$= \sum_{y \in \mathcal{Y}} (1 - p(x, y)) \Phi(-h(x, y))$$

**Theorem 25** ($\mathcal{H}$-**consistency bound of** $\Phi^{\mathrm{cstnd}}_{\mathrm{hinge}}$). *Suppose that $\mathcal{H}$ is symmetric and complete. Then, for any hypothesis $h \in \mathcal{H}$ and any distribution,*

$$\mathcal{R}_{\ell_{0-1}}(h) - \mathcal{R}^*_{\ell_{0-1}, \mathcal{H}} \leq \mathcal{R}_{\Phi^{\mathrm{cstnd}}_{\mathrm{hinge}}}(h) - \mathcal{R}^*_{\Phi^{\mathrm{cstnd}}_{\mathrm{hinge}}, \mathcal{H}} + \mathcal{M}_{\Phi^{\mathrm{cstnd}}_{\mathrm{hinge}}, \mathcal{H}} - \mathcal{M}_{\ell_{0-1}, \mathcal{H}}. \tag{41}$$

*Proof.* For the constrained hinge loss $\Phi^{\mathrm{cstnd}}_{\mathrm{hinge}}$, by (40), the conditional $\Phi^{\mathrm{cstnd}}_{\mathrm{hinge}}$-risk can be expressed as follows:

$$\mathcal{C}_{\Phi^{\mathrm{cstnd}}_{\mathrm{hinge}}}(h, x) = \sum_{y \in \mathcal{Y}} (1 - p(x, y)) \max\{0, 1 + h(x, y)\}$$

$$= \sum_{y \in \{y_{\max}, h(x)\}} (1 - p(x, y)) \max\{0, 1 + h(x, y)\} + \sum_{y \notin \{y_{\max}, h(x)\}} (1 - p(x, y)) \max\{0, 1 + h(x, y)\}$$

Let $h \in \mathcal{H}$ be a hypothesis such that $h(x) \neq y_{\max}$. For any $x \in \mathcal{X}$, if $h(x, y_{\max}) \leq -1$, define the hypothesis $\overline{h} \in \mathcal{H}$ by

$$\overline{h}(x, y) = \begin{cases} h(x, y) & \text{if } y \notin \{y_{\max}, h(x)\} \\ h(x, y_{\max}) & \text{if } y = h(x) \\ h(x, h(x)) & \text{if } y = y_{\max}. \end{cases}$$

Otherwise, define the hypothesis $\overline{h} \in \mathcal{H}$ by

$$\overline{h}(x, y) = \begin{cases} h(x, y) & \text{if } y \notin \{y_{\max}, h(x)\} \\ -1 & \text{if } y = h(x) \\ h(x, y_{\max}) + h(x, h(x)) + 1 & \text{if } y = y_{\max}. \end{cases}$$

By the completeness of $\mathcal{H}$, the new hypothesis $\overline{h}$ is in $\mathcal{H}$ and satisfies that $\sum_{y \in \mathcal{Y}} \overline{h}(x, y) = 0$. Since $\sum_{y \in \mathcal{Y}} h(x, y) = 0$, there must be non-negative scores. By definition of $h(x)$ as a maximizer, we must thus have $h(x, h(x)) \geq 0$. Therefore, the minimal conditional $\Phi^{\mathrm{cstnd}}_{\mathrm{hinge}}$-risk satisfies:

$$\mathcal{C}^*_{\Phi^{\mathrm{cstnd}}_{\mathrm{hinge}}, \mathcal{H}}(x) \leq \mathcal{C}_{\Phi^{\mathrm{cstnd}}_{\mathrm{hinge}}}(\overline{h}, x)$$

$$= \begin{cases} (1 - p(x, y_{\max}))(1 + h(x, h(x))) + \sum_{y \notin \{y_{\max}, h(x)\}} (1 - p(x, y))(1 + h(x, y)) & \text{if } h(x, y_{\max}) \leq -1 \\ (1 - p(x, y_{\max}))(h(x, y_{\max}) + h(x, h(x)) + 2) + \sum_{y \notin \{y_{\max}, h(x)\}} (1 - p(x, y))(1 + h(x, y)) & \text{otherwise.} \end{cases}$$

By the definition and using the fact that $\mathsf{H}(x) = \mathcal{Y}$ when $\mathcal{H}$ is symmetric, we obtain

$$\Delta\mathcal{C}_{\Phi^{\mathrm{cstnd}}_{\mathrm{hinge}}, \mathcal{H}}(h, x) = \mathcal{C}_{\Phi^{\mathrm{cstnd}}_{\mathrm{hinge}}}(h, x) - \mathcal{C}^*_{\Phi^{\mathrm{cstnd}}_{\mathrm{hinge}}, \mathcal{H}}(x)$$

$$\geq \mathcal{C}_{\Phi^{\mathrm{cstnd}}_{\mathrm{hinge}}}(h, x) - \mathcal{C}_{\Phi^{\mathrm{cstnd}}_{\mathrm{hinge}}}(\overline{h}, x)$$

$$= (1 + h(x, h(x)))(p(x, y_{\max}) - p(x, h(x)))$$

$$\geq \max_{y \in \mathcal{Y}} p(x, y) - p(x, h(x)) \qquad (h(x, h(x)) \geq 0)$$

$$= \Delta\mathcal{C}_{\ell_{0-1}, \mathcal{H}}(h, x) \qquad (\text{by Lemma 3 and } \mathsf{H}(x) = \mathcal{Y})$$

$$\geq [\Delta\mathcal{C}_{\ell_{0-1}, \mathcal{H}}(h, x)]_\epsilon \qquad ([t]_\epsilon \leq t)$$

for any $\epsilon \geq 0$. Therefore, taking $\mathcal{P}$ be the set of all distributions, $\mathcal{H}$ be the symmetric and complete hypothesis set, $\epsilon = 0$ and $\Psi(t) = t$ in Theorem 4, or, equivalently, $\Gamma(t) = t$ in Theorem 5, we obtain for any hypothesis $h \in \mathcal{H}$ and any distribution,

$$\mathcal{R}_{\ell_{0-1}}(h) - \mathcal{R}^*_{\ell_{0-1}, \mathcal{H}} \leq \mathcal{R}_{\Phi^{\mathrm{cstnd}}_{\mathrm{hinge}}}(h) - \mathcal{R}^*_{\Phi^{\mathrm{cstnd}}_{\mathrm{hinge}}, \mathcal{H}} + \mathcal{M}_{\Phi^{\mathrm{cstnd}}_{\mathrm{hinge}}, \mathcal{H}} - \mathcal{M}_{\ell_{0-1}, \mathcal{H}}.$$

$\square$

**Theorem 26** ($\mathcal{H}$**-consistency bound of** $\Phi_{\text{sq-hinge}}^{\text{cstnd}}$**).** *Suppose that $\mathcal{H}$ is symmetric and complete. Then, for any hypothesis $h \in \mathcal{H}$ and any distribution,*

$$\mathcal{R}_{\ell_{0-1}}(h) - \mathcal{R}_{\ell_{0-1},\mathcal{H}}^* \leq \left( \mathcal{R}_{\Phi_{\text{sq-hinge}}^{\text{cstnd}}}(h) - \mathcal{R}_{\Phi_{\text{sq-hinge}}^{\text{cstnd}},\mathcal{H}}^* + \mathcal{M}_{\Phi_{\text{sq-hinge}}^{\text{cstnd}},\mathcal{H}} \right)^{\frac{1}{2}} - \mathcal{M}_{\ell_{0-1},\mathcal{H}}. \qquad (42)$$

*Proof.* For the constrained squared hinge loss $\Phi_{\text{sq-hinge}}^{\text{cstnd}}$, by (40), the conditional $\Phi_{\text{sq-hinge}}^{\text{cstnd}}$-risk can be expressed as follows:

$$\mathcal{C}_{\Phi_{\text{sq-hinge}}^{\text{cstnd}}}(h,x) = \sum_{y \in \mathcal{Y}}(1 - p(x,y))\max\{0, 1 + h(x,y)\}^2$$

$$= \sum_{y \in \{y_{\max}, \mathsf{h}(x)\}}(1 - p(x,y))\max\{0, 1 + h(x,y)\}^2 + \sum_{y \notin \{y_{\max}, \mathsf{h}(x)\}}(1 - p(x,y))\max\{0, 1 + h(x,y)\}^2$$

Let $h \in \mathcal{H}$ be a hypothesis such that $\mathsf{h}(x) \neq y_{\max}$. For any $x \in \mathcal{X}$, if $h(x, y_{\max}) \leq -1$, define the hypothesis $\overline{h} \in \mathcal{H}$ by

$$\overline{h}(x,y) = \begin{cases} h(x,y) & \text{if } y \notin \{y_{\max}, \mathsf{h}(x)\} \\ h(x, y_{\max}) & \text{if } y = \mathsf{h}(x) \\ h(x, \mathsf{h}(x)) & \text{if } y = y_{\max}. \end{cases}$$

Otherwise, define the hypothesis $\overline{h} \in \mathcal{H}$ by

$$\overline{h}(x,y) = \begin{cases} h(x,y) & \text{if } y \notin \{y_{\max}, \mathsf{h}(x)\} \\ \frac{1 - p(x, y_{\max})}{2 - p(x, y_{\max}) - p(x, \mathsf{h}(x))}(2 + h(x, y_{\max}) + h(x, \mathsf{h}(x))) - 1 & \text{if } y = \mathsf{h}(x) \\ \frac{1 - p(x, \mathsf{h}(x))}{2 - p(x, y_{\max}) - p(x, \mathsf{h}(x))}(2 + h(x, y_{\max}) + h(x, \mathsf{h}(x))) - 1 & \text{if } y = y_{\max}. \end{cases}$$

By the completeness of $\mathcal{H}$, the new hypothesis $\overline{h}$ is in $\mathcal{H}$ and satisfies that $\sum_{y \in \mathcal{Y}} \overline{h}(x,y) = 0$. Since $\sum_{y \in \mathcal{Y}} h(x,y) = 0$, there must be non-negative scores. By definition of $\mathsf{h}(x)$ as a maximizer, we must thus have $h(x, \mathsf{h}(x)) \geq 0$. Therefore, the minimal conditional $\Phi_{\text{sq-hinge}}^{\text{cstnd}}$-risk satisfies:

$$\mathcal{C}_{\Phi_{\text{sq-hinge}}^{\text{cstnd}},\mathcal{H}}^*(x) \leq \mathcal{C}_{\Phi_{\text{sq-hinge}}^{\text{cstnd}}}(\overline{h},x)$$

$$= \begin{cases} (1 - p(x, y_{\max}))(1 + h(x, \mathsf{h}(x)))^2 + \sum_{y \notin \{y_{\max}, \mathsf{h}(x)\}}(1 - p(x,y))(1 + h(x,y)) & \text{if } h(x, y_{\max}) \leq -1 \\ \frac{(1 - p(x, y_{\max}))(1 - p(x, \mathsf{h}(x)))(2 + h(x, y_{\max}) + h(x, \mathsf{h}(x)))^2}{2 - p(x, y_{\max}) - p(x,y)} + \sum_{y \notin \{y_{\max}, \mathsf{h}(x)\}}(1 - p(x,y))(1 + h(x,y)) & \text{otherwise.} \end{cases}$$

By the definition and using the fact that $\mathsf{H}(x) = \mathcal{Y}$ when $\mathcal{H}$ is symmetric, we obtain

$$\Delta\mathcal{C}_{\Phi_{\text{sq-hinge}}^{\text{cstnd}},\mathcal{H}}(h,x) = \mathcal{C}_{\Phi_{\text{sq-hinge}}^{\text{cstnd}}}(h,x) - \mathcal{C}_{\Phi_{\text{sq-hinge}}^{\text{cstnd}},\mathcal{H}}^*(x)$$

$$\geq \mathcal{C}_{\Phi_{\text{sq-hinge}}^{\text{cstnd}}}(h,x) - \mathcal{C}_{\Phi_{\text{sq-hinge}}^{\text{cstnd}}}(\overline{h},x)$$

$$= \begin{cases} (1 + h(x, \mathsf{h}(x)))^2(p(x, y_{\max}) - p(x, \mathsf{h}(x))) & \text{if } h(x, y_{\max}) \leq -1 \\ g(1 - p(x, y_{\max}), 1 - p(x, \mathsf{h}(x)), 1 + h(x, y_{\max}), 1 + h(x, \mathsf{h}(x))) & \text{otherwise} \end{cases}$$

$$\geq (1 + h(x, \mathsf{h}(x)))^2 \left( \max_{y \in \mathcal{Y}} p(x,y) - p(x, \mathsf{h}(x)) \right)^2 \qquad \text{(property of } g \text{ and } p(x, y_{\max}) \leq 1)$$

$$\geq \left( \max_{y \in \mathcal{Y}} p(x,y) - p(x, \mathsf{h}(x)) \right)^2 \qquad (h(x, \mathsf{h}(x)) \geq 0)$$

$$= \left( \Delta\mathcal{C}_{\ell_{0-1},\mathcal{H}}(h,x) \right)^2 \qquad \text{(by Lemma 3 and } \mathsf{H}(x) = \mathcal{Y})$$

$$\geq \left( \left[ \Delta\mathcal{C}_{\ell_{0-1},\mathcal{H}}(h,x) \right]_\epsilon \right)^2 \qquad ([t]_\epsilon \leq t)$$

for any $\epsilon \geq 0$, where $g(x,y,\alpha,\beta) = \frac{x^2\alpha^2 + y^2\beta^2 - 2xy\alpha\beta}{x+y} \geq \beta^2(x-y)^2$ when $0 \leq x \leq y \leq 1$, $x + y \geq 1$ and $1 \leq \alpha \leq \beta$. Therefore, taking $\mathcal{P}$ be the set of all distributions, $\mathcal{H}$ be the symmetric and complete hypothesis set, $\epsilon = 0$ and $\Psi(t) = t^2$ in Theorem 4, or, equivalently, $\Gamma(t) = \sqrt{t}$ in Theorem 5, we obtain for any hypothesis $h \in \mathcal{H}$ and any distribution,

$$\mathcal{R}_{\ell_{0-1}}(h) - \mathcal{R}_{\ell_{0-1},\mathcal{H}}^* \leq \left( \mathcal{R}_{\Phi_{\text{sq-hinge}}^{\text{cstnd}}}(h) - \mathcal{R}_{\Phi_{\text{sq-hinge}}^{\text{cstnd}},\mathcal{H}}^* + \mathcal{M}_{\Phi_{\text{sq-hinge}}^{\text{cstnd}},\mathcal{H}} \right)^{\frac{1}{2}} - \mathcal{M}_{\ell_{0-1},\mathcal{H}}.$$

$\square$

**Theorem 27** ($\mathcal{H}$-**consistency bound of** $\Phi_{\exp}^{\mathrm{cstnd}}$). *Suppose that $\mathcal{H}$ is symmetric and complete. Then, for any hypothesis $h \in \mathcal{H}$ and any distribution,*

$$\mathcal{R}_{\ell_{0-1}}(h) - \mathcal{R}_{\ell_{0-1},\mathcal{H}}^* \le \sqrt{2}\Big(\mathcal{R}_{\Phi_{\exp}^{\mathrm{cstnd}}}(h) - \mathcal{R}_{\Phi_{\exp}^{\mathrm{cstnd}},\mathcal{H}}^* + \mathcal{M}_{\Phi_{\exp}^{\mathrm{cstnd}},\mathcal{H}}\Big)^{\frac{1}{2}} - \mathcal{M}_{\ell_{0-1},\mathcal{H}}. \qquad (43)$$

*Proof.* For the constrained exponential loss $\Phi_{\exp}^{\mathrm{cstnd}}$, by (40), the conditional $\Phi_{\exp}^{\mathrm{cstnd}}$-risk can be expressed as follows:

$$\mathcal{C}_{\Phi_{\exp}^{\mathrm{cstnd}}}(h, x) = \sum_{y \in \mathcal{Y}} (1 - p(x, y)) \exp(h(x, y))$$

$$= \sum_{y \in \{y_{\max}, \mathsf{h}(x)\}} (1 - p(x, y)) \exp(h(x, y)) + \sum_{y \notin \{y_{\max}, \mathsf{h}(x)\}} \exp(h(x, y))$$

Let $h \in \mathcal{H}$ be a hypothesis such that $\mathsf{h}(x) \ne y_{\max}$. For any $x \in \mathcal{X}$, define the hypothesis $\overline{h}_\mu \in \mathcal{H}$ by

$$\overline{h}_\mu(x, y) = \begin{cases} h(x, y) & \text{if } y \notin \{y_{\max}, \mathsf{h}(x)\} \\ h(x, y_{\max}) + \mu & \text{if } y = \mathsf{h}(x) \\ h(x, \mathsf{h}(x)) - \mu & \text{if } y = y_{\max} \end{cases}$$

for any $\mu \in \mathbb{R}$. By the completeness of $\mathcal{H}$, the new hypothesis $\overline{h}_\mu$ is in $\mathcal{H}$ and satisfies that $\sum_{y \in \mathcal{Y}} \overline{h}_\mu(x, y) = 0$. Since $\sum_{y \in \mathcal{Y}} h(x, y) = 0$, there must be non-negative scores. By definition of $\mathsf{h}(x)$ as a maximizer, we must thus have $h(x, \mathsf{h}(x)) \ge 0$. Therefore, the minimal conditional $\Phi_{\exp}^{\mathrm{cstnd}}$-risk satisfies that for any $\mu \in \mathbb{R}$,

$$\mathcal{C}_{\Phi_{\exp}^{\mathrm{cstnd}},\mathcal{H}}^*(x) \le \mathcal{C}_{\Phi_{\exp}^{\mathrm{cstnd}}}(\overline{h}_\mu, x)$$
$$= (1 - p(x, y_{\max}))e^{h(x,\mathsf{h}(x))-\mu} + (1 - p(x, \mathsf{h}(x)))e^{h(x,y_{\max})+\mu} + \sum_{y \notin \{y_{\max}, \mathsf{h}(x)\}} (1 - p(x, y)) \exp(h(x, y)).$$

By the definition and using the fact that $\mathsf{H}(x) = \mathcal{Y}$ when $\mathcal{H}$ is symmetric, we obtain

$$\Delta\mathcal{C}_{\Phi_{\exp}^{\mathrm{cstnd}},\mathcal{H}}(h, x) = \mathcal{C}_{\Phi_{\exp}^{\mathrm{cstnd}}}(h, x) - \mathcal{C}_{\Phi_{\exp}^{\mathrm{cstnd}},\mathcal{H}}^*(x)$$

$$\ge \mathcal{C}_{\Phi_{\exp}^{\mathrm{cstnd}}}(h, x) - \mathcal{C}_{\Phi_{\exp}^{\mathrm{cstnd}}}(\overline{h}_\mu, x)$$

$$\ge \left(\sqrt{(1 - p(x, \mathsf{h}(x)))e^{h(x,\mathsf{h}(x))}} - \sqrt{(1 - p(x, y_{\max}))e^{h(x,y_{\max})}}\right)^2$$
$$\text{(taking supremum with respect to } \mu)$$

$$\ge e^{h(x,\mathsf{h}(x))}\left(\sqrt{(1 - p(x, \mathsf{h}(x)))} - \sqrt{(1 - p(x, y_{\max}))}\right)^2$$
$$(e^{h(x,\mathsf{h}(x))} \ge e^{h(x,y_{\max})} \text{ and } p(x, \mathsf{h}(x)) \le p(x, y_{\max}))$$

$$\ge \left(\sqrt{(1 - p(x, \mathsf{h}(x)))} - \sqrt{(1 - p(x, y_{\max}))}\right)^2 \qquad (h(x, \mathsf{h}(x)) \ge 0)$$

$$= \left(\frac{p(x, y_{\max}) - p(x, \mathsf{h}(x))}{\sqrt{(1 - p(x, \mathsf{h}(x)))} + \sqrt{(1 - p(x, y_{\max}))}}\right)^2$$

$$\ge \frac{1}{2}\left(\max_{y \in \mathcal{Y}} p(x, y) - p(x, \mathsf{h}(x))\right)^2 \qquad (0 \le p(x, y_{\max}) + p(x, \mathsf{h}(x)) \le 1)$$

$$= \frac{1}{2}(\Delta\mathcal{C}_{\ell_{0-1},\mathcal{H}}(h, x))^2 \qquad \text{(by Lemma 3 and } \mathsf{H}(x) = \mathcal{Y})$$

$$\ge \frac{1}{2}\big([\Delta\mathcal{C}_{\ell_{0-1},\mathcal{H}}(h, x)]_\epsilon\big)^2 \qquad ([t]_\epsilon \le t)$$

for any $\epsilon \ge 0$. Therefore, taking $\mathcal{P}$ be the set of all distributions, $\mathcal{H}$ be the symmetric and complete hypothesis set, $\epsilon = 0$ and $\Psi(t) = \frac{t^2}{2}$ in Theorem 4, or, equivalently, $\Gamma(t) = \sqrt{2t}$ in Theorem 5, we obtain for any hypothesis $h \in \mathcal{H}$ and any distribution,

$$\mathcal{R}_{\ell_{0-1}}(h) - \mathcal{R}_{\ell_{0-1},\mathcal{H}}^* \le \sqrt{2}\Big(\mathcal{R}_{\Phi_{\exp}^{\mathrm{cstnd}}}(h) - \mathcal{R}_{\Phi_{\exp}^{\mathrm{cstnd}},\mathcal{H}}^* + \mathcal{M}_{\Phi_{\exp}^{\mathrm{cstnd}},\mathcal{H}}\Big)^{\frac{1}{2}} - \mathcal{M}_{\ell_{0-1},\mathcal{H}}.$$

$\square$

**Theorem 28** ($\mathcal{H}$-**consistency bound of** $\Phi_\rho^{\mathrm{cstnd}}$). *Suppose that $\mathcal{H}$ is symmetric and satisfies that for any $x \in \mathcal{X}$, there exists a hypothesis $h \in \mathcal{H}$ such that $h(x,y) \leq -\rho$ for any $y \neq y_{\max}$. Then, for any hypothesis $h \in \mathcal{H}$ and any distribution,*

$$\mathcal{R}_{\ell_{0-1}}(h) - \mathcal{R}^*_{\ell_{0-1},\mathcal{H}} \leq \mathcal{R}_{\Phi_\rho^{\mathrm{cstnd}}}(h) - \mathcal{R}^*_{\Phi_\rho^{\mathrm{cstnd}},\mathcal{H}} + \mathcal{M}_{\Phi_\rho^{\mathrm{cstnd}},\mathcal{H}} - \mathcal{M}_{\ell_{0-1},\mathcal{H}}. \tag{44}$$

*Proof.* Since $\sum_{y \in \mathcal{Y}} h(x,y) = 0$, by definition of $\mathsf{h}(x)$ as a maximizer, we must thus have $h(x, \mathsf{h}(x)) \geq 0$. For the constrained $\rho$-margin loss $\Phi_\rho^{\mathrm{cstnd}}$, by (40), the conditional $\Phi_\rho^{\mathrm{cstnd}}$-risk can be expressed as follows:

$$
\begin{aligned}
\mathcal{C}_{\Phi_\rho^{\mathrm{cstnd}}}(h,x) &= \sum_{y \in \mathcal{Y}} (1 - p(x,y)) \min\left\{ \max\left\{ 0, 1 + \frac{h(x,y)}{\rho} \right\}, 1 \right\} \\
&= \sum_{y \in \mathcal{Y}: h(x,y) \geq 0} (1 - p(x,y)) + \sum_{y \in \mathcal{Y}: h(x,y) < 0} (1 - p(x,y)) \max\left\{ 0, 1 + \frac{h(x,y)}{\rho} \right\} \\
&\geq 1 - p(x, \mathsf{h}(x)) \\
&\geq 1 - \max_{y \in \mathcal{Y}} p(x,y).
\end{aligned}
$$

By the assumption, the equality can be achieved by some $h_\rho^* \in \mathcal{H}$ with the constraint $\sum_{y \in \mathcal{Y}} h(x,y) = 0$ such that $h_\rho^*(x,y) \leq -\rho$ for any $y \neq y_{\max}$ and $h_\rho^*(x, y_{\max}) = -\sum_{y' \neq y_{\max}} h_\rho^*(x,y') \geq 0$. Therefore, the minimal conditional $\Phi_\rho^{\mathrm{cstnd}}$-risk can be expressed as follows:

$$\mathcal{C}^*_{\Phi_\rho^{\mathrm{cstnd}},\mathcal{H}}(x) = 1 - \max_{y \in \mathcal{Y}} p(x,y).$$

By the definition and using the fact that $\mathsf{H}(x) = \mathcal{Y}$ when $\mathcal{H}$ is symmetric, we obtain

$$
\begin{aligned}
\Delta\mathcal{C}_{\Phi_\rho^{\mathrm{cstnd}},\mathcal{H}}(h,x) &= \mathcal{C}_{\Phi_\rho^{\mathrm{cstnd}}}(h,x) - \mathcal{C}^*_{\Phi_\rho^{\mathrm{cstnd}},\mathcal{H}}(x) \\
&= \sum_{y \in \mathcal{Y}: h(x,y) \geq 0} (1 - p(x,y)) + \sum_{y \in \mathcal{Y}: h(x,y) < 0} (1 - p(x,y)) \max\left\{ 0, 1 + \frac{h(x,y)}{\rho} \right\} - \left( 1 - \max_{y \in \mathcal{Y}} p(x,y) \right) \\
&\geq 1 - p(x, \mathsf{h}(x)) - \left( 1 - \max_{y \in \mathcal{Y}} p(x,y) \right) \\
&= \max_{y \in \mathcal{Y}} p(x,y) - p(x, \mathsf{h}(x)) \\
&= \Delta\mathcal{C}_{\ell_{0-1},\mathcal{H}}(h,x) &&\text{(by Lemma 3 and } \mathsf{H}(x) = \mathcal{Y}) \\
&\geq \left[ \Delta\mathcal{C}_{\ell_{0-1},\mathcal{H}}(h,x) \right]_\epsilon &&([t]_\epsilon \leq t)
\end{aligned}
$$

for any $\epsilon \geq 0$. Therefore, taking $\mathcal{P}$ be the set of all distributions, $\mathcal{H}$ be the symmetric hypothesis set, $\epsilon = 0$ and $\Psi(t) = t$ in Theorem 4, or, equivalently, $\Gamma(t) = t$ in Theorem 5, we obtain for any hypothesis $h \in \mathcal{H}$ and any distribution,

$$\mathcal{R}_{\ell_{0-1}}(h) - \mathcal{R}^*_{\ell_{0-1},\mathcal{H}} \leq \mathcal{R}_{\Phi_\rho^{\mathrm{cstnd}}}(h) - \mathcal{R}^*_{\Phi_\rho^{\mathrm{cstnd}},\mathcal{H}} + \mathcal{M}_{\Phi_\rho^{\mathrm{cstnd}},\mathcal{H}} - \mathcal{M}_{\ell_{0-1},\mathcal{H}}.$$

$\square$

# L  Proof of negative results for adversarial robustness

**Theorem 14** (**Negative results for convex functions**). *Fix $c = 2$. Suppose that $\Phi$ is convex and non-increasing, and $\mathcal{H}$ contains $0$ and satisfies the condition that there exists $x \in \mathcal{X}$ such that $\mathcal{H}_\gamma(x) \neq \varnothing$. If for a non-decreasing function $f: \mathbb{R}_+ \to \mathbb{R}_+$, the following $\mathcal{H}$-consistency bound holds for any hypothesis $h \in \mathcal{H}$ and any distribution $\mathcal{D}$:*

$$\mathcal{R}_{\ell_\gamma}(h) - \mathcal{R}^*_{\ell_\gamma,\mathcal{H}} \leq f\left( \mathcal{R}_{\widetilde{\ell}}(h) - \mathcal{R}^*_{\widetilde{\ell},\mathcal{H}} \right), \tag{16}$$

*then, $f$ is lower bounded by $\frac{1}{2}$, for $\widetilde{\ell} = \widetilde{\Phi}^{\mathrm{max}}$, $\widetilde{\Phi}^{\mathrm{sum}}$ and $\widetilde{\Phi}^{\mathrm{cstnd}}$.*

*Proof.* Consider the distribution that supports on a singleton domain $\{x\}$ with $x$ satisfying that $\mathcal{H}_\gamma(x) \neq \varnothing$. When $\mathcal{H}_\gamma(x) \neq \varnothing$, $\mathsf{H}_\gamma(x)$ is also non-empty. Take $y_1 \in \mathcal{H}_\gamma(x)$ and let $y_2 \neq y_1$. We define $p(x)$ as $p(x, y_1) = p(x, y_2) = \frac{1}{2}$. Let $h_0 = 0 \in \mathcal{H}$. By Lemma 11 and the fact that $\mathcal{H}_\gamma(x) \neq \varnothing$ and $y_1 \in \mathsf{H}_\gamma(x)$, the minimal conditional $\ell_\gamma$-risk is

$$\mathcal{R}^*_{\ell_\gamma, \mathcal{H}} = \mathcal{C}^*_{\ell_\gamma, \mathcal{H}}(x) = 1 - \max_{y \in \mathsf{H}_\gamma(x)} p(x, y) = 1 - p(x, y_1) = \frac{1}{2}.$$

For $h = h_0$, we have

$$\mathcal{R}_{\ell_\gamma}(h_0) = \mathcal{C}_{\ell_\gamma}(h_0, x) = \sum_{y \in \mathcal{Y}} p(x, y) \sup_{x': \|x - x'\|_p \leq \gamma} \mathbb{1}_{\rho_h(x', y) \leq 0} = 1.$$

For the adversarial max loss with non-increasing $\Phi$, the conditional $\widetilde{\Phi}^{\max}$-risk can be expressed as follows:

$$\begin{aligned}
\mathcal{C}_{\widetilde{\Phi}^{\max}}(h, x) &= \sum_{y \in \mathcal{Y}} p(x, y) \sup_{x': \|x - x'\|_p \leq \gamma} \Phi(\rho_h(x', y)) \\
&= \sum_{y \in \mathcal{Y}} p(x, y) \Phi\left( \inf_{x': \|x - x'\|_p \leq \gamma} \rho_h(x', y) \right) \\
&= \frac{1}{2} \Phi\left( \inf_{x': \|x - x'\|_p \leq \gamma} (h(x', y_1) - h(x', y_2)) \right) + \frac{1}{2} \Phi\left( \inf_{x': \|x - x'\|_p \leq \gamma} (h(x', y_2) - h(x', y_1)) \right) \\
&= \frac{1}{2} \Phi\left( \inf_{x': \|x - x'\|_p \leq \gamma} (h(x', y_1) - h(x', y_2)) \right) + \frac{1}{2} \Phi\left( - \sup_{x': \|x - x'\|_p \leq \gamma} (h(x', y_1) - h(x', y_2)) \right)
\end{aligned}$$

If $\Phi$ is convex and non-increasing, we obtain for any $h \in \mathcal{H}$,

$$\begin{aligned}
\mathcal{R}_{\widetilde{\Phi}^{\max}}(h) &= \mathcal{C}_{\widetilde{\Phi}^{\max}}(h, x) \\
&= \frac{1}{2} \Phi\left( \inf_{x': \|x - x'\|_p \leq \gamma} (h(x', y_1) - h(x', y_2)) \right) + \frac{1}{2} \Phi\left( \sup_{x': \|x - x'\|_p \leq \gamma} (h(x', y_1) - h(x', y_2)) \right) \\
&\geq \Phi\left( \frac{1}{2} \inf_{x': \|x - x'\|_p \leq \gamma} (h(x', y_1) - h(x', y_2)) - \frac{1}{2} \sup_{x': \|x - x'\|_p \leq \gamma} (h(x', y_1) - h(x', y_2)) \right) \\
&\hspace{8cm} (\Phi \text{ is convex}) \\
&\geq \Phi(0), \hspace{6cm} (\Phi \text{ is non-increasing})
\end{aligned}$$

where the equality can be achieved by $h_0$. Therefore,

$$\mathcal{R}^*_{\widetilde{\Phi}^{\max}, \mathcal{H}} = \mathcal{C}^*_{\widetilde{\Phi}^{\max}, \mathcal{H}}(x) = \mathcal{R}_{\widetilde{\Phi}^{\max}}(h_0) = \Phi(0).$$

If (16) holds for some non-decreasing function $f$ and $\widetilde{\ell} = \widetilde{\Phi}^{\max}$, then, we obtain for any $h \in \mathcal{H}$,

$$\mathcal{R}_{\ell_\gamma}(h) - \frac{1}{2} \leq f\left( \mathcal{R}_{\widetilde{\Phi}^{\max}}(h) - \Phi(0) \right).$$

Let $h = h_0$, then $f(0) \geq 1/2$. Since $f$ is non-decreasing, for any $t \geq 0$, $f(t) \geq 1/2$.

For the adversarial sum loss with non-increasing $\Phi$, the conditional $\widetilde{\Phi}^{\text{sum}}$-risk can be expressed as follows:

$$\begin{aligned}
\mathcal{C}_{\widetilde{\Phi}^{\text{sum}}}(h, x) &= \sum_{y \in \mathcal{Y}} p(x, y) \sup_{x': \|x - x'\|_p \leq \gamma} \sum_{y' \neq y} \Phi(h(x', y) - h(x', y')) \\
&= \frac{1}{2} \Phi\left( \inf_{x': \|x - x'\|_p \leq \gamma} (h(x', y_1) - h(x', y_2)) \right) + \frac{1}{2} \Phi\left( \inf_{x': \|x - x'\|_p \leq \gamma} (h(x', y_2) - h(x', y_1)) \right) \\
&= \frac{1}{2} \Phi\left( \inf_{x': \|x - x'\|_p \leq \gamma} (h(x', y_1) - h(x', y_2)) \right) + \frac{1}{2} \Phi\left( - \sup_{x': \|x - x'\|_p \leq \gamma} (h(x', y_1) - h(x', y_2)) \right)
\end{aligned}$$

If $\Phi$ is convex and non-increasing, we obtain for any $h \in \mathcal{H}$,

$$\mathcal{R}_{\widetilde{\Phi}^{\mathrm{sum}}}(h) = \mathcal{C}_{\widetilde{\Phi}^{\mathrm{sum}}}(h, x)$$

$$= \frac{1}{2}\Phi\left(\inf_{x':\|x-x'\|_p \leq \gamma}(h(x', y_1) - h(x', y_2))\right) + \frac{1}{2}\Phi\left(\sup_{x':\|x-x'\|_p \leq \gamma}(h(x', y_1) - h(x', y_2))\right)$$

$$\geq \Phi\left(\frac{1}{2}\inf_{x':\|x-x'\|_p \leq \gamma}(h(x', y_1) - h(x', y_2)) - \frac{1}{2}\sup_{x':\|x-x'\|_p \leq \gamma}(h(x', y_1) - h(x', y_2))\right)$$

$$\qquad\qquad\qquad\qquad\qquad\qquad\qquad\qquad\qquad\qquad\qquad\qquad\qquad (\Phi \text{ is convex})$$

$$\geq \Phi(0), \qquad\qquad\qquad\qquad\qquad\qquad\qquad\qquad\qquad (\Phi \text{ is non-increasing})$$

where the equality can be achieved by $h_0$. Therefore,

$$\mathcal{R}^*_{\widetilde{\Phi}^{\mathrm{sum}}, \mathcal{H}} = \mathcal{C}^*_{\widetilde{\Phi}^{\mathrm{sum}}, \mathcal{H}}(x) = \mathcal{R}_{\widetilde{\Phi}^{\mathrm{sum}}}(h_0) = \Phi(0).$$

If (16) holds for some non-decreasing function $f$ and $\widetilde{\ell} = \widetilde{\Phi}^{\mathrm{sum}}$, then, we obtain for any $h \in \mathcal{H}$,

$$\mathcal{R}_{\ell_\gamma}(h) - \frac{1}{2} \leq f\left(\mathcal{R}_{\widetilde{\Phi}^{\mathrm{sum}}}(h) - \Phi(0)\right).$$

Let $h = h_0$, then $f(0) \geq 1/2$. Since $f$ is non-decreasing, for any $t \geq 0$, $f(t) \geq 1/2$.

For the adversarial constrained loss with non-increasing $\Phi$, using the fact that $h(x, y_1) + h(x, y_2) = 0$, the conditional $\widetilde{\Phi}^{\mathrm{cstnd}}$-risk can be expressed as follows:

$$\mathcal{C}_{\widetilde{\Phi}^{\mathrm{cstnd}}}(h, x) = \sum_{y \in \mathcal{Y}} p(x, y) \sup_{x':\|x-x'\|_p \leq \gamma} \sum_{y' \neq y} \Phi(-h(x', y'))$$

$$= \frac{1}{2}\Phi\left(\inf_{x':\|x-x'\|_p \leq \gamma}(-h(x', y_2))\right) + \frac{1}{2}\Phi\left(\inf_{x':\|x-x'\|_p \leq \gamma}(-h(x', y_1))\right)$$

$$= \frac{1}{2}\Phi\left(\inf_{x':\|x-x'\|_p \leq \gamma}h(x', y_1)\right) + \frac{1}{2}\Phi\left(-\sup_{x':\|x-x'\|_p \leq \gamma}h(x', y_1)\right)$$

If $\Phi$ is convex and non-increasing, we obtain for any $h \in \mathcal{H}$,

$$\mathcal{R}_{\widetilde{\Phi}^{\mathrm{cstnd}}}(h) = \mathcal{C}_{\widetilde{\Phi}^{\mathrm{cstnd}}}(h, x)$$

$$= \frac{1}{2}\Phi\left(\inf_{x':\|x-x'\|_p \leq \gamma}h(x', y_1)\right) + \frac{1}{2}\Phi\left(-\sup_{x':\|x-x'\|_p \leq \gamma}h(x', y_1)\right)$$

$$\geq \Phi\left(\frac{1}{2}\inf_{x':\|x-x'\|_p \leq \gamma}h(x', y_1) - \frac{1}{2}\sup_{x':\|x-x'\|_p \leq \gamma}h(x', y_1)\right) \qquad (\Phi \text{ is convex})$$

$$\geq \Phi(0), \qquad\qquad\qquad\qquad\qquad\qquad\qquad\qquad\qquad (\Phi \text{ is non-increasing})$$

where the equality can be achieved by $h_0$. Therefore,

$$\mathcal{R}^*_{\widetilde{\Phi}^{\mathrm{cstnd}}, \mathcal{H}} = \mathcal{C}^*_{\widetilde{\Phi}^{\mathrm{cstnd}}, \mathcal{H}}(x) = \mathcal{R}_{\widetilde{\Phi}^{\mathrm{cstnd}}}(h_0) = \Phi(0).$$

If (16) holds for some non-decreasing function $f$ and $\widetilde{\ell} = \widetilde{\Phi}^{\mathrm{cstnd}}$, then, we obtain for any $h \in \mathcal{H}$,

$$\mathcal{R}_{\ell_\gamma}(h) - \frac{1}{2} \leq f\left(\mathcal{R}_{\widetilde{\Phi}^{\mathrm{cstnd}}}(h) - \Phi(0)\right).$$

Let $h = h_0$, then $f(0) \geq 1/2$. Since $f$ is non-decreasing, for any $t \geq 0$, $f(t) \geq 1/2$.

$\qquad\qquad\qquad\qquad\qquad\qquad\qquad\qquad\qquad\qquad\qquad\qquad\qquad\qquad\qquad\qquad \square$

# M   Proof of $\mathcal{H}$-consistency bounds for adversarial max losses $\widetilde{\Phi}^{\mathrm{max}}$

**Theorem 15** ($\mathcal{H}$-**consistency bound of** $\widetilde{\Phi}^{\mathrm{max}}_\rho$). *Suppose that $\mathcal{H}$ is symmetric. Then, for any hypothesis $h \in \mathcal{H}$ and any distribution $\mathcal{D}$, we have*

$$\mathcal{R}_{\ell_\gamma}(h) - \mathcal{R}^*_{\ell_\gamma, \mathcal{H}} \leq \frac{\mathcal{R}_{\widetilde{\Phi}^{\mathrm{max}}_\rho}(h) - \mathcal{R}^*_{\widetilde{\Phi}^{\mathrm{max}}_\rho, \mathcal{H}} + \mathcal{M}_{\widetilde{\Phi}^{\mathrm{max}}_\rho, \mathcal{H}}}{\min\left\{1, \frac{\inf_{x \in \{x \in \mathcal{X}: \mathcal{H}_\gamma(x) \neq \varnothing\}} \sup_{h \in \mathcal{H}_\gamma(x)} \inf_{x':\|x-x'\|_p \leq \gamma} \rho_h(x', \mathsf{h}(x))}{\rho}\right\}} - \mathcal{M}_{\ell_\gamma, \mathcal{H}}. \quad (18)$$

*Proof.* By the definition, the conditional $\widetilde{\Phi}_\rho^{\max}$-risk can be expressed as follows:

$$\mathcal{C}_{\widetilde{\Phi}_\rho^{\max}}(h,x) = \sum_{y\in\mathcal{Y}} p(x,y) \sup_{x':\|x-x'\|_p\leq\gamma} \Phi_\rho(\rho_h(x',y))$$

$$= \begin{cases} 1 - p(x,\mathsf{h}(x)) + \max\left\{0, 1 - \dfrac{\inf_{x':\|x-x'\|_p\leq\gamma}\rho_h(x',\mathsf{h}(x))}{\rho}\right\} p(x,\mathsf{h}(x)) & h\in\mathcal{H}_\gamma(x) \\ 1 & \text{otherwise.} \end{cases} \tag{45}$$

$$= \begin{cases} 1 - \min\left\{1, \dfrac{\inf_{x':\|x-x'\|_p\leq\gamma}\rho_h(x',\mathsf{h}(x))}{\rho}\right\} p(x,\mathsf{h}(x)) & h\in\mathcal{H}_\gamma(x) \\ 1 & \text{otherwise.} \end{cases}$$

Since $\mathcal{H}$ is symmetric, for any $x\in\mathcal{X}$, either for any $y\in\mathcal{Y}$,

$$\sup_{h\in\{h\in\mathcal{H}_\gamma(x):\mathsf{h}(x)=y\}} \inf_{x':\|x-x'\|_p\leq\gamma} \rho_h(x',\mathsf{h}(x)) = \sup_{h\in\mathcal{H}_\gamma(x)} \inf_{x':\|x-x'\|_p\leq\gamma} \rho_h(x',\mathsf{h}(x))$$

or $\mathcal{H}_\gamma(x)=\varnothing$. When $\mathcal{H}_\gamma(x)=\varnothing$, (45) implies that $\mathcal{C}^*_{\widetilde{\Phi}_\rho^{\max},\mathcal{H}}(x)=1$. When $\mathcal{H}_\gamma(x)\neq\varnothing$,

$$\mathcal{C}^*_{\widetilde{\Phi}_\rho^{\max},\mathcal{H}}(x) = 1 - \min\left\{1, \frac{\sup_{h\in\mathcal{H}_\gamma(x)} \inf_{x':\|x-x'\|_p\leq\gamma} \rho_h(x',\mathsf{h}(x))}{\rho}\right\} \max_{y\in\mathcal{Y}} p(x,y).$$

Therefore, the minimal conditional $\widetilde{\Phi}_\rho^{\max}$-risk can be expressed as follows:

$$\mathcal{C}^*_{\widetilde{\Phi}_\rho^{\max},\mathcal{H}}(x) = 1 - \min\left\{1, \frac{\sup_{h\in\mathcal{H}_\gamma(x)} \inf_{x':\|x-x'\|_p\leq\gamma} \rho_h(x',\mathsf{h}(x))}{\rho}\right\} \max_{y\in\mathcal{Y}} p(x,y) \mathbb{1}_{\mathcal{H}_\gamma(x)\neq\varnothing}$$

When $\mathcal{H}_\gamma(x)=\varnothing$, $\mathcal{C}_{\widetilde{\Phi}_\rho^{\max}}(h,x)\equiv 1$, which implies that $\Delta\mathcal{C}_{\widetilde{\Phi}_\rho^{\max},\mathcal{H}}(h,x)\equiv 0$. When $\mathcal{H}_\gamma(x)\neq\varnothing$, using the fact that $\mathsf{H}_\gamma(x)=\mathcal{Y}\iff\mathcal{H}_\gamma(x)\neq\varnothing$ when $\mathcal{H}$ is symmetric,

$$\Delta\mathcal{C}_{\widetilde{\Phi}_\rho^{\max},\mathcal{H}}(h,x) = \min\left\{1, \frac{\sup_{h\in\mathcal{H}_\gamma(x)} \inf_{x':\|x-x'\|_p\leq\gamma} \rho_h(x',\mathsf{h}(x))}{\rho}\right\} \max_{y\in\mathcal{Y}} p(x,y)$$

$$- \min\left\{1, \frac{\inf_{x':\|x-x'\|_p\leq\gamma} \rho_h(x',\mathsf{h}(x))}{\rho}\right\} p(x,\mathsf{h}(x))\mathbb{1}_{h\in\mathcal{H}_\gamma(x)}$$

$$\geq \min\left\{1, \frac{\sup_{h\in\mathcal{H}_\gamma(x)} \inf_{x':\|x-x'\|_p\leq\gamma} \rho_h(x',\mathsf{h}(x))}{\rho}\right\} \left(\max_{y\in\mathcal{Y}} p(x,y) - p(x,\mathsf{h}(x))\mathbb{1}_{h\in\mathcal{H}_\gamma(x)}\right)$$

$$= \min\left\{1, \frac{\sup_{h\in\mathcal{H}_\gamma(x)} \inf_{x':\|x-x'\|_p\leq\gamma} \rho_h(x',\mathsf{h}(x))}{\rho}\right\} \Delta\mathcal{C}_{\ell_\gamma,\mathcal{H}}(h,x)$$

$$\geq \min\left\{1, \frac{\sup_{h\in\mathcal{H}_\gamma(x)} \inf_{x':\|x-x'\|_p\leq\gamma} \rho_h(x',\mathsf{h}(x))}{\rho}\right\} \left[\Delta\mathcal{C}_{\ell_\gamma,\mathcal{H}}(h,x)\right]_\epsilon$$

$$\geq \min\left\{1, \frac{\inf_{x\in\{x\in\mathcal{X}:\mathcal{H}_\gamma(x)\neq\varnothing\}} \sup_{h\in\mathcal{H}_\gamma(x)} \inf_{x':\|x-x'\|_p\leq\gamma} \rho_h(x',\mathsf{h}(x))}{\rho}\right\} \left[\Delta\mathcal{C}_{\ell_\gamma,\mathcal{H}}(h,x)\right]_\epsilon$$

for any $\epsilon\geq 0$. Therefore, taking $\mathcal{P}$ be the set of all distributions, $\mathcal{H}$ be the symmetric hypothesis set, $\epsilon=0$ and

$$\Psi(t) = \min\left\{1, \frac{\inf_{x\in\{x\in\mathcal{X}:\mathcal{H}_\gamma(x)\neq\varnothing\}} \sup_{h\in\mathcal{H}_\gamma(x)} \inf_{x':\|x-x'\|_p\leq\gamma} \rho_h(x',\mathsf{h}(x))}{\rho}\right\} t$$

in Theorem 12, or, equivalently, $\Gamma(t)=\Psi^{-1}(t)$ in Theorem 13, we obtain for any hypothesis $h\in\mathcal{H}$ and any distribution,

$$\mathcal{R}_{\ell_\gamma}(h) - \mathcal{R}^*_{\ell_\gamma,\mathcal{H}} \leq \frac{\mathcal{R}_{\widetilde{\Phi}_\rho^{\max}}(h) - \mathcal{R}^*_{\widetilde{\Phi}_\rho^{\max},\mathcal{H}} + \mathcal{M}_{\widetilde{\Phi}_\rho^{\max},\mathcal{H}}}{\min\left\{1, \frac{\inf_{x\in\{x\in\mathcal{X}:\mathcal{H}_\gamma(x)\neq\varnothing\}} \sup_{h\in\mathcal{H}_\gamma(x)} \inf_{x':\|x-x'\|_p\leq\gamma} \rho_h(x',\mathsf{h}(x))}{\rho}\right\}} - \mathcal{M}_{\ell_\gamma,\mathcal{H}}.$$

$\square$

# N  Proof of $\mathcal{H}$-consistency bounds for adversarial sum losses $\widetilde{\Phi}^{\mathrm{sum}}$

**Theorem 16** ($\mathcal{H}$-**consistency bound of** $\widetilde{\Phi}_\rho^{\mathrm{sum}}$). *Assume that $\mathcal{H}$ is symmetric and that for any $x \in \mathcal{X}$, there exists a hypothesis $h \in \mathcal{H}$ inducing the same ordering of the labels for any $x' \in \{x' : \|x - x'\|_p \le \gamma\}$ and such that $\inf_{x' : \|x-x'\|_p \le \gamma} |h(x', i) - h(x', j)| \ge \rho$ for any $i \ne j \in \mathcal{Y}$. Then, for any hypothesis $h \in \mathcal{H}$ and any distribution $\mathcal{D}$, the following inequality holds:*

$$\mathcal{R}_{\ell_\gamma}(h) - \mathcal{R}_{\ell_\gamma, \mathcal{H}}^* \le \mathcal{R}_{\widetilde{\Phi}_\rho^{\mathrm{sum}}}(h) - \mathcal{R}_{\widetilde{\Phi}_\rho^{\mathrm{sum}}, \mathcal{H}}^* + \mathcal{M}_{\widetilde{\Phi}_\rho^{\mathrm{sum}}, \mathcal{H}} - \mathcal{M}_{\ell_\gamma, \mathcal{H}}. \tag{20}$$

*Proof.* For any $x \in \mathcal{X}$, we define $p_{[1]}(x), p_{[2]}(x), \ldots, p_{[c]}(x)$ by sorting the probabilities $\{p(x, y) : y \in \mathcal{Y}\}$ in increasing order. Similarly, for any $x \in \mathcal{X}$ and $h \in \mathcal{H}$, we define $h(x, \{1\}_x), h(x, \{2\}_x), \ldots, h(x, \{c\}_x)$ by sorting the scores $\{h(x, y) : y \in \mathcal{Y}\}$ in increasing order. In particular, we have

$$h(x, \{1\}_x) = \min_{y \in \mathcal{Y}} h(x, y), \quad h(x, \{c\}_x) = \max_{y \in \mathcal{Y}} h(x, y), \quad h(x, \{i\}_x) \le h(x, \{j\}_j), \ \forall i \le j.$$

If there is a tie for the maximum, we pick the label with the highest index under the natural ordering of labels, i.e. $\{c\}_x = \mathsf{h}(x)$. By the definition, the conditional $\widetilde{\Phi}_\rho^{\mathrm{sum}}$-risk can be expressed as follows:

$$
\begin{aligned}
\mathcal{C}_{\widetilde{\Phi}_\rho^{\mathrm{sum}}}(h, x) &= \sum_{y \in \mathcal{Y}} p(x, y) \sup_{x' : \|x-x'\|_p \le \gamma} \sum_{y' \ne y} \Phi_\rho(h(x', y) - h(x', y')) \\
&= \sum_{y \in \mathcal{Y}} \sup_{x' : \|x-x'\|_p \le \gamma} p(x, y) \sum_{y' \ne y} \Phi_\rho(h(x', y) - h(x', y')) \\
&= \sum_{i=1}^c \sup_{x' : \|x-x'\|_p \le \gamma} p(x, \{i\}_{x'}) \left[ \sum_{j=1}^{i-1} \Phi_\rho(h(x', \{i\}_{x'}) - h(x', \{j\}_{x'})) + \sum_{j=i+1}^c \Phi_\rho(h(x', \{i\}_{x'}) - h(x', \{j\}_{x'})) \right] \\
&= \sum_{i=1}^c \sup_{x' : \|x-x'\|_p \le \gamma} p(x, \{i\}_{x'}) \left[ \sum_{j=1}^{i-1} \Phi_\rho(h(x', \{i\}_{x'}) - h(x', \{j\}_{x'})) + c - i \right]
\end{aligned}
$$

$$\tag{46}$$

By the assumption, there exists a hypothesis $h \in \mathcal{H}$ inducing the same ordering of the labels for any $x' \in \{x' : \|x - x'\|_p \le \gamma\}$ and such that $\inf_{x' : \|x-x'\|_p \le \gamma} |h(x', i) - h(x', j)| \ge \rho$ for any $i \ne j \in \mathcal{Y}$, i.e. $\{k\}_{x'} = \{k\}_x$ for any $k \in \mathcal{Y}$ and $x' \in \{x' : \|x - x'\|_p \le \gamma\}$. Since $\mathcal{H}$ is symmetric, we can always choose $h^*$ among these hypotheses such that $h^*$ and $p(x)$ induce the same ordering of the labels, i.e. $p(x, \{k\}_x) = p_{[k]}(x)$ for any $k \in \mathcal{Y}$. Then, by (46), we have

$$
\begin{aligned}
\mathcal{C}_{\widetilde{\Phi}_\rho^{\mathrm{sum}}, \mathcal{H}}^*(x) \le\ & \mathcal{C}_{\widetilde{\Phi}_\rho^{\mathrm{sum}}}(h^*, x) \\
&= \sum_{i=1}^c \sup_{x' : \|x-x'\|_p \le \gamma} p(x, \{i\}_{x'}) \left[ \sum_{j=1}^{i-1} \Phi_\rho(h^*(x', \{i\}_{x'}) - h^*(x', \{j\}_{x'})) + c - i \right] \text{ (by (46))} \\
&= \sum_{i=1}^c \sup_{x' : \|x-x'\|_p \le \gamma} p(x, \{i\}_{x'})(c - i) \\
&\qquad (\inf_{x' : \|x-x'\|_p \le \gamma} |h^*(x', i) - h^*(x', j)| \ge \rho \text{ for any } i \ne j \text{ and } \Phi_\rho(t) = 0, \ \forall t \ge \rho) \\
&= \sum_{i=1}^c p(x, \{i\}_x)(c - i) \\
&\qquad (h^* \text{ induces the same ordering of the labels for any } x' \in \{x' : \|x - x'\|_p \le \gamma\}) \\
&= \sum_{i=1}^c p_{[i]}(x)(c - i) \qquad (h^* \text{ and } p(x) \text{ induce the same ordering of the labels}) \\
&= c - \sum_{i=1}^c i\, p_{[i]}(x) \qquad (\textstyle\sum_{i=1}^c p_{[i]}(x) = 1)
\end{aligned}
$$

By the assumption, $\mathcal{H}_\gamma(x) \neq \varnothing$ and $\mathsf{H}_\gamma(x) = \mathcal{Y}$ since $\mathcal{H}$ is symmetric. Thus, for any $h \in \mathcal{H}$,

$$
\begin{aligned}
&\Delta\mathcal{C}_{\widetilde{\Phi}_\rho^{\mathrm{sum}},\mathcal{H}}(h,x) \\
&= \mathcal{C}_{\widetilde{\Phi}_\rho^{\mathrm{sum}}}(h,x) - \mathcal{C}^*_{\widetilde{\Phi}_\rho^{\mathrm{sum}},\mathcal{H}}(x) \\
&= \sum_{i=1}^{c} \sup_{x':\|x-x'\|_p \leq \gamma} p(x,\{i\}_{x'}) \left[\sum_{j=1}^{i-1} \Phi_\rho(h(x',\{i\}_{x'}) - h(x',\{j\}_{x'})) + c - i\right] - \left(c - \sum_{i=1}^{c} i\, p_{[i]}(x)\right) \\
&\hspace{9cm} (\Phi_\rho(t) = 1 \text{ for } t \leq 0) \\
&\geq p(x,\mathsf{h}(x))\mathbb{1}_{h \notin \mathcal{H}_\gamma(x)} + \sum_{i=1}^{c} \sup_{x':\|x-x'\|_p \leq \gamma} p(x,\{i\}_{x'})(c-i) - \left(c - \sum_{i=1}^{c} i\, p_{[i]}(x)\right) \hspace{1cm} (\Phi_\rho \geq 0) \\
&\geq p(x,\mathsf{h}(x))\mathbb{1}_{h \notin \mathcal{H}_\gamma(x)} + \sum_{i=1}^{c} p(x,\{i\}_x)(c-i) - \left(c - \sum_{i=1}^{c} i\, p_{[i]}(x)\right) \quad \text{(lower bound the supremum)} \\
&= p(x,\mathsf{h}(x))\mathbb{1}_{h \notin \mathcal{H}_\gamma(x)} + \sum_{i=1}^{c} i\, p_{[i]}(x) - \sum_{i=1}^{c} i\, p(x,\{i\}_x) \hspace{2.5cm} (\sum_{i=1}^{c} p(x,\{i\}) = 1) \\
&= p(x,\mathsf{h}(x))\mathbb{1}_{h \notin \mathcal{H}_\gamma(x)} + \max_{y \in \mathcal{Y}} p(x,y) - p(x,\mathsf{h}(x)) + \begin{bmatrix} c-1 \\ c-1 \\ c-2 \\ \vdots \\ 1 \end{bmatrix} \cdot \begin{bmatrix} p_{[c]}(x) \\ p_{[c-1]}(x) \\ p_{[c-2]}(x) \\ \vdots \\ p_{[1]}(x) \end{bmatrix} - \begin{bmatrix} c-1 \\ c-1 \\ c-2 \\ \vdots \\ 1 \end{bmatrix} \cdot \begin{bmatrix} p(x,\{c\}_x) \\ p(x,\{c-1\}_x) \\ p(x,\{c-2\}_x) \\ \vdots \\ p(x,\{1\}_x) \end{bmatrix} \\
&\hspace{5cm} (p_{[c]}(x) = \max_{y \in \mathcal{Y}} p(x,y) \text{ and } \{c\}_x = \mathsf{h}(x)) \\
&\geq p(x,\mathsf{h}(x))\mathbb{1}_{h \notin \mathcal{H}_\gamma(x)} + \max_{y \in \mathcal{Y}} p(x,y) - p(x,\mathsf{h}(x)) \hspace{3cm} \text{(by Lemma 21)} \\
&= \max_{y \in \mathcal{Y}} p(x,y) - p(x,\mathsf{h}(x))\mathbb{1}_{h \in \mathcal{H}_\gamma(x)} \\
&= \Delta\mathcal{C}_{\ell_\gamma,\mathcal{H}}(h,x) \hspace{5cm} \text{(by Lemma 11 and } \mathsf{H}_\gamma(x) = \mathcal{Y}) \\
&\geq \left[\Delta\mathcal{C}_{\ell_\gamma,\mathcal{H}}(h,x)\right]_\epsilon \hspace{6.5cm} ([t]_\epsilon \leq t)
\end{aligned}
$$

for any $\epsilon \geq 0$. Therefore, taking $\mathcal{P}$ be the set of all distributions, $\mathcal{H}$ be the symmetric hypothesis set, $\epsilon = 0$ and $\Psi(t) = t$ in Theorem 12, or, equivalently, $\Gamma(t) = t$ in Theorem 13, we obtain for any hypothesis $h \in \mathcal{H}$ and any distribution,

$$
\mathcal{R}_{\ell_\gamma}(h) - \mathcal{R}^*_{\ell_\gamma,\mathcal{H}} \leq \mathcal{R}_{\widetilde{\Phi}_\rho^{\mathrm{sum}}}(h) - \mathcal{R}^*_{\widetilde{\Phi}_\rho^{\mathrm{sum}},\mathcal{H}} + \mathcal{M}_{\widetilde{\Phi}_\rho^{\mathrm{sum}},\mathcal{H}} - \mathcal{M}_{\ell_\gamma,\mathcal{H}}.
$$

$\square$

# O   Proof of $\mathcal{H}$-consistency bounds for adversarial constrained losses $\widetilde{\Phi}^{\mathrm{cstnd}}$

**Theorem 17** ($\mathcal{H}$-consistency bound of $\widetilde{\Phi}_\rho^{\mathrm{cstnd}}$). *Suppose that $\mathcal{H}$ is symmetric and satisfies that for any $x \in \mathcal{X}$, there exists a hypothesis $h \in \mathcal{H}$ with the constraint $\sum_{y \in \mathcal{Y}} h(x,y) = 0$ such that $\sup_{x':\|x-x'\|_p \leq \gamma} h(x',y) \leq -\rho$ for any $y \neq y_{\max}$. Then, for any hypothesis $h \in \mathcal{H}$ and any distribution,*

$$
\mathcal{R}_{\ell_\gamma}(h) - \mathcal{R}^*_{\ell_\gamma,\mathcal{H}} \leq \mathcal{R}_{\widetilde{\Phi}_\rho^{\mathrm{cstnd}}}(h) - \mathcal{R}^*_{\widetilde{\Phi}_\rho^{\mathrm{cstnd}},\mathcal{H}} + \mathcal{M}_{\widetilde{\Phi}_\rho^{\mathrm{cstnd}},\mathcal{H}} - \mathcal{M}_{\ell_\gamma,\mathcal{H}}. \tag{22}
$$

*Proof.* Define $y_{\max}$ by $y_{\max} = \mathrm{argmax}_{y \in \mathcal{Y}} p(x,y)$. If there is a tie, we pick the label with the highest index under the natural ordering of labels. Since $\sum_{y \in \mathcal{Y}} h(x,y) = 0$, by definition of $\mathsf{h}(x)$ as a maximizer, we must thus have $h(x,\mathsf{h}(x)) \geq 0$. By the definition, the conditional $\widetilde{\Phi}_\rho^{\mathrm{cstnd}}$-risk can be

expressed as follows:

$$\mathcal{C}_{\widetilde{\Phi}_\rho^{\text{cstnd}}}(h,x) = \sum_{y \in \mathcal{Y}} p(x,y) \sup_{x':\|x-x'\|_p \leq \gamma} \sum_{y' \neq y} \Phi_\rho(-h(x',y'))$$

$$= \sum_{y \in \mathcal{Y}} \sup_{x':\|x-x'\|_p \leq \gamma} p(x,y) \sum_{y' \neq y} \Phi_\rho(-h(x',y'))$$

$$= \sum_{y \in \mathcal{Y}} \sup_{x':\|x-x'\|_p \leq \gamma} p(x,y) \left[ \sum_{y' \neq y: h(x',y')>0} \Phi_\rho(-h(x',y')) + \sum_{y' \neq y: h(x',y') \leq 0} \Phi_\rho(-h(x',y')) \right]$$

$$\geq \sum_{y \neq h(x')} \sup_{x':\|x-x'\|_p \leq \gamma} p(x,y)$$

$$\geq 1 - \max_{y \in \mathcal{Y}} p(x,y). \qquad\qquad (\Phi_\rho \geq 0 \text{ and } \Phi_\rho(t) = 1 \text{ for } t \leq 0)$$

By the assumption, the equality can be achieved by some $h_\rho^* \in \mathcal{H}$ with the constraint $\sum_{y \in \mathcal{Y}} h(x,y) = 0$ such that $\sup_{x':\|x-x'\|_p \leq \gamma} h_\rho^*(x',y) \leq -\rho$ for any $y \neq y_{\max}$ and $h_\rho^*(x',y_{\max}) = -\sum_{y' \neq y_{\max}} h_\rho^*(x',y')$ for any $x' \in \{x' : \|x - x'\|_p \leq \gamma\}$. Therefore, the minimal conditional $\widetilde{\Phi}_\rho^{\text{cstnd}}$-risk can be expressed as follows:

$$\mathcal{C}_{\widetilde{\Phi}_\rho^{\text{cstnd}},\mathcal{H}}^*(x) = 1 - \max_{y \in \mathcal{Y}} p(x,y).$$

By the assumption, $\mathcal{H}_\gamma(x) \neq \varnothing$ and $\mathsf{H}_\gamma(x) = \mathcal{Y}$ since $\mathcal{H}$ is symmetric. Thus, for any $h \in \mathcal{H}$ with the constraint that $\sum_{y \in \mathcal{Y}} h(x,y) = 0$,

$$\Delta\mathcal{C}_{\widetilde{\Phi}_\rho^{\text{cstnd}},\mathcal{H}}(h,x) = \mathcal{C}_{\widetilde{\Phi}_\rho^{\text{cstnd}}}(h,x) - \mathcal{C}_{\widetilde{\Phi}_\rho^{\text{cstnd}},\mathcal{H}}^*(x)$$

$$= \sum_{y \in \mathcal{Y}} \sup_{x':\|x-x'\|_p \leq \gamma} p(x,y) \sum_{y' \neq y} \Phi_\rho(-h(x',y')) - \left(1 - \max_{y \in \mathcal{Y}} p(x,y)\right)$$

$$\geq \sum_{y \in \mathcal{Y}} p(x,y) \sum_{y' \neq y} \Phi_\rho(-h(x,y')) - \left(1 - \max_{y \in \mathcal{Y}} p(x,y)\right) \qquad \text{(lower bound the supremum)}$$

$$= \sum_{y \in \mathcal{Y}} (1 - p(x,y))\Phi_\rho(-h(x,y)) - \left(1 - \max_{y \in \mathcal{Y}} p(x,y)\right) \qquad \text{(swap } y \text{ and } y')$$

$$\geq p(x,h(x))\mathbb{1}_{h \notin \mathcal{H}_\gamma(x)} + 1 - p(x,h(x)) - \left(1 - \max_{y \in \mathcal{Y}} p(x,y)\right)$$

$$= \max_{y \in \mathcal{Y}} p(x,y) - p(x,h(x))\mathbb{1}_{h \in \mathcal{H}_\gamma(x)}$$

$$= \Delta\mathcal{C}_{\ell_\gamma,\mathcal{H}}(h,x) \qquad \text{(by Lemma 11 and } \mathsf{H}_\gamma(x) = \mathcal{Y})$$

$$\geq \left[\Delta\mathcal{C}_{\ell_\gamma,\mathcal{H}}(h,x)\right]_\epsilon \qquad ([t]_\epsilon \leq t)$$

for any $\epsilon \geq 0$. Therefore, taking $\mathcal{P}$ be the set of all distributions, $\mathcal{H}$ be the symmetric hypothesis set, $\epsilon = 0$ and $\Psi(t) = t$ in Theorem 12, or, equivalently, $\Gamma(t) = t$ in Theorem 13, we obtain for any hypothesis $h \in \mathcal{H}$ and any distribution,

$$\mathcal{R}_{\ell_\gamma}(h) - \mathcal{R}_{\ell_\gamma,\mathcal{H}}^* \leq \mathcal{R}_{\widetilde{\Phi}_\rho^{\text{cstnd}}}(h) - \mathcal{R}_{\widetilde{\Phi}_\rho^{\text{cstnd}},\mathcal{H}}^* + \mathcal{M}_{\widetilde{\Phi}_\rho^{\text{cstnd}},\mathcal{H}} - \mathcal{M}_{\ell_\gamma,\mathcal{H}}.$$

$$\square$$