# OpenReview forum: "Multi-Class $H$-Consistency Bounds"
_NeurIPS.cc/2022/Conference — NeurIPS 2022 Accept_

### Official Review · Reviewer_haLt · 2022-07-11

**Rating:** 4
**Confidence:** 3
**Soundness:** 3 good
**Presentation:** 3 good
**Contribution:** 2 fair

**Summary:**

The aim of this work is to study the H-consistency bounds for multi-class classification. To this end, the authors give a series bounds for surrogate multi-class losses, including max losses, sum losses and constrained loss. Then they extend their results to adversarial setting. Lastly, they prove that no non-trivial H-consistency bound can be given in some cases.

**Questions:**

As mentioned above, I have some concerns about the contribution of this work, so I would appreciate it if the authors can describe the novelty of their proof techniques.

**Ethics Review Area:**

["I don’t know"]

**Strengths And Weaknesses:**

Strength:
1. To my knowledge, this paper is the first work to study multi-class H-consistency.
2. This paper is well-written and easy to follow.
3. The theoretical results and proofs are sound, based on my judgement.

Weakness:
The H-consistency for binary setting has been well studied in [1], which is somewhat similar to this paper in structure. Most results are also generalized from [1]. Besides, some important aspects of [1] are not covered in this paper, e.g. the tightness of the bounds.


[1] Pranjal Awasthi, et al. "H-Consistency Estimation Error of Surrogate Loss Minimizers." *International Conference on Machine Learning* (2022).

---

> ### Author Response · Authors · 2022-08-02
> **Response to Reviewer haLt**
>
> Thank you for your comments. We have carefully addressed all the questions raised. Please find our responses below.
>
> **Weaknesses: The H-consistency for binary setting has been well studied in [1], which is somewhat similar to this paper in structure. Most results are also generalized from [1]. Besides, some important aspects of [1] are not covered in this paper, e.g. the tightness of the bounds.**
>
> **Questions: As mentioned above, I have some concerns about the contribution of this work, so I would appreciate it if the authors can describe the novelty of their proof techniques.**
>
> **Response:** Thank you for your comments. Let us first emphasize that we are of course referencing (Awasthi et al., ICML 2022) and giving full credit to that elegant previous work, some of which we use in this paper.
>
> However, the binary classification results of Awasthi et al. (2022) do not readily extend to the multi-class setting.  This is because the study of the conditional risk is more complex in the multi-class setting and because the form of the surrogate losses is more diverse.
>
> In general, in the multi-class setting, the analysis is more involved and requires entirely novel proof techniques. This is true even in the classical study of Bayes-consistency, where the multi-class setting (Tewari and Bartlett 2007) does not readily follow the binary case (Bartlett et al. 2006) and required an alternative analysis and new proofs.
>
> Note that in the multi-class setting, surrogate losses are more diverse: we distinguished max losses, sum losses, and constrained losses and presented an analysis of each of these categories with various auxiliary functions for each category.
>
> More specifically, the need for novel proof techniques stems from the following and the novelty of our proof techniques can be briefly described as follows:
>
> To use the general tools Theorem 1 and Theorem 2, we need to find $\Psi$ and $\Gamma$ such that the inequality conditions in these theorems hold. This requires us to characterize the conditional risk and the minimal conditional risk of the multi-class zero-one loss function and the corresponding ones for diverse surrogate loss functions in both the non-adversarial and adversarial scenario.
>
> Unlike the binary case, such a characterization in the multi-class setting is very difficult. For example, for the constrained loss, solving the minimal conditional risk given a hypothesis set is equivalent to solving a c-dimensional constrained optimization problem, which does not admit an analytical expression.
>
> Instead, in the binary case, solving the minimal conditional risk is equivalent to solving a minimization problem for a univariate function and the needed function $\Psi$ can be characterized explicitly by the $H$-estimation error transformation, as shown in (Awasthi et al., 2022).
>
> Unfortunately, such binary classification transformation tools cannot be adapted to the multi-class setting. Instead, in our proof for the multi-class setting, we adopt a new idea that avoids directly characterizing the explicit expression of the minimal conditional risk.
>
> For example, for the constrained loss, we leverage the constraint condition of Lee et al. (2004) that the scores sum to zero, and appropriately choose a hypothesis $\overline{h}$ that differs from $h$ only for its scores for $\sf{h}(x)$ and $y_{\max}$ (see 4 theorems and their proofs in Appendix K). Then, we can upper bound the minimal conditional risk by the conditional risk of $\overline{h}$ without having to derive the closed form of the minimal conditional risk. Therefore, the conditional regret of the surrogate loss can be lower bounded by that of the zero-one loss with an appropriate function $\Psi$. To the best of our knowledge, this proof idea and technique are entirely novel. We believe that they can be used for the analysis of other multi-class surrogate losses. Furthermore, all of our multi-class $H$-consistency results are new.
>
> Likewise, our proofs of the $H$-consistency bounds for sum losses for the squared hinge loss and exponential loss used a similarly new technique and idea, and so did the proof for the $\rho$-margin loss (Appendix J).
>
> Furthermore, note that we also present an analysis of the adversarial scenario, for which the multi-class proofs are also novel.
>
> Finally, note that our bounds in the multi-class setting are more general and that for c = 2 we recover the binary classification bounds of (Awasthi et al., 2022). Thus, our bounds benefit from the same tightness guarantees shown by Awasthi et al. (2022). A further analysis of the tightness of our guarantees in the multi-class setting is left to future work.

---

### Official Review · Reviewer_UKY2 · 2022-07-12

**Rating:** 7
**Confidence:** 1
**Soundness:** 4 excellent
**Presentation:** 3 good
**Contribution:** 3 good

**Summary:**

This work is interested in deriving theoretical guarantees on the generalization error for the 0-1 loss in a multi-class setting when the learner minimize a surrogate loss like the hinge loss. This problem has already been widely studied in the literature with notions like Bayes-consistency or H-consistency. However, the authors argue that all the previous notions either hold only for the full family of measurable functions and/or are asymptotic. In contrast, this work aims at deriving non-asymptotic guarantees of consistency for a specific set H called H-consistency bounds. A previous paper have investigated the case of binary classification. This paper shows similar results for the multi-class setting also proving negative results when such extensions were not possible.

**Questions:**

.

**Limitations:**

.

**Strengths And Weaknesses:**

The main contribution of the paper is to show H-consistency bounds (notion introduced in a previous paper) in the multi-class setting. Four surrogates are studied. The results are summarized in the three tables of the paper.
The paper is well written and the mathematical statements are clear and precise.
Nevertheless, I found the paper hard to follow. This may be due to my lack of familiarity with this kind of bounds.
The paper may benefit from moving some of the theorems to the appendices (keeping the tables unchanged) and using the remaining space to add more informal explanations, maybe focusing on one result.
The paper is clean but a bit arid in the current state.
I was not able to assess the relevance and the impact of this work.

---

> ### Author Response · Authors · 2022-08-02
> **Response to Reviewer UKY2**
>
> Thank you for your appreciation of our work and suggestions for improving its readability. We will take them all into account when preparing the final version. We wish to emphasize that our results are the first $H$-consistency bounds proven for the multi-class setting, even in the special case of $H$ being the family of all measurable functions, whether in the non-adversarial or adversarial setting. All of our results are novel, including our proof techniques, which are likely to be useful in the analysis of other such guarantees. Specifically, when $H$ is the family of all measurable functions, our quantitative bounds in Table 2 and Table 3 imply the asymptotic consistency results of those multi-class losses in (Tewari and Bartlett, 2007), which shows that our results are stronger and more significant.  We also provide bounds for multi-class losses using a non-convex auxiliary function, which were not studied in previous work on multi-class consistency. Therefore, our contributions are significant and novel. Below, please find responses to specific questions.
>
> **Strengths And Weaknesses: The main contribution of the paper is to show H-consistency bounds ...  The paper is clean but a bit arid in the current state. I was not able to assess the relevance and the impact of this work.**
>
> **Response:** Thank you for the comments. We fully understand that the paper looks theoretically dense. This is because we present an extensive set of results tackling both the non-adversarial and adversarial cases, with diverse surrogate losses including max losses, sum losses, and constrained losses. We have sought to discuss the strong motivation for this work and give intuitive explanations for the theoretical results presented in the paper, such as before or after all our main theorems. We will further discuss in the final version the significance of our results and their relevance to the choice of a surrogate loss for a given hypothesis set. We also wish to point you to our response to Reviewer HTLU for more discussions on the contributions of our work, and our response to Reviewer haLt for the novelty of our proof techniques.

---

### Official Review · Reviewer_HTLU · 2022-07-15

**Rating:** 7
**Confidence:** 3
**Soundness:** 3 good
**Presentation:** 4 excellent
**Contribution:** 3 good

**Summary:**

The target loss in multi-class classification problems is usually the 0/1 loss, while the learning algorithms usually optimize with respect to surrogate losses. The work provided a comprehensive study of how well an optimization problem with respect to surrogate losses represents the results using the target loss in multi-class classification problems. The work considered the max loss, the sum loss, and the constrained loss, and provide the corresponding bounds or negative results.

**Questions:**

For the negative result in Theorem 6, it is not yet clear to me whether the problem comes from using convex \Phi or having a h giving equal scores. Will the problem be solved if one removes such h in the hypothesis class?

How can one estimate the bounds? Can the results indicate which surrogate loss is a better choice for a given hypothesis set H?

Figure 1 showed \rho-margin loss but didn’t specify the value of \rho in the figure.

L175, L187: any distribution “D”

**Limitations:**

The reviewer has not yet seen the potential negative societal impact of the work.

**Strengths And Weaknesses:**

The paper is well presented. The basis of the results (Theorem 1 - Theorem 5) is a generalization of the results in Awasthi et al. (2022) from binary classification to multi-class classification. However, since loss functions used in binary classification and multi-class classification are different, most of the results in the following sections are new. The work provided a systematic and comprehensive study of surrogate losses in multi-class classification problems, which can most likely lead the community to a better understanding of the use of different surrogate losses under different problem settings.

---

> ### Author Response · Authors · 2022-08-02
> **Response to Reviewer HTLU**
>
> Thank you for your encouraging review. We will take your suggestions into account when preparing the final version. We wish to emphasize that our results are the first $H$-consistency bounds proven for the multi-class setting, even in the special case of $H$ being the family of all measurable functions, whether in the non-adversarial or adversarial setting. All of our results are novel, including our proof techniques, which are likely to be useful in the analysis of other such guarantees. Specifically, when $H$ is the family of all measurable functions, our quantitative bounds in Table 2 and Table 3 imply the asymptotic consistency results of those multi-class losses in (Tewari and Bartlett, 2007), which shows that our results are stronger and more significant.  We also provide bounds for multi-class losses using a non-convex auxiliary function, which were not studied in previous work on multi-class consistency. Therefore, our contributions are significant and novel. Below, please find responses to specific questions.
>
> **1. For the negative result in Theorem 6, it is not yet clear to me whether the problem comes from using convex \Phi or having a h giving equal scores. Will the problem be solved if one removes such h in the hypothesis class?**
>
> **Response:** That’s a good question! As with common input space and hypothesis sets, it is natural to assume that $H$ does contain a predictor $h$ assigning equal scores to all $y$s for some specific input $x$. We believe that the proof can be extended to cases where the predictor assigns approximately equal scores to all labels, but it would be interesting to generalize it beyond even such relaxed assumptions and we will pursue that in the future work. We conjecture that the convexity of $\Phi$ is the main culprit here.
>
> **2. How can one estimate the bounds? Can the results indicate which surrogate loss is a better choice for a given hypothesis set H?**
>
> **Response:** this is a fundamental question, which we believe this work is strongly contributing to. Our theory suggests three key ingredients for the choice of a surrogate loss: (1) the functional form of the $H$-consistency bound: for example, for constrained losses, the bound for the hinge loss admits a linear dependency, while that of the exponential and squared hinge losses admits a square-root dependency and thus a less favorable convergence rate; (2) smoothness of the loss and more generally its optimization virtues: for example, the hinge loss is less smooth than the exponential and squared hinge losses and, more generally, surrogate losses with more favorable bounds might lead to more difficult optimizations; in fact, the zero-one loss is a surrogate for itself with the tightest bound for any hypothesis set, however it is known to lead to NP-complete optimization problems for many common choices of $H$; (3) approximation properties of the surrogate loss function; for example, given a choice of $H$, the minimizability gap for a surrogate loss may be more or less favorable. Our quantitative $H$-consistency bounds can help select the most favorable surrogate loss among surrogate losses with good optimization merits and comparable approximation properties.
>
> **3. Figure 1 showed \rho-margin loss but didn’t specify the value of \rho in the figure.**
>
> **Response:** Thank you for pointing this out. In Figure 1, the value of $\rho$ is 0.8. We will clarify that in the final version.
>
> **4. L175, L187: any distribution “D”.**
>
> **Response:** Thank you for the suggestion. We will modify it accordingly in the final version.

---

> > ### Comment · Reviewer_HTLU · 2022-08-08
> > **Response**
> >
> > Thank you for the comprehensive answer. I maintain my score for the paper, but as suggested by other reviewers, it would be better to add more explanations or discussions to the results.

---

### Author Response · Authors · 2022-08-07
**General comment to all the reviewers**

Dear reviewers,

Thank you all for your useful feedback. We have carefully addressed all the questions raised. Please let us know if there is any other clarification we can provide and otherwise please update your ratings.

Thank you.

---

### Meta-Review · Area_Chair_87v4 · 2022-08-25

**Recommendation:** Accept
**Confidence:** Less certain

**Metareview:**

This work studies the generalization error in the multi-class classification setting for the minimizer of the ERM with a surrogate loss. The authors present non-asymptotic guarantees using the concept of H-consistency. The present work extends a previous work which was only considering binary classification. The reviewers have found the contribution important and relevant. They also found that the presentation could be improved (e.g., by better referencing to known proof techniques) to have the paper more accessible to non-expert. I do recommend acceptance and ask the authors to revise the paper accordingly for the camera-ready version.

**Award:**

No

---

### Decision · Program_Chairs · 2022-09-14

Accept